# Working at the limit: A review of thermodynamics and optimality of the Earth system

Axel Kleidon[1]

[1]Max-Planck-Institute for Biogeochemistry, Hans-Knoell-Str. 10, 07745 Jena, Germany

**Correspondence:** Axel Kleidon (akleidon@bgc-jena.mpg.de)

**Abstract.** Optimality concepts related to energy and entropy have long been proposed to govern Earth system processes, for instance in form of propositions that certain processes maximize or minimize entropy production. These concepts, however, remain quite obscure, seem contradictory to each other, and have so far been mostly disregarded. This review aims to clarify the role of thermodynamics and optimality in Earth system science by showing that these play a central role in how, and how much,

work can be derived from the solar forcing, and that this imposes a major constraint to the dynamics of dissipative structures of the Earth system. This is, however, not as simple as it may sound. It requires a consistent formulation of Earth system processes in thermodynamic terms, including their linkages and interactions. Thermodynamics then constrains the ability of the Earth system to derive work and generate free energy from the solar radiative forcing, which limits the ability to maintain motion, mass transport, geochemical cycling, and biotic activity. It thus limits directly the generation of atmospheric motion and other

processes indirectly through their need for transport. I demonstrate the application of this thermodynamic Earth system view by deriving first-order estimates associated with atmospheric motion, hydrologic cycling, and terrestrial productivity that agree very well with observations. This supports the notion that the emergent simplicity and predictability inherent in observed climatological variations can be attributed to these processes working as hard as they can, reflecting thermodynamic limits directly or indirectly. I discuss how this thermodynamic interpretation is consistent with established theoretical concepts in the

respective disciplines, interpret other optimality concepts in light of this thermodynamic Earth system view, and describe its utility for Earth system science.

## 1   Introduction

The Earth system is an incredibly complex system, with many processes interacting with each other, from the small, local scale up to the planetary scale. With human activity playing an increasing role, it appears that the system becomes even more

complicated. This may seem to make the Earth a highly unpredictable and chaotic system, with arbitrary evolutionary directions and outcomes. It would seem that the only contribution from physics to constrain the dynamics of this complex system comes from the basic conservation laws, as these provide the accounting basis for energy, mass, and momentum and other conserved quantities.

     Yet, on the other hand, we observe various forms of relatively simple, emergent patterns in the Earth system that reflect

highly predictable outcomes. Such emergent simplicity is, for instance, reflected in highly predictable seasonal and geographic

variations of temperature and precipitation that have led to climate classifications (e.g. Koeppen, 1900), in typical surface energy balance partitioning and associated hydrologic classification schemes, such as the Aridity index of Budyko (1974) that can be used to describe clear and predictable changes in partitioning with increasing aridity, or the well-documented variation of terrestrial biomes along gradients in climate (e.g. von Humboldt, 1845; Holdridge, 1947; Whittaker, 1962; Prentice et al., 1992). How does this simplicity emerge from the dynamics of such a complex system? It would seem that there are further constraints at play when it comes to such predictable aspects of the Earth system. Are these constraints arbitrary, too specific to the example being considered, or do they result from further physical constraints that are currently not in the focus of our attention?

The aim of this review is to demonstrate that it is thermodynamics which sets an additional, highly relevant constraint to the dynamics of the Earth system. This additional constraint is based on the explicit consideration of entropy and the second law of thermodynamics. Entropy is a key thermodynamic concept that describes, loosely speaking, how dispersed energy is at the microscopic scale of atoms and molecules (e.g., Atkins and de Paula, 2010). At this microscopic scale, energy is quantised, distributed over discrete number of states. It is through the introduction of entropy and its maximisation that the microscopic distribution of energy is linked to macroscopic variables, such as temperature, density, and concentrations, that are commonly used to describe Earth system processes. Its central importance concerns two aspects: first, it provides a fundamental direction for energy conversions towards higher entropy, and second, it sets hard limits to the magnitude of conversions, as reflected by the well-established Carnot limit. Every time energy is being converted on Earth, for instance, from solar radiation to heat upon absorption, or from heat to kinetic energy when motion is generated, overall, entropy can only stay the same or increase, imposing a constraint known as the second law of thermodynamics.

The extent to which Earth system processes increase entropy is described by their rates of entropy production. Taken together with the entropy exchange by radiation, these then yield entropy budgets in which the entropy exchange due to radiation is balanced by the various contributions to the entropy production within the system. Such entropy budgets have been estimated for the Earth system already for quite some time (e.g., Aoki, 1983; Peixoto et al., 1991; Li et al., 1994; Goody, 2000; Raymond, 2013), with tools being available to diagnose these (e.g., Lembo et al., 2019). How is the second law reflected in these budgets? Figure 1 provides an illustration of the major contributions to this budget. Solar radiation is emitted by the Sun at a very high temperature, with relatively short wavelengths in the visible light. At the Earth's orbit, this radiative flux is confined to a narrow solid angle, that is, it covers only a small fraction of the sky. When absorbed and re-emitted, this takes place at a much colder temperature, and the radiative flux covers a much wider solid angle. This constitutes by far the largest contribution to the entropy budget: the conversion of low entropy solar radiation emitted by the hot Sun into high entropy terrestrial radiation emitted by the cold Earth. This observation was, in fact, already noted by Boltzmann (1886).

To understand how the second law can impose a constraint on the dynamics of the Earth system, we need to look a little closer into how this entropy is being produced. When solar radiation is absorbed, the electromagnetic wave interacts with electrons or molecules with an uneven charge distribution. The energy associated with the electromagnetic wave gets absorbed, raising these charged particles into excited states. When these excited states decay, this typically takes place through a sequence of intermediate states, thereby emitting more photons, each representing less energy, degrading energy to levels of the kinetic

temperature of the environment. This decay is referred to as thermalisation (see box "Thermalisation" in Figure 1): solar radiation turns to electric energy and further into the random motion of particles we describe as temperature. This produces by far the most entropy in the Earth system, because of the huge temperature difference between the emission temperature of the Sun and the temperature at which solar radiation is absorbed by the Earth surface. There is some further degradation as the radiation that is emitted from the surface is re-absorbed by the atmosphere at an even lower temperature, which results in further entropy production, but at a much smaller magnitude (see box "Radiative transfer" in Figure 1).

There are, however, alternative pathways by which entropy is not being produced right away. These are shown by the yellow boxes in Figure 1 and describe processes that generate "free" energy, energy that can perform work and produces entropy when this energy is converted into heat by dissipative processes. This is the case when atmospheric motion is generated in form of kinetic energy by a heat engine. This energy is then dissipated into heat by friction, producing entropy. On Earth, this involves low temperature differences, so that this contribution to the entropy budget is much smaller than the contribution by thermalisation. There are two further means to generate free energy directly from solar radiation before it is thermalised, photosynthesis and photovoltaics (Figure 1). Photosynthesis uses the electrons excited by the absorption of solar radiation before they are thermalised to produce free energy, incorporates it into carbohydrates, which are then being dissipated by metabolic activities associated with life. Photovoltaics is different in the sense that it exports the excited electrons right away and delivers free energy in form of electricity that is dissipated into heat when consumed by human activities. Entropy is thus at the very core of how these processes generate free energy to drive dissipative activities of Earth system processes, no matter whether these are physical, biological, or technical in their nature.

How does the constraint imposed by the second law going to play out? A series of publications consider the role of the second law in climate science (Ozawa et al., 2003; Singh and O'Neill, 2022) and ecology (Chapman et al., 2016; Vallino and Algar, 2016; Nielsen et al., 2020), and related it to thermodynamic limits or optimality approaches, such as the proposed hypothesis of Maximum Entropy Production (MEP, e.g., Paltridge, 1975; Lorenz et al., 2001; Kleidon, 2004), maximum power (e.g., Lotka, 1922a, b; Odum and Pinkerton, 1955; Kleidon and Renner, 2013a), minimum entropy production (e.g., Prigogine, 1955; Essex, 1984), minimum dissipation (e.g., West et al., 1997, 1999) or energy expenditure (e.g., Rinaldo et al., 1992, 1996; Rodriguez-Iturbe and Rinaldo, 1997). While maximization and minimization seem rather contrary to each other, all of these propositions are related to each other. In the climatological mean, power balances dissipation. When dissipation releases heat at a certain temperature, this results in entropy production. Hence, the notions of maximising power, dissipation or entropy production are very closely related. On the other hand, the maximum power transfer theorem in electrical engineering states that maximum power can be extracted from the power source by minimizing dissipation within the source. So maximization and minimization may simply reflect different sides of the same coin, depending on which process one chooses to look at within the system of interest.

However, applications of these principles have typically been to specific systems, and it is yet unresolved how general such proposed optimality approaches are. The main application of MEP in climate science has been to atmospheric heat transport (Paltridge, 1975; Lorenz et al., 2001), while applications to other fields, such as hydrology, have remained at the conceptual (Kleidon and Schymanski, 2008) or semi-empirical level (Lin, 2010; Zehe et al., 2013). The theoretical basis for MEP that

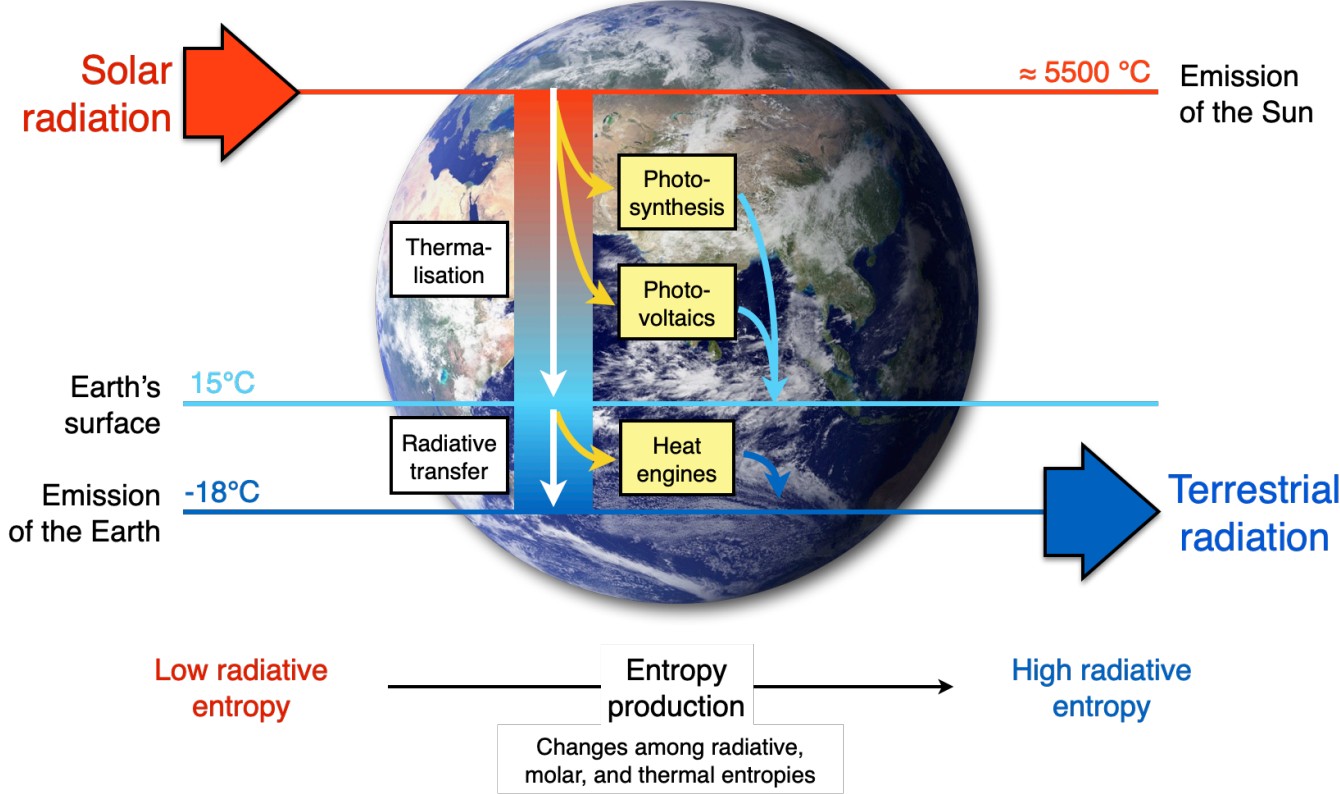

**Figure 1.** Schematic diagram illustrating the main thermodynamic setting of the whole planet. Energy is degraded and entropy is increased as solar radiation is converted into different forms of energy and entropy by Earth system processes and eventually is emitted to space as terrestrial radiation at much longer wavelengths. For the dynamics of the Earth system, it makes a big difference whether this entropy increase involves the generation of free energy, energy able to perform work (yellow boxes, "heat engines", "photosynthesis", and "photovoltaics") or not (white boxes, "thermalisation", "radiative transfer").

was developed by Dewar (2003; 2005a; 2005b) has been criticized (Grinstein and Linsker, 2007; Bruers, 2007), while the applications to atmospheric motion were criticized by Goody (2007), for instance for neglecting the role of planetary rotation rate on atmospheric motion. This raises a range of questions: Is the success of previous applications merely coincidental? What are the conditions that are necessary for such extremum principles to apply? What would be the associated dynamics 100 and feedbacks that result in the maximization (or minimization)? How can these approaches help us to better understand the functioning of the Earth system, particularly for conditions where less information is available (e.g., to past or future conditions of the Earth)? And, finally, can we link the emergence of simple, predictable patterns in the climatological mean to processes evolving to and operating at such limits and thermodynamically optimal states?

This review aims to clarify the applicability of such extremum principles in Earth system science and show that thermody- 105 namics indeed strongly constrains the functioning of the Earth system. In doing so, it will favour specific viewpoints instead of

providing an unbiased overview of the field. In the author's view, this bias has the advantage of providing a consistent picture of how thermodynamics and optimality applies to the whole Earth system that can resolve some of the seeming contradictions, it has the power to provide first-order estimates of its processes and its characteristics, and it can be linked to existing theories to show where thermodynamics can provide additional information and constraints. This can then serve as a basis for future discussions on the applicability of thermodynamic optimality in Earth system science and provide a perspective of its relevance. In particular, this review focuses on work and power rather than entropy production, and on the application of the maximum power limit to the generation of atmospheric motion, rather than each and every process. Since motion and its frictional dissipation are intimately linked to mass exchange, the intensity of hydrologic cycling as well as of biotic activity are shaped by this constraint as well, albeit indirectly. This, then, results in the simplicity of the emergent climatological patterns and associated optimal functioning. It suggests a more differentiated picture than the simple maximization of entropy production of a certain process. On the one hand, thermodynamics and maximization is very powerful in predicting a range of climatological patterns. On the other, the application of maximum power is quite specific to the setting of how atmospheric motion is generated, so that this case of maximization cannot be easily generalized to other Earth system processes. Simultaneously, it emphasizes the utility of viewing the Earth system in terms of the processes that derive free energy from sunlight that then allow work to be maintained within the system. The further conversions of this energy into other forms may then also be subjected to certain optimality approaches in energy conversion sequences.

To describe this rather different approach of the Earth as a system at work and that builds on a thermodynamic foundation requires three major components: (i) the concept of free energy, energy that results from work being done and that is free of entropy, (ii) the inclusion of the consequences of the dynamics, as these alter the boundary conditions under which free energy is generated, and (iii) optimality and resulting simplicity in climatological patterns being related primarily to the maximum power associated with the generation of motion and transport. This then results in a hierarchical view of the Earth system that is driven by the low-entropy solar energy input, from which free energy can be generated by only a few mechanisms (yellow boxes in Fig. 2). This drives sequences of further energy conversions and associated dynamics, which then feed back to the boundary conditions by transporting heat, changing rates, material properties, or radiative conditions. This picture of the Earth system is certainly more complex than simply stating that systems maximise or minimise some thermodynamic aspect. It nevertheless allows for a relatively simple way to be represented mathematically, it can be quantified, compared to observations and thus supported. We can then draw conclusions from it regarding the role of thermodynamic constraints and how these relate to optimality in the Earth system.

In the following, I first provide some basics of thermodynamics in Section 2 that are less common, but central for the thermodynamic description of the Earth system. These include the description of three forms of entropy which are obtained from the scaling of energy quanta in quantum physics (rather than just thermal entropy, which is central to classical thermodynamics), a definition of free energy that is somewhat different to the concept of Gibbs (or Helmholtz) free energy in classical thermodynamics, and a general derivation of thermodynamic limits from the first and second laws of thermodynamics. In Section 3, I will then describe how thermodynamics constrains Earth system processes directly or indirectly by describing applications to atmospheric motion, hydrologic cycling, and the productivity of the terrestrial biosphere. For each of the examples, I will

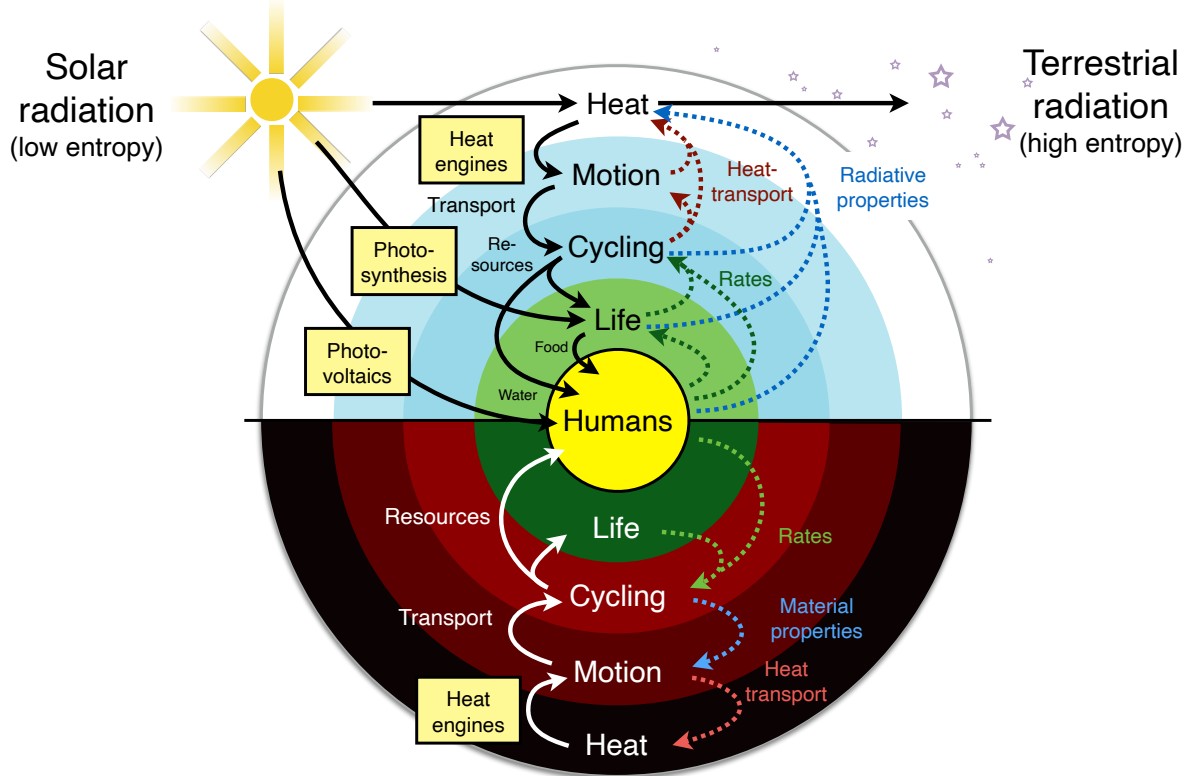

**Figure 2.** A hierarchical view of the Earth system in which thermodynamics constrains the process that generate free energy from low entropy sunlight (yellow boxes) that then fuels the dissipative dynamics of Earth system processes. Effects (dotted lines) of these processes feed back to the thermodynamic boundary conditions by transporting heat and changing radiative or material properties. Updated after Kleidon (2010; 2012; 2016).

first describe the examples in thermodynamic terms, describe how they relate to optimality, derive simple estimates, link these to established concepts in their respective fields, and then relate these examples back to the general picture shown in Figure 2. In Section 4 I then describe some of the limitations and potential future extensions, place previously proposed thermodynamic optimality approaches into the thermodynamic Earth system view described here and discuss potential applications of this approach to do simple, physics-based Earth system science. I close with a brief summary and conclusions.

## 2   Thermodynamics and Earth system functioning: What's missing?

All energy conversions within the Earth system are governed by the laws of thermodynamics. The first law of thermodynamics states that energy is overall conserved when it is converted from one form into another, while the second law requires that, overall, entropy can only remain the same or increase when energy is converted, although this increase may take place outside

the system when a non-isolated system is considered. Taken together, these laws set the limit to how much work can at best be derived from an energy source. A state of maximum entropy defines a reference point in thermodynamics, which corresponds to a state of thermodynamic equilibrium, a state from which no work can be derived. All of this is well-established textbook knowledge.

So which components are missing when we want to apply thermodynamics and optimality to Earth system science? In this
section, I describe a few components that are less commonly known but essential for describing a full picture of the Earth as a thermodynamic system: This includes more general definitions of entropy and free energy that go beyond heat, a more general derivation of limits that goes beyond specific thermodynamic cycles and is solely based on the laws of thermodynamics, and the inclusion of interactions with the boundary conditions in that limit.

Historically, the laws of thermodynamics were developed in the 19th century around the time of steam engines and the onset
of industrialisation, focusing on the energy conversions related to heat, or thermal energy, and mechanical work. Boltzmann's statistical interpretation of entropy in the latter part of the 19th century formed the basis to extend the concept of entropy to other forms of energy beyond heat, prominently reflected in Planck's theoretical derivation of the radiation laws. This, in turn, has led to the revolution of quantum physics at the onset of the 20th century. This has generalised the applicability of the laws of thermodynamics beyond the conversions of heat into mechanical work to all forms of conversion of energy into work. This
is relevant to the Earth system because its primary forcing, solar radiation, is an energy source of very low entropy that does not come in form of heat, but in form of electromagnetic waves. This is captured by the entropy of radiation, rather than heat, a concept that, while well established, is much less present in common climatology textbooks. Hence, the notion of entropy and the laws of thermodynamics apply to far more energy conversions than simply to the conversion of heat into mechanical work (in fact, it would therefore be more appropriate to refer to the *energetics* of the Earth system, rather than its *thermodynamics*).

The question of how dynamics then form in the Earth system relates primarily to how work can be derived from the solar forcing. The solar forcing represents an energy source of very low entropy, as it is in thermodynamic disequilibrium with the thermal conditions of the Earth system. From this disequilibrium, work is derived, which sustains the dynamics of Earth system processes, before this energy is dissipated and degrades to higher entropy. Essentially all Earth system processes, such as atmospheric motion, flows and transport processes, chemical transformations, metabolic activities, and socioeconomic
activities of human societies, are sustained by work being done and reflect thermodynamic disequilibrium in different forms. Interior processes within the Earth perform work, but with much lower magnitude, so that they are omitted here (but see, e.g., Dyke et al., 2011, for an application). Thermodynamics and optimality of Earth system processes thus relates intimately to the question of how much work can be derived from the solar forcing.

In the following, the concepts just described are provided in greater detail, as these are typically not treated in textbooks of
180 classical thermodynamics and of Earth system science. The description is aimed at a general level for a broader readership. For a fuller and more detailed description, the reader may be referred to textbooks in non-equilibrium thermodynamics (Kondepudi and Prigogine, 1998), literature on radiation entropy (Landsberg and Tonge, 1979; Kabelac, 1994; Wu and Liu, 2010), or the application of these concepts to Earth system science (Kleidon, 2016).

## 2.1 Three forms of entropy

Entropy is one of the key concepts in thermodynamics. Its meaning has evolved during the formulation of thermodynamics in the 19th century, and with the advent of quantum physics its meaning has extended well beyond the initial scope within classical thermodynamics. This extension is important when we describe the thermodynamics of the whole Earth system because some processes, like photosynthesis and photovoltaics, use solar radiation as a low-entropy energy source that does not involve heat, but use radiation directly and thereby avoid its thermalization.

Originally, entropy was introduced in thermodynamics by Clausius, expressing a change of entropy as the change in heat divided by the temperature at which it is added or removed. The concept of entropy obtained a mechanistic interpretation when Boltzmann developed his kinetic gas theory, in which he interpreted entropy as the probability for distributing a given amount of energy, represented by discrete amounts or quanta, across different states of a certain number of molecules. A state of maximum entropy then becomes the most probable macroscopic state of the gas. Planck extended this approach to radiation by introducing photons as the carriers of discrete quanta of energy in electromagnetic waves. He then used maximum entropy to derive radiation laws. The success of this approach set the basis for the revolution in physics at the onset of the 20th century and amounted in the development of quantum physics, with the well-established notion that energy at the molecular scale comes in discrete, quantised amounts. This broadened the role of entropy as it applies to forms of energy well beyond heat, and thus beyond the scope of classical thermodynamics. Entropy in physics thus originates from the quantisation of energy at the molecular scale.

When we deal with physical variables that characterise Earth system processes, we do not want to deal with these microscopic details of how energy is distributed at the molecular scale. This is where entropy comes into play (Figure 3). Formally, entropy is defined as a macroscopic variable that describes the probability of distributing a given amount of energy (hence, a certain number of quanta) over a certain number of quantum states. Since both are discrete, they can be counted, and hence, this defines a probability. This probability is given by Boltzmann's famous equation, $S = k_b \log W$: entropy $S$ is directly proportional to the logarithm of this probability $W$, with the proportionality described by the Boltzmann constant, $k_b$, a fundamental constant in physics. The state of thermodynamic equilibrium is then the state with maximum entropy, which simply means the most probable distribution of energy at the microscopic scale. From this state of maximum entropy, macroscopic variables such as temperature or pressure can then be derived. Note that Boltzmann's expression is sometimes also used to define information entropy. This concept, however, does not relate to the distribution of energy at the microscopic scale of quantum physics and is outside the scope of this paper.

A critical point to recognise is that when we deal with Earth system processes, we do not just deal with heat and Boltzmann's application of entropy to an ideal gas. Entropy also applies to energy distributions associated with photons and electrons, yielding three distinct forms of entropy (Figure 3): radiation entropy, molar entropy, and the more common form of thermal entropy. Radiative processes are associated with distributing energy quanta in form of photons with different frequencies, with the blackbody spectrum representing the distribution at thermodynamic equilibrium. Phase transitions and chemical conversions alter the energy levels of electrons in atoms and molecules, thereby yielding different values of specific molar entropies of

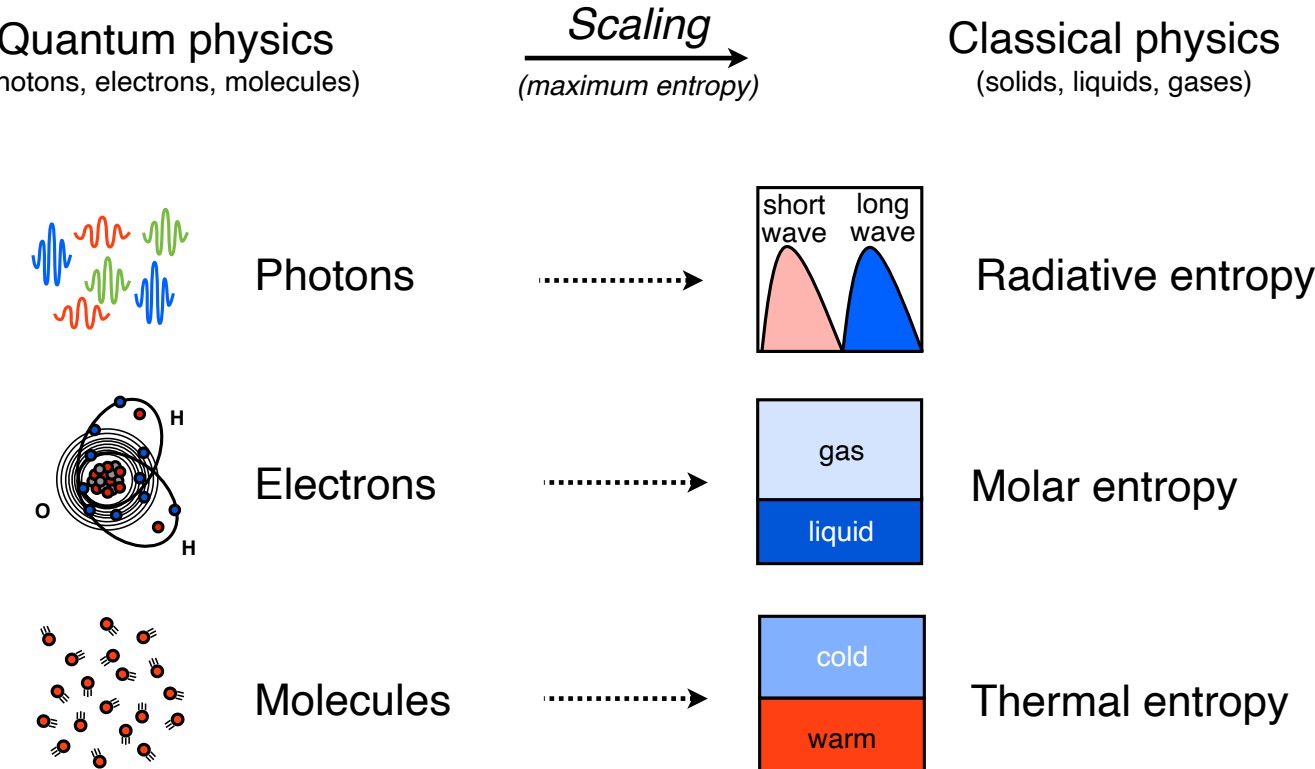

**Figure 3.** Schematic diagram to illustrate three types of entropy that follow from the quantisation of energy and that are relevant to Earth system processes. Energy is quantised and distributed over a discrete number of states at the scale of atoms and molecules (left). The associated scaling to variables in classical physics (right) is done by assuming maximum entropy. Depending on where the energy is stored at the microscopic scale, there are three different forms of entropy, associated with energy being distributed over photons, electrons, and molecules. After Kleidon (2016).

substances. Processes dealing with heat and pressure are associated with energy quanta being distributed over different vibrational, rotational or translational modes of molecules associated with heat. These three forms of entropy are at thermodynamic
equilibrium associated with the different distribution functions in statistical physics: the Bose-Einstein statistics for photons, the Fermi-Dirac statistics for electrons, and the Maxwell-Boltzmann statistics for random molecular motion.

    This distinction between different forms of entropy is hidden in assumptions that are implicitly made when describing the conversion of the energy contained in solar radiation by Earth system processes. When solar radiation is absorbed and heats the surface, it is not just the energy that changes its form from radiative to thermal energy. Also, the entropy changes its form, from
entropy representing how photons are being distributed over wavelengths (radiative entropy) to how energy is distributed over the random vibrations and motions of molecules (thermal entropy). The second law of thermodynamics nevertheless applies. We usually do not recognize these changes in forms, because commonly, local thermodynamic equilibrium is being assumed

during the conversion. When the Stefan-Boltzmann law is used to calculate the emission of radiation from a surface, it implicitly assumes thermodynamic equilibrium between the kinetic temperature of the surface associated with random vibrations and motion of molecules and the spectral composition of the emitted radiation.

The relevance of this distinction becomes clearer when we focus on how much work can be derived from converting solar radiation (cf. Figure 1). There, it makes a substantial difference if solar radiation is first thermalized and turned into heat after it has been absorbed and then into work (as is the case for a heat engine), or if it is used to change electronic states in photochemical or photovoltaic conversions, generates chemical or electric energy before it turns into heat (as are the cases for photosynthesis and human-made photovoltaic technology). The latter conversions can derive substantially more work from the solar energy source than the former. We will get back to this important difference further below.

## 2.2 From energy to free energy

The ability to perform work is closely connected to the term free energy. Free energy is commonly associated with a somewhat narrower definition and specific expressions in thermodynamics, such as those for the Helmholtz or Gibbs free energy. Yet, when we think of it more broadly, it refers to energy that resulted from work being performed. Examples within the Earth system include the work done by acceleration to maintain motion against friction and the work of lifting water vapour against gravity that maintains hydrologic cycling. This work results in free energy being generated, which can then be converted into other forms of free energy. In the two examples, free energy is the kinetic energy of motion and the potential energy of water at a certain height.

A more general definition of free energy is that it represents energy at the macroscopic scale that has no entropy associated with it. This notion of free energy links closely to the term "exergy" that is used in some engineering literature (e.g., Rant, 1956; Petela, 1964; Bejan, 2002; Rosen and Scott, 2003; Herrmann, 2006; Tailleux, 2013) – free energy can be seen as consisting only of exergy. Its description as being free of entropy becomes clearer in the derivation of the Carnot limit further below (Sect. 2.3). This use of free energy is consistent with the more common notions of Gibbs (or Helmholtz) free energy in classical thermodynamics, except that we focus here on the difference of the Gibbs (or Helmholtz) free energy with respect to its minimum. These provide specific expressions for specific forms of free energy, but there is no general expression that applies to all forms of free energy. Free energy applies also to its further conversions that do not directly involve entropy. Examples are the conversion of kinetic energy of winds to electricity by wind turbines, or the conversion of potential energy of rainwater into the kinetic energy of river flow. This requires a more general notion of free energy for its various forms within the Earth system than its limited definition in classical thermodynamics. The definition is therefore kept at this qualitative level here.

Thermodynamics and the Earth system context enter here as these constrain the generation of free energy from solar radiation. The generation of free energy represents work (or power, being equal to work over time), using an energy source of low entropy. The resulting free energy can be converted further into other forms, or it can be dissipated, that is, converted back into heat (or radiation), with a certain entropy that is typically higher (Figure 4).

We can express the resulting dynamics in form of a free energy budget, with free energy $A$ reflecting the budgeting of generation $G$, dissipation $D$, or the conversion into other forms, $G_{conv}$:

$$\frac{dA}{dt} = G - D - G_{conv} \tag{1}$$

For instance, the kinetic energy of the atmosphere as a form of free energy, $A_{ke}$, is generated out of differential radiative heating and cooling at different temperatures, which represent energy sources and sinks of different entropies. The free energy

is dissipated by friction (i.e., representing frictional dissipation, $D$) back into heat at a certain temperature, thus it gains a certain entropy again, or it is converted further into other forms of free energy ($G_{conv}$), for instance, by generating waves at the ocean surface or by generating electricity when used by wind turbines. Ultimately, these converted forms of energy eventually end up as heat (or radiation) as well when they are dissipated. At the end, the energy is emitted into space in form of terrestrial radiation, with a certain, higher radiative entropy.

Free energy budgets, as expressed by Eq. 1, are central to describe the dynamics of Earth system processes. Atmospheric dynamics are about free energy in kinetic form. Hydrologic cycling is about free energy in potential form associated with water at a certain height, which can then be converted into the kinetic energy of falling raindrops or into horizontal river flow once it rained down on land. The dynamics of ecosystems are about free energy in chemical form associated with the carbohydrates that make up biomass. These all can be formulated in terms of free energy budgets, a concept that is rarely used in Earth

system science, but that has the potential to provide a unified description because we deal with comparable processes and quantities, and processes where thermodynamics imposes restrictions on the magnitude and the direction of these conversions. These budgets are very different from energy budgets commonly used in climatology, as those merely deal with the accounting of heating and cooling terms in the budgeting of thermal energy. It is also not entirely clear how these budgets should be formulated consistently (see, e.g., Tailleux (2010) regarding the role of buoyancy for the ocean energy cycle).

We can now link these free energy dynamics back to the more general thermodynamic concepts of disequilibrium and entropy production. When free energy is available, it means that it can be converted back into heat, so it has entropy attached to it. That is, free energy within a system represents thermodynamic disequilibrium because when it is dissipated, the added heat results in an increase in entropy in the system. Hence, the dissipative terms of the free energy dynamics are closely associated with the intensity of entropy production that are due to the dynamics within a system. Yet, when focusing only at entropy

production in steady state, one cannot distinguish whether this entropy was produced by dynamics that involves free energy (such as motion, labeled "Path B" in Figure 4) or not (such as diffusion, thermalisation, or radiative transfer, labeled "Path A" in Figure 4). At the planetary scale, these two paths link to the difference illustrated by the white and yellow boxes in Fig. 1. We will see that this distinction is relevant when exploring thermodynamic optimality principles because it will allow us to understand why and how a system may evolve to a thermodynamically optimal state.

**Figure 4.** Schematic diagram to illustrate the difference between Earth system processes that involve free energy from those that do not. Free energy is generated by work being performed (or power, work over time) from an energy source with low entropy. This energy can then be further converted into other forms of free energy (not shown), or it is dissipated back into heat, which then has higher entropy. This allows us to differentiate processes that merely dissipate and produce entropy, such as thermalisation, diffusion or radiative transfer (labeled as "Path A", like the white boxes of thermalisation and radiative transfer in Fig. 1), from those that involve free energy and macroscopic dynamics, such as atmospheric motion (labeled as "Path B", like the yellow boxes in Fig. 1).

## 2.3 Limits on generating free energy

When we now ask how much free energy can at best be derived from a certain energy source, the laws of thermodynamics set a firm upper limit. The most common limit is the Carnot limit of a heat engine, that is, the limit of how much work, or free energy, can be derived from a heating source. In textbooks, e.g., the Feynman lectures on physics or Kondepudi and Prigogine (1998), this limit is typically derived for a specific, so-called thermodynamic cycle, involving different steps of expansion and compression. However, the Carnot limit can, in fact, be derived more generally and more directly from the first and second laws of thermodynamics in a few steps, also for a flux-driven system in disequilibrium, and also for energy sources of low entropy other than heat.

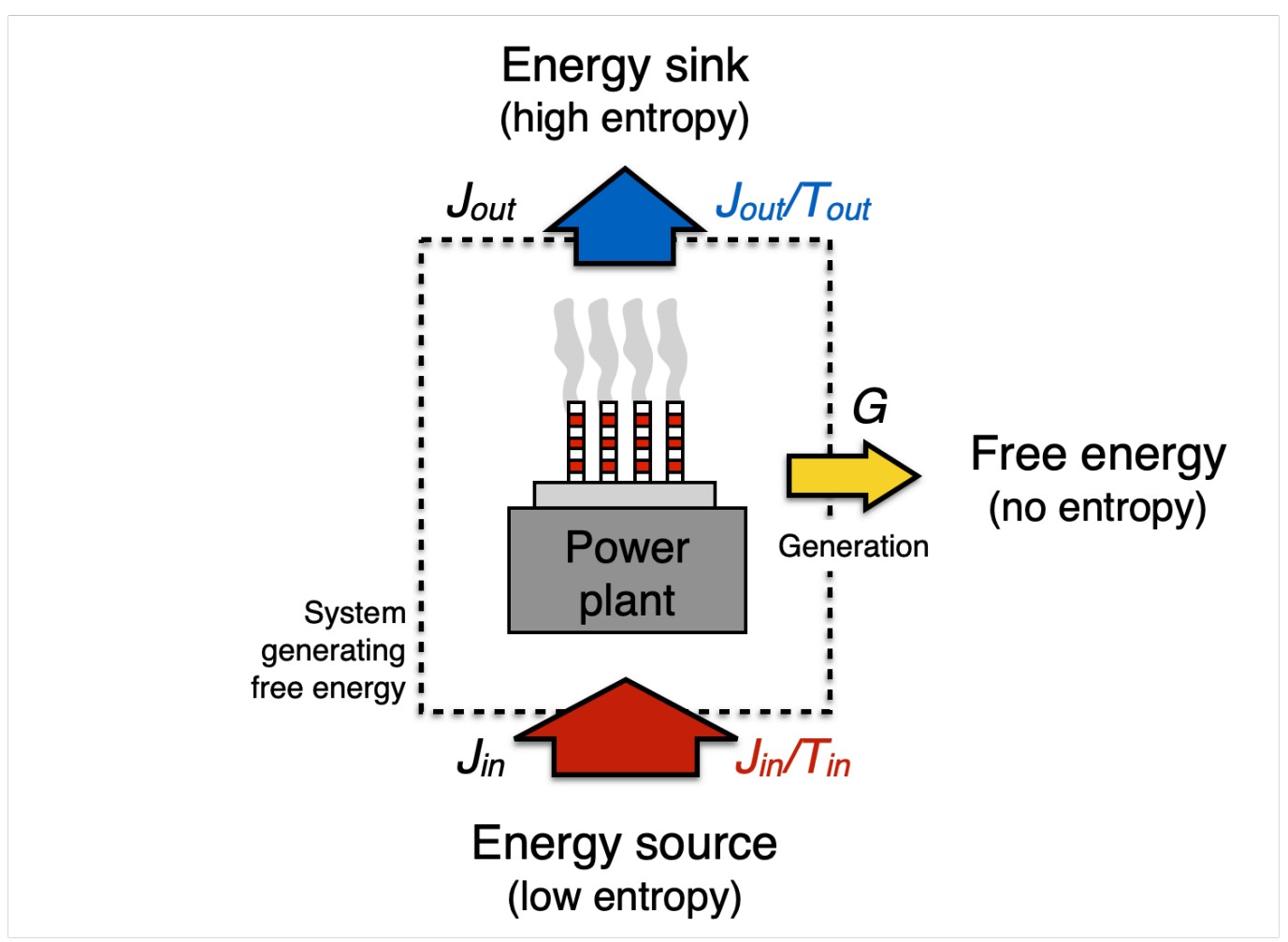

**Figure 5.** Schematic diagram to illustrate how the generation of free energy is constrained by the first and second law by setting the upper limit ("Carnot limit") on performing work, using a "power plant" as an example. The first law requires that in the process of generating free energy, $G$, the fluxes (symbols in black) in, $J_{in}$, and out, $J_{out}$, of the generating system are balanced, so that $J_{in} = J_{out} + G$. The second law requires that the entropy flux out of the system ($J_{out}/T_{out}$, in blue) is greater or equal to the entropy flux into the system ($J_{in}/T_{in}$, in red), so that $J_{out}/T_{out} \geq J_{in}/T_{in}$. At the Carnot limit, the entropy fluxes in and out of the system balance each other, yielding the greatest generation rate $G$.

The starting point is the first law of thermodynamics. We consider an energy conversion process, such as a conventional power plant (Figure 5). This power plant generates heat by combustion of the chemical free energy stored in fossil fuels at a rate $J_{in}$, it converts some of it into electricity at a rate $G$ (or motion, in case of the internal combustion engine), while another


part, the so-called waste heat flux, $J_{out}$, leaves the power plant, as can be seen by the plumes emerging from the cooling towers. The first law requires that in steady state with no changes in heat content, these energy fluxes are in balance, so that:

$$J_{in} = J_{out} + G \tag{2}$$

Note that $G$, the generation rate of electricity, represents the continuous work performed by the system, that is, its power.

The second ingredient shaping the upper limit on energy conversion comes from the entropy budget of the system and the requirement imposed by the second law. When heat is added to the system, it is added with a certain temperature, $T_{in}$, which increases the entropy of the system at a rate $J_{in}/T_{in}$. The waste heat flux removes heat at a different temperature, $T_{out}$, so it reduces the entropy of the system at a rate $J_{out}/T_{out}$. The heat fluxes thus accomplish the entropy exchange with the surroundings. We consider a steady state (as in Eq. 2), so we have no changes in heat storage within the system, and hence, no

change in entropy of the system in time. The difference between these two entropy exchange fluxes is balanced by the entropy production, $\sigma$, within the system, so that our entropy budget in steady state is represented by

$$\frac{J_{in}}{T_{in}} + \sigma = \frac{J_{out}}{T_{out}} \tag{3}$$

Note that the generation rate $G$ is not a part of the entropy budget because it generates free energy, that is, energy without entropy attached to it (as described in Section 2.2 above). To fulfill the requirement of the second law, the entropy production

in Eq. 3 can only be $\sigma \geq 0$.

     In the best case, $\sigma = 0$, so no entropy is being produced by this conversion process. Then, Eq. 3 simply yields $J_{out} = J_{in} \cdot T_{out}/T_{in}$, which can be combined with Eq. 2 to yield the limit of how much free energy can at best be generated:

$$G \leq J_{in} \cdot \frac{T_{in} - T_{out}}{T_{in}} \tag{4}$$

This expression is known as the Carnot limit. It states that only a fraction $(T_{in} - T_{out})/T_{in}$, known as the Carnot efficiency,

can at best be converted from the heat flux $J_{in}$ from the energy source into power $G$.

     This derivation of the Carnot limit is general, as it makes no specific assumptions about how this conversion process actually looks like. It includes merely the fluxes and conditions of the surroundings, and the two laws of thermodynamics. It is also general because instead of heat, one could use the same steps to derive an equivalent limit for the direct conversion of solar radiation into free energy (without heat being involved in the conversion), as it has been done to derive limits for photosynthesis

and photovoltaics (Press, 1976; Landsberg and Tonge, 1979, 1980).

     When we place such a conversion process into the Earth system, we also need to account for the fact that eventually, the generated free energy is dissipated back into heat. That is, in steady state where there is no change of free energy in time, the free energy budget is given by $G = D$, where $D$ is the dissipation of energy (i.e., conversion back into heat), see Eqn. 1. The dissipated heat is added back to the system. In principle, we can think of two extreme cases where this dissipation occurs and

the heat is added back to the system: (i) it is added at the warm side where heat enters the generation process, so it would contribute to $J_{in}$, or (ii) it is added at the cold side where it contributes to the waste heat flux $J_{out}$ that cools the system. As can easily be shown (see Appendix A), the second case yields the same limit as Eq. 4, while the first case yields a slightly different limit where the temperature in the denominator is replaced by $T_{out}$. This limit has been referred to as the limit of a so-called dissipative heat engine (Renno and Ingersoll, 1996; Bister and Emanuel, 1998). It yields slightly more power in applications to atmospheric science than Eq. 3 because $T_{out} < T_{in}$. The relevance of this added power was described for hurricanes in Emanuel (1999).

Note that there is also a thermodynamic limit of a heat engine that is referred to as the Curzon-Ahlborn limit (Curzon and Ahlborn, 1975). This limit includes a dissipative loss term for the heat transfer into and out of the heat engine, so that the limit yields lower power than the Carnot limit. In the atmosphere, the heating and cooling takes place mostly due to absorption 340 and emission of radiation (or condensational heating), so that such a dissipative loss term for the heat transfer into and out of the atmosphere does not apply. Hence, the Carnot limit is applicable for applications to radiatively-forced energy conversions within the Earth system.

Looking at free energy generation yields a more differentiated view than just looking at entropy production, as has been done in applications of the proposed MEP hypothesis. We can see this by evaluating the entropy budget of the whole system 345 that includes generation and dissipation of free energy. This system produces entropy by the dissipation, with its rate given by the steady-state condition $G = D$. When we use the Carnot limit for $G$ and assume that the dissipation takes place at the temperature $T_{out}$, we obtain for the entropy production $\sigma$:

$$\sigma = \frac{D}{T_{out}} = \frac{G}{T_{out}} = J_{in} \cdot \frac{T_{in} - T_{out}}{T_{in} T_{out}} = \frac{J_{in}}{T_{out}} - \frac{J_{in}}{T_{in}} \tag{5}$$

In other words, the entropy production of the system is entirely determined by the heat fluxes and the temperatures at the 350 boundaries of the system, but it does not depend on whether the system generates free energy or not.

This points out another deficit when looking at a system in a steady state only in terms of its entropy production: We cannot distinguish the case in which a system is so inefficient that it does not generate any free energy from the opposite case in which a system generates free energy at the Carnot limit and shows the strongest dynamics allowed for by the laws of thermodynamics. In the first case, all entropy production results from diffusion-like processes such as heat diffusion or radiative transfer, while in 355 the second case, all entropy production results from dynamics that involve the generation and dissipation of free energy. When we aim to understand a habitable planet like Earth, this distinction is critical – after all, the planetary role of life is fuelled by the free energy it generates and the work it does in chemically transforming its environment. This may then feed back to the planetary conditions to generating more free energy by life. Such potential feedbacks would be more concrete and testable than the general notion that life enhances planetary entropy production (see also discussion in Sect. 4.3 and Volk and Pauluis, 2010; 360 Frank et al., 2017).

## 2.4   Interacting boundary conditions and the maximum power limit

When we want to apply thermodynamic limits to the Earth system, we need to recognise that the boundary conditions are often not fixed, but react to the dynamics within the system that result from the generated free energy. This is a different situation compared to typical cases in classical thermodynamics where the boundary conditions of a system are fixed. In the Earth system, the only aspect that is truly fixed is the rate of incoming solar radiation at the top of the atmosphere. This lack of fixed boundary conditions is particularly relevant for those cases where free energy is generated and the resulting dynamics alters these boundary conditions.

This is the case for the surface-atmosphere system (Figure 6), where the absorption of solar radiation heats the surface, emission of radiation cools the atmosphere, resulting in a differential heating of the surface-atmosphere system. This differential heating is used to perform the work to generate convective motion, this motion transports heat, and this heat transport depletes the radiative heating difference. This results in a lower temperature difference, thereby affecting the boundary conditions. Mathematically, this is represented by a dependence of the second term (the efficiency term) on the heat flux in the Carnot limit (cf. Eq. 4). This interaction can easily be accounted for and leads to a maximum power limit. The maximum power limit has been recognised widely, for instance, in electrical engineering and in some relevant literature concerning the Earth system (Lotka, 1922a, b; Odum and Pinkerton, 1955), but typically not in thermodynamics.

In the following, I want to use convective motion in the surface-atmosphere system as an example to illustrate that these interactions can easily be accounted for by formulating the associated energy balances. These energy balances determine the temperature difference that drives the generation process of the heat engine, but are affected by the heat flux that is associated with the convective motion that is being generated. In other words, we consider atmospheric convection as being the result of an atmospheric heat engine that operates from the differential radiative heating, with solar radiation heating the surface, and thermal emission to space cooling the atmosphere (as described in Kleidon and Renner (2013a)).

To start, we consider two energy balances to describe the system (Figure 6): (i) the surface energy balance, where most of the solar radiation is absorbed and from which the surface temperature $T_s$ can be inferred from the emission of terrestrial radiation, and (ii) the energy balance of the whole system, which balances total absorption of solar radiation with total emission of terrestrial radiation to space, which sets the radiative temperature $T_r$.

The surface energy balance consists of the absorption of solar radiation, $R_s$, and downwelling terrestrial radiation, $R_{l,down}$, both of which heat the surface (i.e., a conversion of radiative into thermal energy), and the cooling by emission of radiation (i.e., a conversion of thermal into radiative energy) as well as a heat flux $J$ that results from the generation of vertical motion (the sensible and latent heat flux, combined here for simplicity):

$$R_s + R_{l,down} = \sigma T_s^4 + J \tag{6}$$

Here, we assume that the surface emits like a blackbody, with $\sigma$ being the Stefan-Boltzmann constant ($\sigma = 5.67 \cdot 10^{-8}$ W m$^{-2}$ K$^{-4}$). We further neglect changes in heat storage or net horizontal transport of heat, which can be relevant for ocean surfaces.

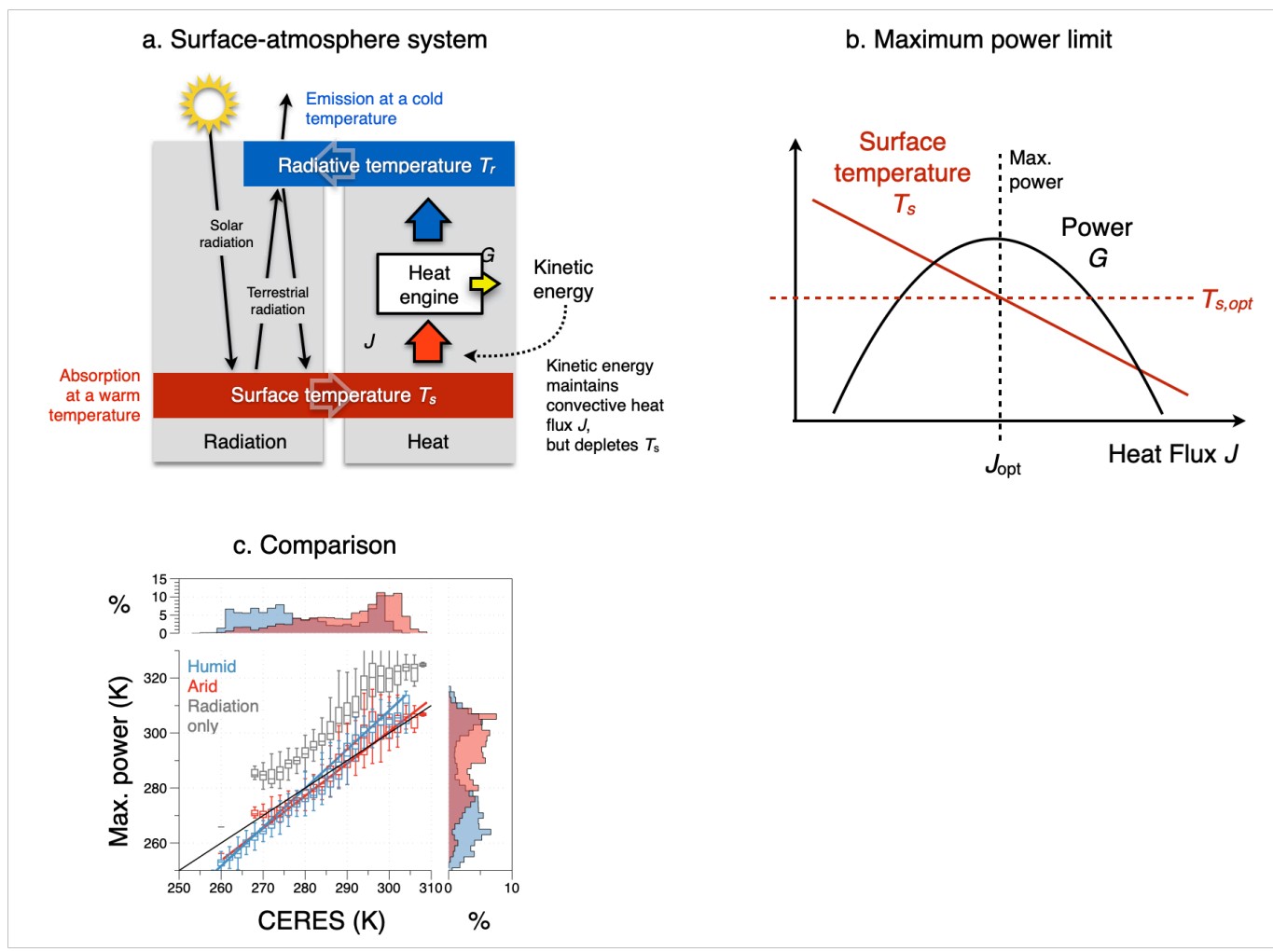

**Figure 6.** a. Schematic diagram of the energy balances of the surface-atmosphere system (red box: surface energy balance; blue box: atmospheric energy balance). These balances set the boundary conditions for the operation of an atmospheric heat engine that generates vertical convective motion. This engine generates kinetic energy associated with atmospheric convection, which transports heat from the surface into the atmosphere, thereby lowering the temperature difference between the surface and the atmosphere. b. The trade-off between a greater heat flux $J$ resulting in a colder surface temperature $T_s$ results in a maximum power limit for the heat engine. This limit is associated with an optimum sustained level of turbulent heat fluxes $J_{opt}$ for surface-atmosphere exchange and an optimum surface temperature $T_{s,opt}$, thus providing an additional, thermodynamic constraint to the dynamics of the system. c. Comparison of climatological mean land surface temperatures estimated from the energy balance (y-axis) without turbulent fluxes (grey), and for turbulent heat fluxes inferred from maximum power, separated for humid (blue) and arid (red) regions, to those inferred from the CERES satellite-derived radiation dataset (Loeb et al., 2018; Kato et al., 2018). After Kleidon and Renner (2013a) and Kleidon (2021b).

The energy balance of the whole system is set by the total absorbed solar radiation, $R_{s,tot}$, and the emission of terrestrial radiation to space:

$$R_{s,tot} = \sigma T_r^4 \tag{7}$$

where $T_r$ denotes the radiative temperature. The use of the radiative temperature here has a thermodynamic motivation: Since blackbody radiation is radiation at maximum entropy, the radiative temperature is the lowest temperature at which the absorbed solar radiation can be emitted to space, representing the highest radiative entropy export from the Earth system. It is thus the coldest temperature from which work can be derived in a climatological mean setting, and yields the upper bound on how much power can maximally be derived. Note that this temperature is not the temperature of a specific height within the atmosphere, but entirely focused on the most optimistic entropy export associated with the emission of outgoing longwave radiation.

With these energy balances, we have a formulation of the two temperatures, $T_s$ and $T_r$, and can evaluate the maximum power that can be derived from the Carnot limit (Eq. 4). Such an application of the heat engine to describe atmospheric convection is in itself not new (see, e.g., Renno and Ingersoll, 1996; Emanuel, 1999; Pauluis and Held, 2002a). The difference of what we do in the following is that we explicitly take the interaction of this heat engine with the heat input from the surface into account, which leads to the maximum in power, which in turn provides an additional constraint to close the energy balance. When we consider a greater $J$, then $T_s$ decreases according to the surface energy balance (Eq. 6), while $T_r$ remains unaffected. A greater $J$ thus results in a depleted temperature difference $T_s - T_r$ and a lower efficiency term in the Carnot limit. As a result, the power derived from the heat flux has a maximum (a maximum power limit), which is achieved with an intermediate, optimum value of the heat flux of

$$J_{opt} = \frac{1}{2} \cdot (R_s + R_{l,down} - R_{l,0}) \tag{8}$$

where $R_{l,0}$ is a constant term that originates when the Stefan-Boltzmann law is linearized in the form $R_{l,up}(T) = R_{l,0} + k_r(T - T_0)$ for simplicity (with $T_0$ being a fixed reference temperature, $R_{l,0} = \sigma T_0^4$ and $k_r = 4\sigma T_0^3$ is the linearisation constant). The surface temperature associated with this maximum in power is then given by

$$T_s = T_r + \frac{R_s + R_{l,down} - R_{l,0}}{2k_r} \tag{9}$$

In other words, the maximization of power constrains the dynamics such that the magnitude of the heat flux $J$ as well as the resulting surface temperature $T_s$ can be predicted from it. The outcome then only depends on the radiative forcing and certain assumptions pertaining to how radiative transfer is formulated (e.g., the height of convection, Dhara et al. (2016); or the longwave optical thickness, Conte et al. (2019)).

This maximum power limit results from this lack of fixed boundary conditions at the surface. Yet, the interaction that is caused by the response of the surface temperature to varying magnitudes of the heat flux $J$ is nevertheless constrained by the

response of the surface energy balance and can thus be accounted for. As we will see further below, this maximum power limit
plays a highly relevant role as it can explain observed surface energy balance partitioning, surface temperatures (Fig. 6c), and
associated evaporation rates very well. A more detailed evaluation in Ghausi et al. (2023) shows that this approach can yield
even better estimates for turbulent fluxes and surface temperatures in very close agreement with observations when the radiative
forcing is used in higher temporal resolution. What this implies is that this limit does not just exist, but that atmospheric motion
apparently evolves to and operates at this limit.

At first sight, the focus of the maximum power approach on the boundary conditions may seem at odds with common
atmospheric theory, which focuses on unstable conditions and adiabatic lapse rates within the atmosphere. Yet, when looking
at the land-atmosphere system as a whole and specifically at how stable and unstable conditions are formed, then these are
typically related to the absence or presence of solar radiative heating at the land surface. During the daytime, solar radiative
heating causes unstable conditions and fuels the convective heat engine, while at night, radiative cooling of the surface causes
stable conditions, with no heat engine being active. This difference in boundary conditions to operate the convective heat engine
has important implications – for instance, it has been used to explain the difference in climate sensitivities between day and
night and land and ocean (Kleidon and Renner, 2017). In other words, the presence or absence of stable conditions are closely
linked to the radiative boundary conditions, particularly at the surface. Given that solar radiation is primarily absorbed at the
Earth's surface, it generally causes unstable conditions and provides the fuel for the atmospheric heat engine.

Furthermore, it may seem surprising that the adiabatic lapse rate or a certain height of the atmosphere does not enter this
approach. This is because the Carnot limit does not include conditions within the heat engine, only the conditions at the
boundary are important for its derivation. The atmospheric boundary condition represents the cold sink of the heat engine,
which is formulated in terms of the radiative temperature of the atmosphere. That is, the boundary condition is formulated in
terms of an atmospheric temperature that yields the highest entropy export to space that is thermodynamically possible. This
does not represent a specific height within the atmosphere. The trade-off that leads to maximum power, however, does not
relate to the radiative temperature, but rather to the boundary condition at the surface and its temperature, where instability is
caused by the heating due to the absorption of solar radiation.

The simplicity of the maximum power approach also appears to ignore the vast complexity that is involved in turbulent
processes. This may be interpreted as follows: turbulence appears to be organised in such a complex way that the only limiting
factors relate to the thermodynamics of free energy generation. This is consistent with the derivation of the Carnot limit above,
which only needs the energy- and entropy fluxes at the system boundary, and not the details of the conversion process within
the heat engine. To substantiate this interpretation, however, would require further investigations. Nevertheless, it appears that
the simplicity of the maximum power approach does not appear to be in contradiction with common atmospheric concepts or
the complexity of turbulent motion that is involved. It rather emphasises how important the radiative boundary conditions are
for generating atmospheric motion.

As these estimates only contain energy balances and the assumption of convective motion operating at the maximum power
limit, it would seem that this limit can explain much of the emergent simplicity observed in climatological patterns. We will
look at this implication in greater detail further below.

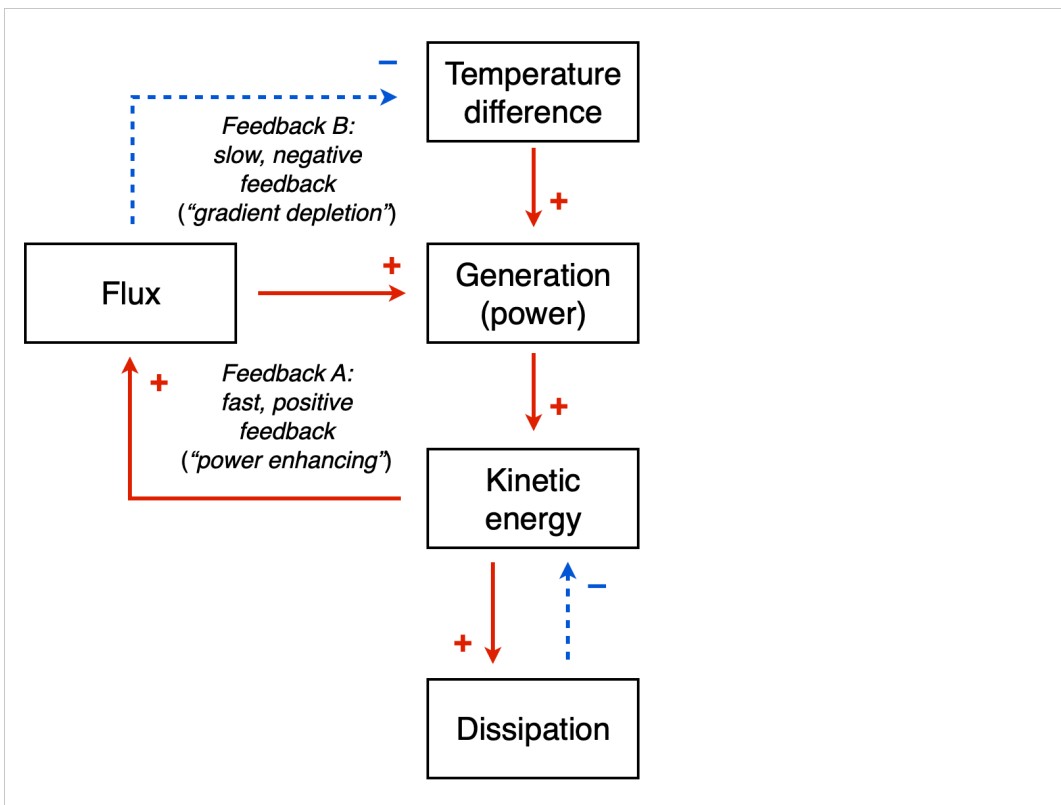

**Figure 7.** A schematic diagram to illustrate how a maximum in power can result from the outcome of two competing feedback processes. The strength of these feedback processes are, in turn, modulated by the intensity of turbulent friction. After Kleidon (2016).

## 2.5 Evolution to maximum power

The existence of a maximum power limit does not tell us if and how a system should evolve towards such a limit. While these aspects have not been resolved, we can nevertheless be somewhat more specific and attribute the outcome of maximum power to two dynamic feedbacks that balance each other at the state of maximum power (Fig. 7, see also Kleidon et al. (2013); Kleidon (2016)).

To do so, we need to first link the generated power explicitly to the dynamics of free energy within the system, using a budget

similar to the one expressed by Eq. 1. Power $G$ generates kinetic energy, $A_{ke}$, which in turn results in the heat flux, $J$, with more kinetic energy resulting in a greater heat flux (so $J = f(A_{ke})$). This greater heat flux then has two effects: (i) it generates more power, but (ii) it also depletes the temperature difference, which reduces the power. These two effects then result in two feedbacks: The first effect results in a fast, positive, "power enhancing" feedback, while the second effect results in a slow, negative "gradient depletion" feedback. The latter is slower as it involves changes in heat storage, that is, there is a thermal

inertia associated with the feedback. One can show that at maximum power, these two feedbacks have the same magnitude, and

since they have opposite signs, they oppose each other at the maximum, making the maximum power state a stable outcome of these feedbacks (Kleidon, 2016).

These dynamics and feedbacks are overall modulated by the intensity by which kinetic energy is dissipated by friction. This can, for instance, expressed as $D = A_{ke}/\tau$, with the intensity of friction being described by the time scale $\tau$. It would seem that attributes associated with turbulent friction and its inherent complexity are the ones that evolve and allow the system to reach the maximum power state. This notion is consistent to climate model simulations, in which a state of Maximum Entropy Production was obtained by changing empirical constants in the parameterisation of friction (a friction time scale in Kleidon et al. (2003) and Pascale et al. (2013) and the von-Karman constant in Kleidon et al. (2006)). Since MEP can be interpreted as the result of maximum power in a climatological mean state, these studies confirm this interpretation.

More generally, we may hypothesise that it is through the evolution and complexity of dissipative structures that result in the evolution to the thermodynamic limit that is set by the boundary conditions of the system. This would, however, require further research to be substantiated.

## 2.6 Putting things together

At the end of this section let us quickly sum up the main concepts of this section and what these imply for thermodynamics and optimality in the Earth system (Figure 2).

The three forms of entropy are at play when we follow the energy conversions within the Earth system. The energy source of solar radiation is energy in form of a flux of electromagnetic radiation with short wavelengths, representing low radiative entropy because solar radiation was emitted at a very high temperature. When it turns into heat upon absorption, it is converted into thermal energy with associated thermal entropy. When solar radiation is used by photosynthesis or photovoltaics, some of the energy is converted into non-thermal forms of energy and molar entropy, but not heat. After being converted by Earth system processes, energy is eventually emitted to space in form of terrestrial radiation at much longer wavelengths, representing much higher radiative entropy because it was emitted at a much colder temperature than the Sun. So, clearly, this distinction of the three different forms of entropy plays a central role in Earth system processes, and it is important to distinguish these as well as the associated forms of energy.

Free energy is generated from solar radiation mainly by three different processes, as shown by the yellow boxes in Figures 1 and 2. The free energy contained in atmospheric motion in form of kinetic energy is generated by heat engines that operate equivalently to the example of the power plant used in the derivation of the Carnot limit (Eq. 4). Photosynthesis generates chemical free energy contained in carbohydrates and oxygen, while photovoltaics produces electric free energy (although currently at a much lower magnitude than photosynthesis). This free energy is associated with dynamics - like atmospheric motion or metabolic activities of living organisms - and it can be converted further into other forms of free energy, e.g., associated with hydrologic (or other geochemical) cycling, ocean dynamics, trophic networks in ecosystems, or food production in human societies. Thus, by evaluating the magnitudes of these free energy generation processes at the beginning of energy conversion chains starting with the conversion of solar radiation, we gain a general understanding of how active the dynamics of the Earth system are in thermodynamic terms.

Thermodynamic limits and optimality then apply first and foremost to these three processes as these convert solar radiation into free energy, constraining the resulting dissipative dynamics by the energy input into the energy conversion sequences. This then leaves the question whether Earth system processes are constrained by thermodynamic limits, whether this is then associated with optimal functioning that maximizes the free energy generation for Earth system dynamics, and how much of these limits is then reflected in the emergent climatological patterns that we can observe.

## 3   Thermodynamic Optimality and Maximization: How does it apply to Earth system processes?

     In the following, I use three examples to illustrate the relevance of thermodynamic limits and associated optimality to Earth system processes and its success in reproducing observed patterns very well. In these examples, only the application to poleward heat transport by the atmospheric circulation represents a direct application of the maximum power limit, while the other two examples of evaporation and terrestrial carbon uptake are interpreted as indirect manifestations of the maximum power limit

associated with motion in the vertical direction (as described in Section 2.4). With this I want to provide a more differentiated view of how thermodynamic optimality applies to the Earth system that on the one hand shows its relevance in constraining motion and explaining climatological patterns, but that it is not a simple, general approach that can be applied to every process. Rather, it requires the specific Earth system context in which this process takes place, particularly concerning the connections and indirect consequences of maximum power associated with motion.

I will specifically use models that are as simple as possible to describe these examples. The motivation for this simplicity in formulation is to provide transparency and accessibility to the reader, and to not obscure outcomes with complex mathematical formulations. This then contributes to the need for a better, and simpler understanding of the Earth system (Held, 2005). Even at this simplest level, the outcomes will show the great importance of thermodynamic constraints because the derived estimates compare very well with observations. One can easily make these models more complicated, include more phenom-

ena and parameterisations, or add greater spatial and temporal resolution, and achieve a better fit with observations. Yet, the thermodynamic constraint will still play out in more or less the same, although in a less transparent way.

     With this reinterpretation of each of these examples, I will then discuss the linkages to previous interpretations of thermodynamic optimality, specifically the proposed MEP hypothesis, and to established theoretical concepts.

### 3.1   Poleward heat transport and the large-scale atmospheric circulation

At its very core, the large-scale atmospheric circulation involves energy conversions as work needs to be done to maintain motion against the friction that occurs at the surface and within the boundary layer. This work is derived from the difference in radiative heating between the warmer, equatorial regions and the colder, polar regions. The large-scale, radiative forcing of the Earth is shown in Fig. 8 in form of the observed radiative fluxes at the top of the atmosphere and the associated surface temperatures. Figure 8 illustrates two important aspects. First, the imbalance between absorbed solar radiation and emitted

terrestrial radiation demonstrates the importance of poleward heat transport, and therefore motion. Heat transport alters the radiative exchange of the planet and maintains this imbalance. Second, when we infer surface temperatures for the case of

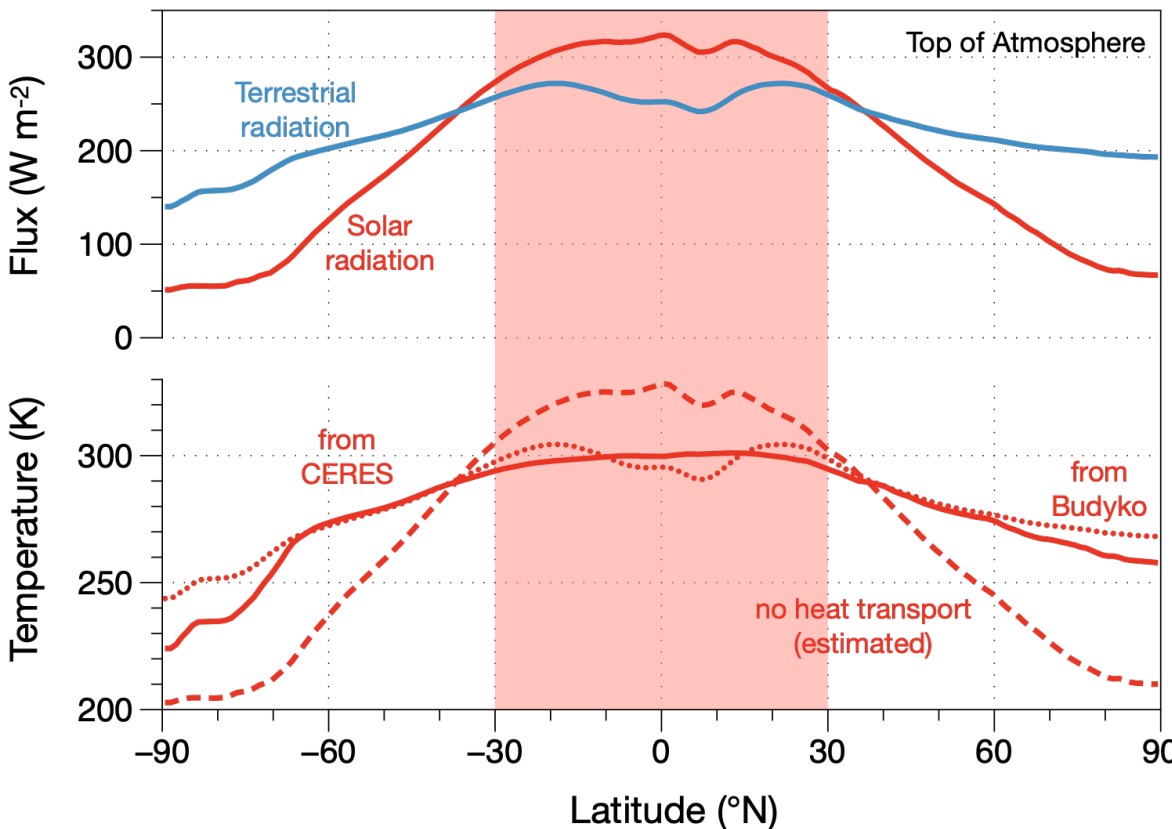

**Figure 8.** Planetary radiative setting in terms of (top) climatological zonal means of net fluxes of solar (red) and terrestrial (blue) radiation at the top of the atmosphere, taken from the CERES satellite-derived radiation dataset (Loeb et al., 2018; Kato et al., 2018). Bottom: Associated surface temperatures and how they react to the redistribution of heat: inferred from mean surface emission from the CERES dataset (solid line, "from CERES"), inferred from Budyko's (1969) empirical parameterisation using the TOA flux of terrestrial radiation (dotted line, "from Budyko", used for estimates in the text), and estimated for the case of no heat transport (dashed line, "no heat transport (estimated)"). The red-shaded area marks the area that represents one half of the surface area of the Earth and represents the tropics in Fig. 9. From Kleidon (2021a).

no heat transport in which emission to space is equal to the absorption of solar radiation, we see that the tropics would be notably warmer, while the poles would be notably colder (dashed line in the lower panel of Fig. 8, estimated using the empirical parameterization of Budyko (1969)). This demonstrates that the thermodynamic forcing of the climate system depends on what
the atmosphere does: Poleward heat transport acts to level out the temperature differences caused by uneven solar heating, and this is reflected in the more even emission to space at the top of the atmosphere, as shown by the blue line in the top panel of Fig. 8. Heat transport thus depletes the driving temperature difference, just as in the example of maximum power for the vertical direction (Section 2.4).

The atmosphere generates its free energy in form of kinetic energy associated with large-scale motion from this imbalance in

solar radiative heating. This motion transports heat, thereby affecting the thermodynamic forcing, and the combination results in a maximum power limit, similar to the example in Section 2.4, but in the horizontal direction. We thus have a thermodynamic process which performs work by converting a heating difference into free energy, generates motion, which is dissipated back into heat due to friction and produces thermal entropy.

Thermodynamic optimality has been applied to atmospheric motion in the past, most prominently in form of the proposed
MEP hypothesis. Since maximising power also maximises dissipation in steady state (and if no free energy is transferred into another system, e.g., oceans), and dissipation produces thermal entropy, both maximisations result in roughly the same outcome. Power can also be related to the rate by which available potential energy is converted into kinetic energy, thus linking this thermodynamic approach to the more common theoretical concept of the Lorenz Energy Cycle (LEC, described by Edward Lorenz in 1955; 1960 and 1967). These linkages are described in more detail further below after we first describe a minimum
model demonstrating the maximum power limit for the large-scale atmospheric circulation.

The maximum power limit for the large-scale circulation is demonstrated with a two-box model, in which we split up the Earth into two regions of same size at the $30°$ latitude to characterise the differential radiative heating from which the kinetic energy is being generated (as shown in Figure 9, as in Kleidon (2021a)). Due to the Earth's geometry, the tropical regions absorb more of the incoming solar radiation per unit surface area than the extratropical regions. This uneven heating results in
a difference in surface temperatures between the tropics and the extratropics. As a result, the absorption of solar radiation in the tropics adds more heat at lower entropy because it is absorbed at a higher temperature compared to the extratropics. The differential solar heating across latitudes thus sets the stage for an atmospheric heat engine, with the greater absorption of solar radiation in the tropics than the extratropics as the heat source, and the greater emission into space than the absorbed solar radiation in the extratropics as the heat sink. This heat engine performs the work to generate large-scale atmospheric motion
and the associated kinetic energy, and is constrained by the Carnot limit (Eq. 4).

We obtain the maximum power limit for this heat engine in an equivalent way as in Section 2.4. We use the poleward heat transport as the heat flux and the temperature difference between the two boxes for the efficiency term. To quantify this maximum, we need a formulation of the energy balances to express how the temperature difference in the Carnot limit is depleted by the poleward heat transport. These temperatures can be inferred from the respective energy balances of the two
boxes, because the thermal emission to space at the top of the atmosphere, $R_l$, depends on surface temperature. We write the two energy balances as

$$R_{s,t} = R_{l,t} + J \qquad R_{s,p} + J = R_{l,p} \qquad (10)$$

where I used the index $t$ for the tropical box (30°S - 30°N), the index $p$ for the extratropical box (latitudes greater than $30°$), $R_s$ for the total absorbed solar radiation, and $J$ for the poleward heat transport.

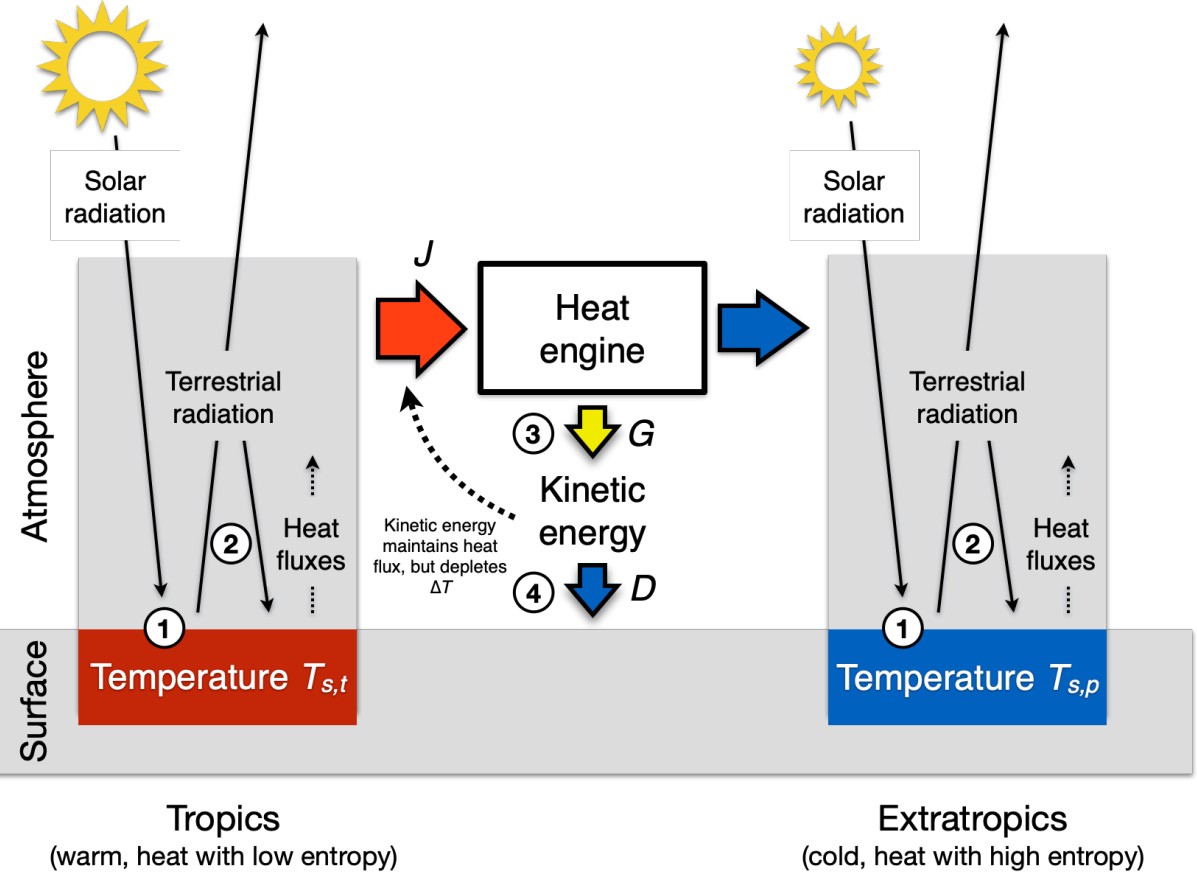

**Figure 9.** Schematic diagram of how kinetic energy is generated within the atmosphere by a heat engine that uses the difference in solar radiative heating between the tropics (left box, red) and the extratropics (right box, blue) as a driver. The generated motion maintains the poleward heat transport $J$, but depletes the temperature difference $T_{s,t} - T_{s,p}$ between the two boxes. This results in a maximum power limit for the strength of the atmospheric circulation. Also marked by the numbered circles are places at which thermodynamic conversions and entropy production take place: (1) conversion of solar radiation into thermal energy by absorption at the surface temperature, producing thermal entropy; (2) absorption of terrestrial radiation and re-emission as well as vertical, convective motions, which both produce thermal entropy; (3) generation of free energy from heat flux $J$; (4) frictional dissipation, that is, conversion of kinetic energy back into thermal energy, which produces thermal entropy.

We then infer surface temperatures from $R_l$ by using the empirical parameterization of Budyko (1969):

$$R_l = a + b \cdot T_s \tag{11}$$

where $a = -388.7 \, \text{W m}^{-2}$ and $b = 2.17 \, \text{W m}^{-2} \, \text{K}^{-1}$ are empirical constants, with $T_s$ is in units of K. This linear relationship between surface temperature and the radiative flux to space compares very well to observations and can be explained by the role of water vapor in the atmosphere (Koll and Cronin, 2018).

When Eqns. 10 and 11 are combined, surface temperatures are functions of absorbed solar radiation and the poleward heat flux $J$:

$$T_{s,t} = \frac{R_{s,t} - a - J}{b} \qquad T_{s,p} = \frac{R_{s,p} - a + J}{b} \tag{12}$$

Note how heat transport depletes the temperature difference due to the opposing signs in these two expressions. For a value of $J = 0$ in the absence of heat transport, temperatures are determined by the local solar radiative forcing only ($R_l = R_s$). This yields the largest temperature difference (Figure 10e), which would then yield the greatest efficiency in the Carnot limit, but no power because $J = 0$. The temperature difference vanishes with a value of $J_{max} = (R_{s,t} - R_{s,p})/2$, yielding a globally uniform surface temperature of $T_s = 290$ K and the efficiency in the Carnot limit vanishes to zero, thus also yielding no power.

There is hence a maximum in the power at an intermediate value of $J$. It is obtained mathematically from the Carnot limit (Eq. 4), the temperature expressions (Eq. 12), an additional factor of 1/2 to account for the fact that heat is transported from one half of the Earth to the other, and then setting $dG/dJ = 0$ (see also Figure 10c). The slight dependence of the power on $T_{s,t}$ in the denominator has been neglected here. This maximum yields a power of $G_{max} = (R_{s,t} - R_{s,p})^2/(16bT_{s,t})$, is associated with an optimum heat flux of $J_{opt} = (R_{s,t} - R_{s,p})/4$, which reduces the temperature difference $T_{s,t} - T_{s,p}$ to half of its maximum value in the absence of heat transport.

We next use the observed radiative forcing from the CERES radiation dataset (Loeb et al., 2018; Kato et al., 2018) to evaluate these expressions. The only information needed is the absorption of solar radiation for the two boxes. The observed flux of terrestrial radiation serves as a means to test the outcome associated with the maximum power estimate. With climatological means of $R_{s,t} = 306 \, \text{W m}^{-2}$ and $R_{s,p} = 177 \, \text{W m}^{-2}$, this yields a maximum in power to generate kinetic energy of 1.6 W m$^{-2}$ (Figure 10c), which agrees quite well to the magnitude of kinetic energy generation of about 2.1 - 2.5 W m$^{-2}$ inferred from the reanalyses (Li et al. 2007). The estimated value for the poleward heat transport of 32 W m$^{-2}$ is a bit less than the 43 W m$^{-2}$ diagnosed from the CERES dataset, resulting in a somewhat greater temperature difference of $T_{s,t} - T_{s,p} \approx 30$ K compared to a value of 21 K derived from CERES. In other words, the maximum power limit predicts the magnitude of the free energy dynamics of the global atmosphere rather well.

There are, obviously, some discrepancies. The optimum heat flux is less than the one diagnosed by CERES, and hence the temperature difference is somewhat overestimated. These discrepancies can be attributed to the omission of the effects of seasonal variations as well as the contribution by the Hadley cell to the mean circulation (see Kleidon, 2021a, for a more detailed discussion). The limit also does not account for the role of rotation rate, which may act as an additional constraint to motion, e.g., on planets with a faster rotation rate (Pascale et al., 2013). This could then result in reduced power and dissipative dynamics, and the maximum set by the thermodynamic boundary conditions would not be reached. Yet, for the Earth's atmosphere the maximum power limit provides a consistent estimate of magnitudes that is not coincidental. It simultaneously estimates power,

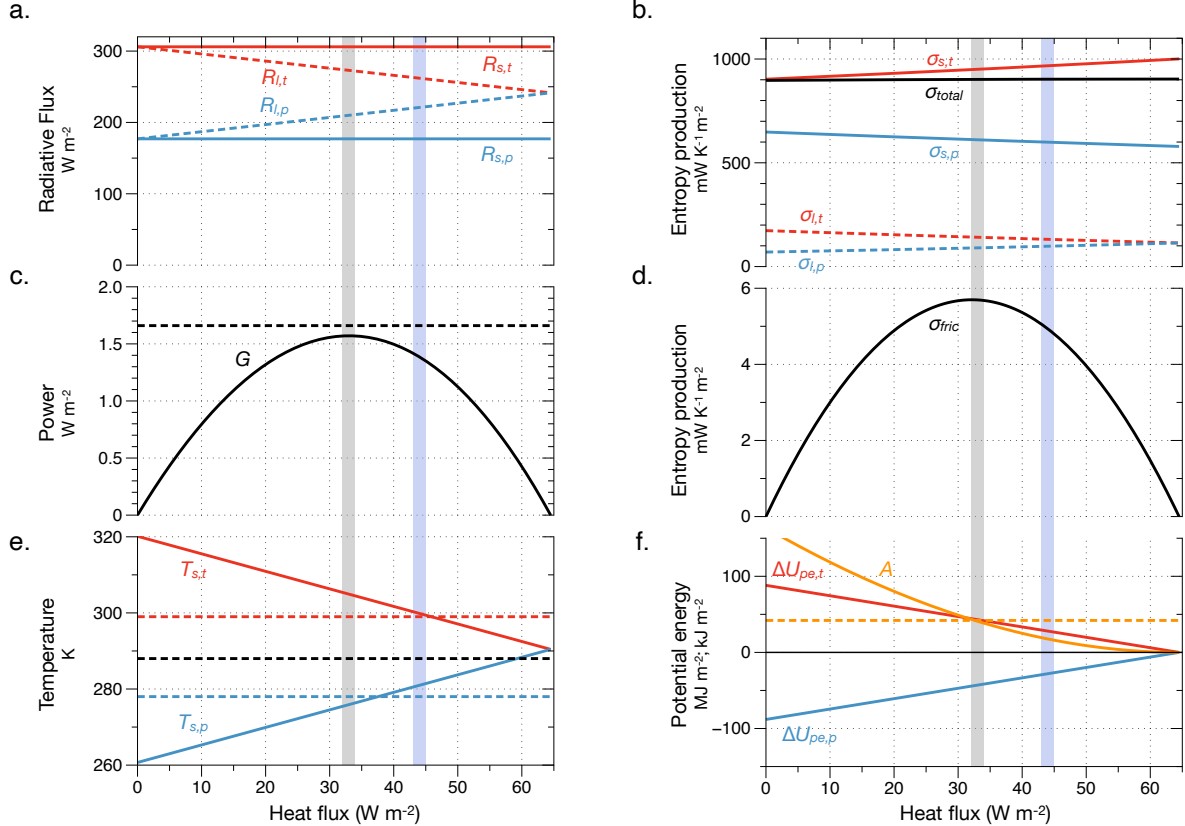

**Figure 10.** Radiative forcing and energetic properties estimated from the two-box model shown in Figure 9 relating to the strength of the atmospheric circulation and maximum power. a. Radiative forcing in terms of absorbed solar radiation ($R_{s,t}$ and $R_{s,p}$) of the tropical (red lines) and extratropical box (blue lines) and emitted terrestrial radiation to space ($R_{l,t}$ and $R_{l,p}$). b. Entropy production by absorption of solar radiation (i.e., thermalisation, $\sigma_{s,t}$ and $\sigma_{s,p}$, circles marked as (1) in Fig. 9), radiative transfer of terrestrial radiation ($\sigma_{l,t}$ and $\sigma_{l,p}$, (2) in Fig. 9), and total entropy production, $\sigma_{total}$ (black line). c. Power $G$ to generate atmospheric motion ((3) in Fig. 9). The horizontal dashed line marks the power inferred directly from the CERES forcing. d. Entropy production by frictional dissipation of kinetic energy ((4) in Fig. 9). e. Surface temperatures of the tropical (red line) and extratropical (blue line) boxes with the surface temperatures estimated from the CERES TOA flux marked by the dashed lines (black line: global average). f. Difference in potential energy compared to an isothermal Earth ($\Delta U_{pe,t}$, $\Delta U_{pe,p}$, in MJ m$^{-2}$, red and blue lines) and the available potential energy ($A$, in kJ m$^{-2}$, orange line, dashed orange line represents published estimates). The vertical shaded areas represent the value of heat transport from maximum power (grey) and inferred from the CERES data (blue). After Kleidon (2021a).

temperature difference, and heat transport, all of which roughly agree with observations. It demonstrates how interlinked these variables are, and how this sets an upper bound on the power and the generation of kinetic energy that can be generated from the solar forcing.

**Table 1.** Thermal entropy production (in mK K$^{-1}$ m$^{-2}$) by different climate system processes estimated by the two box model as described in App. B for the cases of no heat transport, at maximum power, and for an isothermal state.

| Process | No heat transport | Max. power | Isothermal |
|---|---|---|---|
| Absorption of solar radiation (thermalisation) | | | |
| Tropics | 902.7 | 950.2 | 1000.5 |
| Polar | 648.2 | 610.9 | 578.7 |
| Global mean | 775.5 | 780.6 | 789.6 |
| | | | |
| Radiative transfer of terrestrial radiation and convection | | | |
| Tropics | 173.1 | 141.2 | 113.8 |
| Polar | 69.9 | 90.0 | 113.8 |
| Global mean | 121.5 | 115.6 | 113.8 |
| | | | |
| Frictional dissipation | | | |
| Global mean | 0.0 | 5.7 | 0.0 |
| | | | |
| Total mean | 897.0 | 901.9 | 903.4 |

The broad agreement with observations in combination with the support from previously conducted, much more detailed climate model simulations on MEP (e.g. Kleidon et al., 2003, 2006) suggests that the atmosphere works as hard as it can to generate motion. The justification is analogous to the conclusion of Section 2.4: it is the thermodynamics of the boundary conditions in combination with the interaction of the driving temperature difference that sets the magnitude of power that drives the large-scale dynamics without the knowledge of the details of how the atmospheric heat engine is actually implemented. This then yields an explanation for the emergent simplicity in the associated climatological patterns and the robustness of the magnitude of poleward heat transport.

This outcome of maximum power can be linked to the proposed MEP hypothesis and the more established LEC framework of atmospheric energetics.

The link to the proposed MEP hypothesis is established by the thermal entropy production due to frictional dissipation, that is, when the free, kinetic energy is converted back into heat. In the climatological mean state, power balances dissipation, so that maximum power, that is, maximum generation of kinetic energy, equals maximum frictional dissipation, which then is almost the same as maximised thermal entropy production due to frictional dissipation (Figure 10d). This entropy production term is what has been maximised in previous applications of MEP to atmospheric heat transport with similar box models (e.g., Paltridge, 1975, 1979; Lorenz et al., 2001; Kleidon, 2004; Pascale et al., 2012). In other words, maximum power and MEP almost yield the same outcomes. Yet, we should recognise that the entropy production by frictional dissipation is only a tiny

contribution to the overall entropy production of the planet (Table 1, Figure 10b,d, see Appendix B for derivations), because the thermalisation associated with the absorption of radiation produces typically much more entropy than frictional dissipation. Focusing on maximum power rather than on MEP gives us a more specific view on which aspect of the system is maximised and why. After all, the atmospheric circulation needs work to be maintained. Just focusing on entropy production, however, cannot distinguish between processes that involve work from those which do not, a point already made at the end of Section 2.3.

The link to the LEC is done by making the link between heating differences and kinetic energy generation more detailed by including the intermediate step of generating available potential energy (APE). Diabatic heating sources generate potential energy differences, from which only a fraction becomes APE, a concept originally introduced by Margules (1905). APE is then converted into kinetic energy (KE) at a certain rate. It is typically diagnosed from pressure and temperature fields of the atmosphere. The application of maximum power to atmospheric dynamics differs in that it takes the diabatic heating terms directly as the starting point, without the need for the intermediate step of APE generation (Kleidon, 2021a). This can nevertheless be inferred from the conditions associated with the maximum power limit and compared to observations (Figure 10f, see Appendix C for derivations). The value of APE of $41 \cdot 10^5$ J m$^{-2}$ associated with the maximum power limit matches the range of $40 - 50 \cdot 10^5$ J m$^{-2}$ estimated from observations (Peixoto and Oort, 1992; Li et al., 2007) very well. Further conversions between APE and KE associated with the mean flow and eddies is generally considered as reversible within the LEC framework (which is inherent in the notion of geostrophic, frictionless flow as well as the notion of non-dissipative waves), so that these conversions neither generate nor dissipate free energy. Yet, we should also note that APE mostly reflects the state of disequilibrium inherent with the temperature difference $\Delta T$ (cf. Eq. C2), and reflects a form of energy that can freely be converted back and forth into kinetic energy. It therefore does not provide information about how dissipative atmospheric motion is. The focus on power and its maximisation, on the other hand, sets a firm limit to the magnitude of its dissipative behaviour. It thereby sets a relevant constraint for the maximum intensity of the Lorenz Energy Cycle and it appears that the atmospheric circulation operates at this maximum.

To sum up, the large-scale atmospheric circulation provides an example for an Earth system process that maximises power from the large-scale differences in solar radiative heating. Even when applied to a simplest two-box model, it provides an estimate for the magnitude of poleward heat flux that broadly agrees with estimates derived from observations. While the outcome is similar to a maximisation of material entropy production (Lucarini et al., 2014), the focus on power provides us additionally with an estimate of the strength of the dynamics - in terms of kinetic energy generation - and it allows us to link the thermodynamic view to the Lorenz Energy Cycle and its intensity (see also Lucarini et al. (2014)). Hence, this case of maximum power does not contradict established theory, it rather adds a relevant constraint that is currently not acknowledged.

## 3.2 Evaporation and hydrologic cycling

At first sight, hydrologic cycling appears to primarily involve mass fluxes of water. Evaporation, the conversion of liquid water to vapour, takes place mostly at the surface, adding moisture to the atmosphere, while the opposite conversion takes place within the atmosphere, causes precipitation and removes moisture. In the climatological mean, both fluxes balance each other

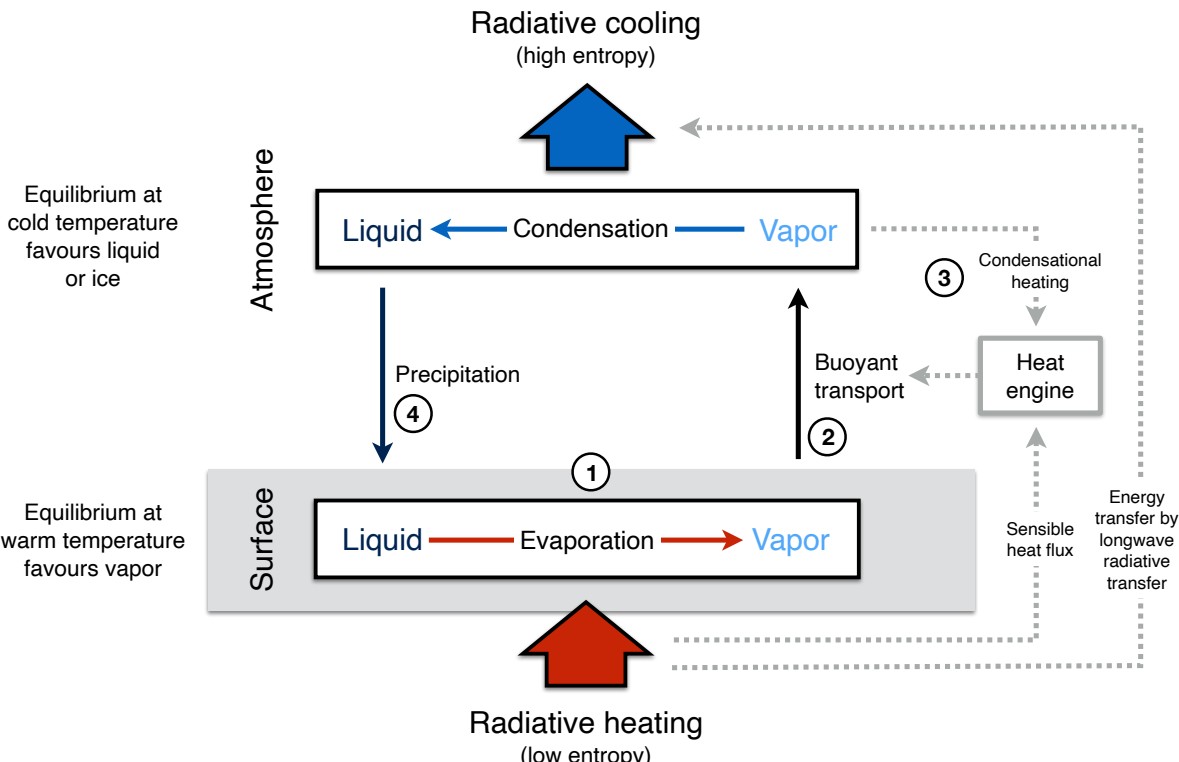

**Figure 11.** Schematic diagram to illustrate the thermodynamics of hydrologic cycling, the associated disequilibrium, its dissipative processes, and its connections to the Earth system. The numbered circles mark places at which thermodynamic conversions and entropy production takes place: (1) Absorption of solar radiation heats the surface, and thereby shifts the thermodynamic equilibrium of a water surface towards higher saturation vapor, resulting in evaporation; (2) The saturated water vapor at the surface mixes with the unsaturated near-surface air, producing entropy; (3) Moist air is lifted until it condenses. The condensed heat drives a moist heat engine that generates kinetic energy for moist convection; (4) falling raindrops dissipate their potential energy that was generated by lifting moistened air, resulting in frictional dissipation and thermal entropy production. The link to the maximum power limit of vertical convection (Sect. 2.4) is made by noting that it requires the work of buoyancy to lift moisture up to the condensation level.

at the global scale. To understand how thermodynamics and maximization constrain the magnitude of hydrologic cycling, we can thus either take an atmospheric perspective and look at precipitation, or a surface perspective and look at the factors that

constrain evaporation.

To understand how thermodynamics applies to hydrologic cycling and constrains its intensity, we need to first describe hydrologic cycling as a thermodynamic process which operates in a state of thermodynamic disequilibrium (Figure 11). Evaporation proceeds at warmer temperatures (circle (1) in Fig. 11), takes up latent heat, and changes thermal into molar entropy, while condensation occurs at colder temperatures, releasing this latent heat, converting molar into thermal entropy. Hydrologic

cycling thus effectively transports heat from the warmer surface into the colder atmosphere, thereby producing thermal entropy.

This entropy production by the hydrologic cycle is caused by diffusive and mixing processes as well as other processes that involve work, free energy, and dissipation, with the magnitude set by $LE \cdot (1/T_s - 1/T_r)$. This contribution is included in the convection part in the entropy budget shown in Table 1. Cloud droplets have potential energy that is generated by the work of lifting water, unsaturated air reflects the work done by dehumidification, and evaporation of seawater involves desalination work and the generation of chemical free energy. The potential energy of cloud droplets is dissipated as raindrops fall (circle (4) in Fig. 11), by gravitational drainage when the precipitated water reaches the land surface, or is converted into the kinetic energy of river flow. The chemical free energy in freshwater drives continental weathering processes that dissipate the disequilibrium between the desalinated rainwater and the solid minerals of the continental crust. So clearly, thermodynamics and free energy are at the very core of hydrologic cycling, and it would appear that these can act to restrict its magnitude.

In the following, I show that thermodynamic optimality and maximum power do not apply directly to hydrologic cycling, and not to the phase transitions, but rather indirectly by constraining surface energy balance partitioning and moisture transport. While the phase transitions of water are associated with changes in molar entropy, these do not contribute to entropy production as they proceed practically at saturation and are therefore reversible. To understand how thermodynamics nevertheless constrains hydrologic cycling and sets its magnitude, I focus on evaporation, because it is less variable and easier to describe than precipitation. By demonstrating this indirect link of maximum power with evaporation, I show that this thermodynamic limit is nevertheless relevant in shaping the magnitude of hydrologic cycling. This indirect effect relates to the maximum power of generating buoyant exchange and vertical transport from radiative heating of the surface, as described in Section 2.4.

When solar radiation heats a surface at which water is sufficiently available, it adds sensible and latent heat to the near-surface air, making it warmer and moister. The upper, thermodynamic limit on moistening is set by maintaining saturated air near the surface, as this represents the thermodynamic equilibrium state. The associated equilibrium partitioning is well known in micrometeorology. It was developed by Schmidt (1915), Penman (1948), and Slayter and McIlroy (1961), and is reflected in approaches to estimate potential evaporation, such as the well-known Priestley-Taylor estimate (Priestley and Taylor, 1972). This implies that thermodynamics and maximum power primarily constrain the magnitude of buoyant transport from the surface to the atmosphere, which combined with the thermodynamic equilibrium at the surface constrains the magnitude of evaporation and thus of hydrologic cycling. In hydrologic terminology, this means that thermodynamics and maximum power set the magnitude of potential evaporation, that is, the rate of evaporation from the land surface in the absence of water limitation.

To make this constraint more explicit, we first apply this equilibrium partitioning to the optimum heat flux, $J_{opt}$, derived from maximum power (Eq. 8). Over a short time interval $dt$, the near-surface air is simultaneously warmed by $dT$ and moistened by an increase in saturation vapor pressure, $de_{sat} = s \cdot dT$, with $s = de_{sat}/dT$ given by the Clausius-Clapeyron equation (this reflects the upper limit of maintaining saturated air). If we consider a small volume of this air, $V$, this addition results in contributions of warming, $dU_H$, of

$$dU_H = c_p \rho V \cdot dT \tag{13}$$

and moistening, $dU_{LE}$, of

$$dU_{LE} = L \cdot \frac{dq_{sat}}{dT} \cdot \rho V \cdot dT = L \cdot \frac{R_a}{R_v} \cdot \frac{1}{p} \cdot s \cdot \rho V \cdot dT \tag{14}$$

with $c_p$ being the heat capacity of air ($c_p = 1005$ J kg$^{-1}$ K$^{-1}$), $\rho$ being air density ($\rho \approx 1.1$ kg m$^{-3}$ for typical surface conditions), $L$ being the latent heat of vaporization ($L \approx 2.5 \cdot 10^6$ J kg$^{-1}$), $q_{sat}$ being the saturation specific humidity (with $q_{sat} = R_a/R_v \cdot e_{sat}/p$), $R_v$ and $R_a$ the ideal gas constants of water vapor and air (with $R_a/R_v \approx 0.622$), $p$ the surface pressure ($p = 1013$ hPa for average surface conditions).

Since $dU = dU_H + dU_{LE}$, we can then derive the relative partitioning among heating and moistening at thermodynamic equilibrium, noting that the terms $\rho V \cdot dT$ cancel out while collecting the other constants in the so-called psychrometric constant $\gamma = c_p p (R_v/R_a)/L \approx 65$ Pa K$^{-1}$. The relative partitioning is then given by the ratios

$$\frac{dU_H}{dU} = \frac{\gamma}{s+\gamma} \qquad \frac{dU_{LE}}{dU} = \frac{s}{s+\gamma} \tag{15}$$

When combined with the optimum heat flux derived from maximum power, we obtain the following partitioning into sensible and latent heat

$$H_{opt} = \frac{\gamma}{s+\gamma} \cdot J_{opt} \qquad LE_{opt} = \frac{s}{s+\gamma} \cdot J_{opt} \tag{16}$$

This partitioning represents two constraints of thermodynamics on evaporation: the maximum power limit sets the magnitude of buoyant exchange, and the thermodynamic equilibrium conditions at the surface set the magnitude of atmospheric moistening by evaporation.

This thermodynamic estimate of evaporation has previously been evaluated (Kleidon and Renner, 2013b; Kleidon et al., 2014; Conte et al., 2019; Kleidon, 2021b; Ghausi et al., 2023) and shown that it can reproduce observations rather well. The evaluation of Kleidon (2021b) used annual means of global radiation datasets (CERES, Loeb et al., 2018; Kato et al., 2018) to estimate potential evaporation, and combined this with a precipitation dataset (GPCP, Adler et al., 2016) to account for the role of water limitation at the land surface to estimate actual evaporation. This estimate for the years 2003 - 2018 is summarized in Fig. 12 and compared to the GLEAM dataset (Miralles et al., 2011; Martens et al., 2017), separately for humid and arid regions. While the agreement for arid regions is no surprise as it reflects the water limitation imposed by the precipitation dataset, the estimate also agrees very well in humid regions where water is abundantly available. This supports the interpretation made here that the magnitude of land surface evaporation, and thus of hydrologic cyling, is indirectly constrained by maximum power through the generated buoyant transport (as indicated by circle (2) in Fig. 11) combined with the thermodynamic equilibrium partitioning of turbulent fluxes into sensible and latent heat.

The indirect application of thermodynamics to evaporation is consistent with boundary layer theory. Basically, the maximisation of power applies to the sensible heat flux, and this links directly to the buoyancy production term in the Turbulent

## a. Evaporation estimate

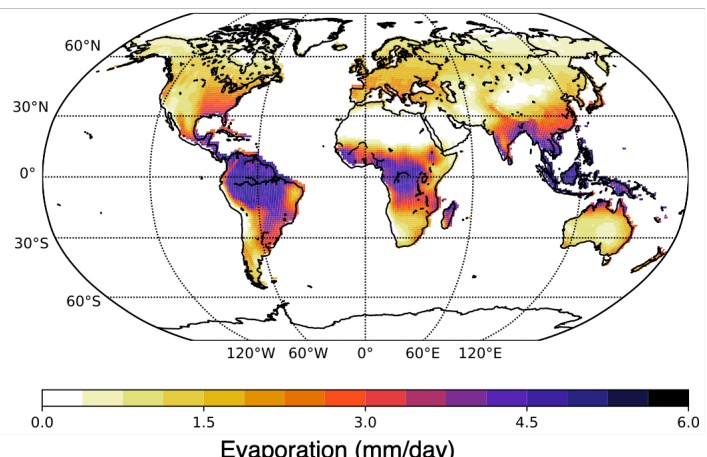

## b. Comparison

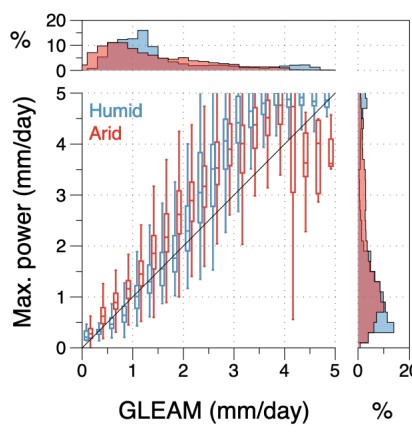

**Figure 12.** a. Annual mean evaporation derived from maximum power, thermodynamic equilibrium partitioning, and precipitation. b. Comparison of the estimate to the GLEAM evaporation dataset (Miralles et al., 2011; Martens et al., 2017). The linear regression among the two estimates yields a correlation coefficient of $r^2 = 0.85$ and a slope of $m = 1.25$. After Kleidon (2021b).

Kinetic Energy (TKE) budget of the boundary layer (see, e.g., textbooks by Stull, 1989; Arya, 1998). It can easily be shown that the buoyancy production term can be rewritten as a Carnot limit (Appendix D). Maximum power then corresponds to a
maximisation of the buoyancy production term, which involves the interaction of the buoyancy flux with surface temperature, that is, that more buoyancy production results in stronger cooling of the surface. This effect is what leads to the maximum power limit, as shown in Section 2.4. Hence, thermodynamics adds an important and relevant constraint to surface-atmosphere exchange as it limits the magnitude of the buoyancy production term in the TKE budget.

The maximisation of buoyancy production, combined with the equilibrium partitioning, then shapes evaporation patterns
on land. This is consistent with expressions for potential evaporation rates in hydrology, particularly the estimate by Priestley and Taylor (1972). While the Priestley-Taylor estimate includes an empirical coefficient of about 1.26, some studies have questioned the need for this coefficient when evaluating this estimate against observations at sub-daily scale (de Bruin et al., 2016; Conte et al., 2019). Hence, it would appear that the concept of potential evaporation is equivalent with evaporation operating at its thermodynamic limit as described here.

This interpretation is, however, quite different to previous studies which applied MEP or related optimality approaches to evaporation (Wang et al., 2004, 2007; Wang and Bras, 2011; Kleidon and Schymanski, 2008; Porada et al., 2011; Yang and Roderick, 2019; Tu et al., 2022). To start, the phase transition associated with evaporation does not produce entropy, as it proceeds at saturation, that is, in thermodynamic equilibrium, and is therefore reversible. This assumption is reflected in the derivation above (cf. Eqn. 14). Entropy is, however, produced when the saturated air is mixed with the unsaturated atmosphere
near the surface as it acts to moisten the lower atmosphere, bringing it closer to saturation and thermodynamic equilibrium (cf.

circle (2) in Fig. 11). This mixing requires motion, which is accomplished mostly by buoyancy, even when the resulting sensible heat flux is often much smaller than the latent heat flux. While the outcome described here can be interpreted as an evaporation rate that is maximised given the constraints of the energy balance and mixing requirements (or its entropy production), it would obscure the central role of buoyant transport in shaping this outcome.

The interpretation of evaporation given here provides complementary information to describe the thermodynamics of the whole hydrologic cycle that includes the critical input from the surface by evaporation. Previous studies (e.g., Pauluis and Held, 2002a, b; Laliberte et al., 2015) focused on the parts of the hydrologic cycle that takes place within the atmosphere, looking at moist convection as the result of a moist heat engine. This moist heat engine is driven by the heat release upon condensation (circle (3) in Fig. 11), generating moist convective motion within clouds, and it results in precipitation and dehumidification within the atmosphere. However, the replenishment of atmospheric moisture relies, in the end, on the moisture input from the surface by evaporation. The approach described here shows how thermodynamics constrains evaporation by setting the magnitude of turbulent fluxes, and by its partitioning into sensible and latent heat. While the sensible heat flux relates to buoyancy production at the surface, maintains surface-atmosphere exchange, and generates the power for dry convection, the power that can be derived from the latent heat flux is only realised when it condenses within the atmosphere. As this happens at the temperature when condensation occurs, which is colder than the surface temperature, the power output is reduced, while the remaining part is dissipated by the mixing of evaporated moisture from the surface into the unsaturated atmosphere.

What this means is that the magnitude of hydrologic cycling is determined by the surface input by evaporation, with the thermodynamic constraints described above. An implication of this perspective is that the hydrologic sensitivity to global warming, that is, the increase of mean precipitation with an increase in mean temperature, can be described from an atmospheric perspective (Takahashi, 2009), but it can equivalently be described using a thermodynamic surface perspective on how evaporation changes with surface temperature (Kleidon and Renner, 2013b; Kleidon et al., 2015). The sensitivity of hydrologic cycling to global warming of 2-3 %/K (Held and Soden, 2006) is notably lower than the 6-7%/K inferred from the Clausius-Clapeyron relationship, and this lower sensitivity is related primarily to the sensitivity of equilibrium partitioning (cf. Eq. 16) to surface temperature (see Kleidon and Renner, 2013b; Li et al., 2013).

In summary, evaporation is an example for an Earth system process that reflects maximum power, but it is not directly involved in the maximisation. Its magnitude is characterised by the thermodynamic equilibrium conditions at the surface where the phase transition takes place and by the buoyant transport, by which the vapour is transported and mixed into the atmosphere. As evaporation balances precipitation in the global mean, the strength of hydrologic cycling is thus strongly constrained by thermodynamics and optimality, although in an indirect way. This interpretation is consistent with boundary layer theory, where maximum power adds a constraint to the magnitude of the buoyancy production term, and with the hydrologic sensitivity to global warming.

### 3.3 Photosynthesis and terrestrial ecosystems

Photosynthesis, ecosystems, and life in general, represent an entirely different set of thermodynamic Earth system processes (Figure 13). They involve chemical transformations associated with their metabolic activities, which sustain their growth and

maintenance of biomass, the dynamics of their populations and resulting food webs and, ultimately, biogeochemical cycles. Chemical reactions, in turn, involve changes in energy and entropy as chemical elements are arranged into different compounds. Specifically, these changes apply to the energy by which electrons are bound to the nuclei and the random motion of molecules that we characterise as heat. These two forms of energy are associated with molar and thermal entropies. The second law of thermodynamics is reflected in these reactions, aiming to minimise their Gibbs free energy, that is, free energy in chemical form (with the definition of free energy used here, this minimisation of Gibbs free energy is equivalent to a reduction of free energy to zero). To understand the factors that limit these dynamics relates to how much chemical free energy was generated in the first place. This free energy is generated by photosynthesis. Hence, the question of how thermodynamics and optimality apply to ecosystems relates first and foremost to how it constrains photosynthetic activity within the Earth system. The generated free energy then sets the magnitude of the dissipative dynamics of the biosphere.

In the following, I describe a primary limitation of photosynthetic activity of terrestrial terrestrial ecosystems in which thermodynamics does not constrain the conversion of light into carbohydrates in photosynthesis directly, but rather an indirect limitation by gas exchange similar to how thermodynamics constrains hydrologic cycling (based on Kleidon, 2021b). To do so, I first review basic, energetic arguments to evaluate the theoretical upper limit on photosynthetic efficiency from past literature. I then provide an explanation for the much lower, observed photosynthetic efficiency that is based on gas exchange limitations and illustrate this interpretation quantitatively with well-established numbers. The question of how thermodynamic optimality applies to vegetation activity then translates into the question of how the gas exchange limitation can be minimised. I provide examples on such optimisations from past literature and relate them to the interpretation given here.

The efficiency of photosynthesis in generating energy from sunlight has been evaluated over the last decades, theoretically (Duysens, 1958; Radmer and Kok, 1977; Landsberg and Tonge, 1980) and by using observations (Monteith, 1972, 1977). The first step in photosynthesis takes place in the photosystems, which use sunlight to split water and perform the work of charge separation (marked by circle (1) in Fig. 13). This converts the radiative energy of the solar photons with 680 and 700 nm wavelengths into electric energy in form of electrons and protons. The photosystems require a minimum of eight photons for deriving the energy to fix one molecule of carbon dioxide (known as the quantum yield efficiency, e.g., Emerson (1958)). These photons yield 14.4 eV of energy, while it takes 13.6 eV to separate the electron from the hydrogen atom. This first step in photosynthesis from radiation to electric energy thus seems highly efficient since most of the energy of the absorbed photons is transferred into electric energy.

The next steps incorporate this electric energy into ATP (Adenosine triphosphate, a chemical energy carrier associated with cell metabolisms) and further into sugars by the Calvin cycle (circles (2) and (3) in Fig. 13). The efficiency of the whole conversion from light to carbohydrates can then be estimated from the ratio of chemical energy output in form of carbohydrates to radiative energy input by absorbed light. Carbohydrates contain about 470 kJ/mol C of energy, while the photons supply an equivalent of about 1390 kJ/mol C (using the Avogadro number to convert the 14.4 eV to energy/mole). The ratio of energy output to input yields an efficiency of 470 kJ/1390 kJ = 34%, which is about what can be seen in the performance of photosynthesis in low light conditions (Hill and Rich, 1983). This would support the notion that photosynthesis operates near its theoretical limit. Observations of terrestrial ecosystems, however, show a much lower efficiency with less than 3% of the

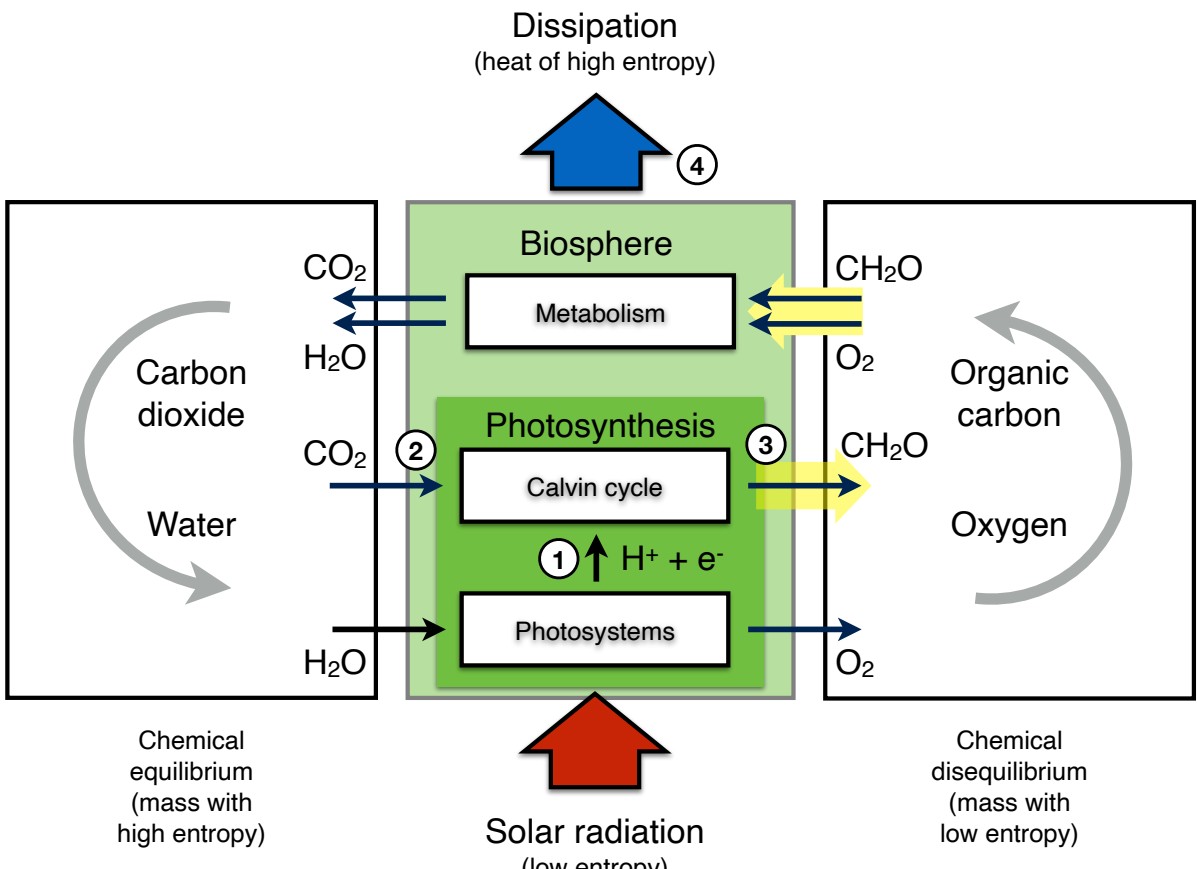

**Figure 13.** Schematic diagram to illustrate the thermodynamics of the biosphere, the associated forms of disequilibrium, and the key processes involved in converting solar radiation into free energy and its dissipation into heat of high entropy. The numbered circles mark places at which thermodynamic conversions and entropy production takes place: (1) Photosystems perform the work of charge separation in the light-dependent reactions; (2) $CO_2$ is taken up from the environment, requiring dissipative mixing as a transport mechanism; (3) The Calvin cycle converts $CO_2$ into carbohydrates in the dark reactions of photosynthesis using short-lived energy carriers; (4) The metabolic activities of the producers as well as of the consumers dissipate the chemical free energy in organic carbon and oxygen back into heat, resulting in thermal entropy production. Overall, the activity of the biosphere is reflected in a state of chemical disequilibrium in form of reduced, organic carbon compounds and oxygen (right box) in contrast to carbon dioxide and water (left box). Note that in the process of taking up $CO_2$, vegetation evaporates a substantial amount of water much greater than what is needed by the chemical conversion shown here.

absorbed solar radiation being converted by carbon uptake (Monteith, 1972, 1977). Does this much lower efficiency observed in ecosystems imply that their activity is not constrained by thermodynamics?

This apparent discrepancy in efficiency can be explained by considering that the gas exchange of carbon dioxide and water between the vegetative cover and the atmosphere limits carbon uptake by terrestrial ecosystems, and that this gas exchange is limited by thermodynamics (Kleidon, 2021b). To bind the solar energy into carbohydrates, vegetation needs to take up carbon

## a. Carbon uptake efficiency estimate

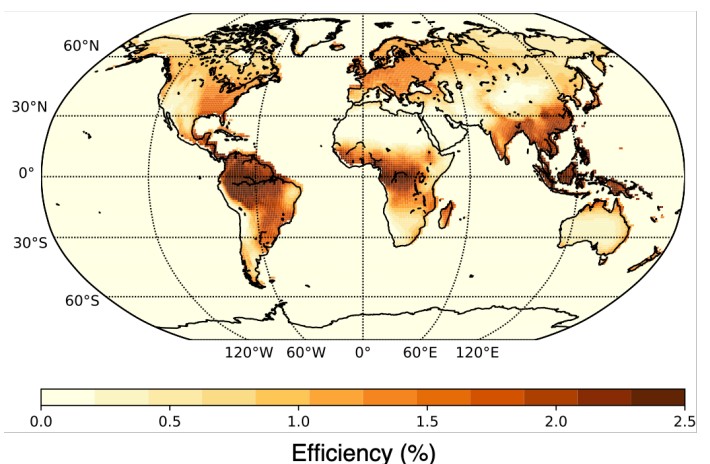

## b. Comparison

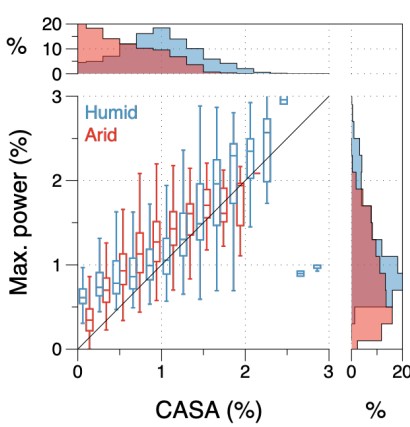

**Figure 14.** a. Thermodynamic efficiency of terrestrial carbon uptake by photosynthesis estimated using the thermodynamically constrained gas exchange based on the evaporation estimate inferred from max. power as shown in Fig. 12. (b.) Comparison of the thermodynamic efficiency to an estimate derived from the CASA biosphere model (Randerson et al., 2017; Ott, 2020). The linear regression among the two estimates yields a correlation coefficient of $r^2 = 0.56$ and a slope of $m = 0.79$. After Kleidon (2021b).

dioxide from the atmosphere, and in doing so, inevitably loses water. This gas exchange is well known. It takes place through the stomata of the leaves of the canopy that make up the vegetative cover, and results in the evaporation rate over vegetated land being dominated by the transpiration of vegetation. This may seem to suggest that plants can control and reduce their water loss through stomatal functioning. However, when focussing at larger scales of terrestrial ecosystems rather than individual plants, it is well known that ecosystems have relatively little control over how much water they evaporate because the vegetative cover

only closes its stomata when the availability of soil water becomes restricting. At the climatological scale, evaporation rates on land are then primarily shaped by radiative conditions and water availability set by precipitation. These two potentially limiting factors then, for instance, define the Budyko framework (Budyko, 1974) that is commonly used in hydrology (e.g., Milly, 1994; Koster and Suarez, 1999; Gerrits et al., 2009; Roderick et al., 2014), which characterises land evaporation of being either energy- or water limited. When evaporation is not restricted by water availability, it is strongly thermodynamically

controlled process, as just described in Section 3.2. Consequently, the gas exchange of water vapour is as well. The ratio of the simultaneous exchanges of water vapour and carbon dioxide, the so-called water use efficiency, is relatively fixed across ecosystems, with values between 2 - 4 grams of carbon uptake for a kilogram of water lost, as inferred from flux measurements (Law et al., 2002) or derived from global datasets (Kleidon, 2021b). This leads to the interpretation that thermodynamics does not constrain the energy conversions involved in terrestrial photosynthesis directly, but rather indirectly by constraining the

intensity of gas exchange through the stomata.

When we combine the water use efficiency with the factors that set the thermodynamic limit of evaporation, we obtain an estimate for the efficiency by which photosynthesis converts sunlight into carbohydrates. This efficiency is the combination of diverse losses from the energy available from the absorbed solar radiation: To start, about 50% of the absorbed solar radiation is partitioned into turbulent heat fluxes at maximum power (Eq. 8) while the other half is lost by the exchange of thermal radiation. Then, about 70% of the turbulent heat fluxes are represented by the latent heat flux (Eq. 16 evaluated for the global mean temperature of $T_s = 15°C$), resulting in the evaporation rate when the supply of water does not limit evaporation (as shown in Section 3.2). We next need to convert the water use efficiency into an energy equivalent value to translate an energetic loss of water to an energetic gain of carbon. To do so, we combine the median water use efficiency from observations of 2 gC/kgH$_2$O (Kleidon, 2021b) with the energetic gain of 470 kJ of chemical energy for the 12 gram of carbon contained in sugar and the energetic loss of about 2.5 MJ of latent heat associated with the water loss of one kilogram. This yields an energy equivalent of the water use efficiency of 2 gC/kgH$_2$O x 470 kJ/12 gC / 2.5 (MJ/kgH$_2$O) = 3.1 %. Taken together, this yields an overall efficiency of 50% (maximum power partitioning) x 70% (equilibrium partitioning into latent heat) x 3.1% (water use efficiency) = 1%.

This estimate reproduces the low efficiency reported for terrestrial ecosystems, although there are clearly some uncertainties associated with it. For instance, the partitioning into latent heat increases with surface temperature (cf. Eq. 16), while the water use efficiency shows some spread (first quantile of 1.6 gC/kgH$_2$O to third quantile of 2.6 gC/kgH$_2$O in Kleidon (2021b)) and variation across biome type (Law et al., 2002). A more detailed evaluation of the resulting pattern of the efficiency of photosynthetic carbon uptake (Figure 14) resembles nevertheless the pattern inferred from a more complex terrestrial carbon cycle model (Randerson et al., 2017; Ott, 2020) quite closely. This implies that the photosynthetic activity of terrestrial ecosystems indeed appears to be thermodynamically constrained, but not directly by the conversion of light into carbohydrates, but rather by solar radiation being the driving force for the gas exchange needed to supply plants with carbon dioxide.

This interpretation of how thermodynamics limits the activity of terrestrial ecosystems is consistent with established notions in plant ecophysiology, such as the central role of water use efficiency in ecosystem productivity and their response to atmospheric concentrations of carbon dioxide (e.g., Schimel et al., 2015). It provides novel insights, because it identifies a different, relevant limitation on ecosystem productivity. Typically, ecosystem productivity is viewed as being either light- or water limited (Churkina and Running, 1998), and these limitations are formulated in semiempirical ways. When looked at it from thermodynamics, it shows that light is not a limiting factor, but rather the gas exchange, which in turn is driven by solar radiative heating. In other words, the correlation in observations between light availability and productivity at the ecosystem scale does not originate from light limitation, but rather from a transport limitation, which in turn is driven by absorption of sunlight. This supports the well-established notion that the large-scale patterns of terrestrial productivity are shaped by climate. What this interpretation adds is that these patterns reflect the operation of photosynthetic carbon uptake at the limit indirectly set by thermodynamics.

The interpretation given here is, however, quite different to previous applications. Previous studies applied maximum power to ecosystems in a quite general way without being explicit of how this relates to the environment (Lotka, 1922a, b; Odum and Pinkerton, 1955). Other studies interpreted the greater absorption of solar radiation by vegetated surfaces compared to

bare ground as a direct manifestation of ecosystems maximizing their entropy production (e.g., Ulanowicz and Hannon, 1987; Schneider and Kay, 1994; Holdaway et al., 2010; Schymanski et al., 2010; del Jesus et al., 2012). Here, again, the interpretation described above does not contradict these more qualitative interpretations, but offers a more differentiated picture. Life does not simply enhance entropy production as this notion does not provide a mechanism by which this should be achieved. Using gas exchange as the bottleneck to photosynthetic activity can provide such a mechanism. As this gas exchange is driven primarily by the absorption of solar radiation, a lower surface albedo can drive more buoyancy and be advantageous in that it reduces the limitation imposed by gas exchange. So there is a beneficial effect for photosynthetic carbon uptake, which allows for more biospheric activity. It also enhances the entropy production by absorption of solar radiation, but this in itself has no benefit for biospheric activity as it merely increases thermalisation. This more differentiated interpretation thus links thermodynamics and optimality of life closer to the actual processes that are needed to sustain and enhance the dissipative activity of the biosphere.

We can then interpret biotic effects on carbon uptake in terms of thermodynamic optimality in that these would act to enhance the free energy generation of the biosphere by altering the gas exchange of water and carbon dioxide with the atmosphere. One example for how this gas exchange can be altered is the rooting depth of terrestrial vegetation. This sets the depth to which soil water can be used to maintain evaporation in dry periods where potential evaporation exceeds precipitation input (a limitation that was not explicitly considered in Sect. 3.2). Deeper roots allow for greater seasonal water storage within the soil, allowing vegetation to enhance its productivity and evaporation during dry periods (Nepstad et al., 1994; Kleidon and Heimann, 1998; Miguez-Macho and Fan, 2012; Wang-Erlandsson et al., 2016; Kleidon, 2023a). From a thermodynamic perspective, this corresponds to maximising the period of gas exchange and thus photosynthetic activity when evaporation is thermodynamically constrained rather than being constrained by water accessibility. Through greater soil water storage, terrestrial vegetation can enhance the productivity of the terrestrial biosphere, and thereby its power by about 10% (e.g., Kleidon, 2023a), with the effect being largest in the seasonal tropics. These effects can then result in rainforest evaporation during dry periods at rates solely set by thermodynamics that compare very well with observations (Conte et al., 2019), and consistent with the hydrologic outcome expected from the Budyko framework.

Another example for altering gas exchange is stomatal functioning, which affects how much carbon is taken up for a given water loss. This interpretation is consistent with the current theory of optimised stomatal functioning (Cowan and Farquhar, 1977; Katul et al., 2009; Medlyn et al., 2011) that assumes that stomata maximise photosynthetic carbon uptake reduced by the carbon cost of water loss (Lin et al., 2015). This optimised stomatal functioning is consistent with observed ecophysiological behaviour. In terms of thermodynamic optimality, this optimised functioning enhances the carbon uptake and photosynthetic activity for a given water loss set by thermodynamic or water availability constraints described in Sec. 3.2 above, that is, it enhances the water use efficiency to some extent. What this does not imply, however, is that vegetation maximizes carbon gain while *minimizing* its water loss, as it is sometimes incorrectly stated in the plant-ecophysiological literature. If water availability does not constrain evaporation, as in humid regions, ecosystems seem to evaporate near their potential rate, as described above and by the Budyko framework, yet aiming to maximise their carbon uptake by stomatal functioning while doing so.

This interpretation of thermodynamic optimality does not contradict the notions that ecosystems maximise and alter their environment, nor that they are dominated by thermodynamic constraints. It rather implies that when they maximise their power

associated with carbon uptake, they reach the limit imposed indirectly by thermodynamics, thus making the process predictable by thermodynamics and optimality and the climatic constraints related to radiative conditions and precipitation. It results in the emergent simplicity in biogeography described in the introduction, and supports classification approaches such as the Budyko framework that are based on this simplicity.

To sum up, carbon uptake by terrestrial vegetation is another example of an Earth system process that appears to reflect maximum power, but in an indirect way, similar to how evaporation is indirectly constrained by maximum power. The bottleneck appears to be the gas exchange between the vegetated surface and the lower atmosphere. It is reflected in the observed, relatively uniform value of water use efficiency across terrestrial ecosystems. This interpretation is consistent with the common recognition of the importance of water use efficiency for terrestrial ecosystems, and it can explain the low efficiency of

photosynthetic carbon uptake in observations. It provides a novel perspective in that it shows how intimately related vegetation productivity is to land-atmosphere exchange. Furthermore, it allows us to link vegetation activity to thermodynamic limits and it allows us to interpret optimality in terms of means to enhance gas exchange to increase the overall power generated by the terrestrial biosphere.

### 3.4   Putting things together

To quickly summarise these examples, let us place them back into the thermodynamic view of the Earth system shown in Figure 2. Generating vertical and large-scale horizontal motion, as described in Sections 2.4 and 3.1, deals with heat engines that convert differences in radiative heating into the free kinetic energy associated with motion (shown at the top of Figure 2). The heat transported by motion reduces the temperature difference (shown by the dashed line on the right of Figure 2), and this interaction results in the maximum power limit of these heat engines. The fluxes and temperatures that result from

this thermodynamic maximisation compare very well with observations at the climatological scale, providing support that thermodynamics and optimality play a dominant role in planetary functioning.

    The other two examples, terrestrial evaporation (Section 3.2) and carbon uptake by vegetation (Section 3.3), are shaped by this thermodynamic limit associated with motion (the next two inner shells shown in Figure 2, entitled "Cycling" and "Life"). For evaporation, it is the combination of the partitioning of absorbed radiation into heating and moistening of near-surface air

with the maximum power limit of generating vertical convective motion that sets the dominant constraint. For carbon uptake of vegetation, the energy supply is not heat, but the direct use of sunlight (as shown by the yellow box "Photosynthesis" in Figure 2). Yet, the magnitude is limited by the gas exchange of carbon dioxide associated with buoyant mixing between the canopy and the lower atmosphere. This constraint set by gas exchange can explain the low observed thermodynamic efficiency of photosynthesis of natural ecosystems. These two examples thus demonstrate that their magnitudes are constrained indirectly

by thermodynamic limits, but not by the heat flux - temperature difference trade-off that is at the center of the maximum power limit for generating motion. This emphasises the importance of looking at the relevant details of the processes involved in limiting planetary processes and their connections to transport as they do not fit into a generalised maximisation template as suggested, for instance, by the proposed MEP hypothesis.

## 4 Thermodynamics, optimality and Earth system science: Synthesis and what's next?

The examples provided in the last section support the overarching view of how thermodynamics and optimality limit the Earth system and shape its functioning. These examples show that it is *not* that maximum power (or related thermodynamic extremum principles, such as MEP) applies to each and every Earth system process. There is no magic, general thermodynamic optimality approach. Yet, because maintaining thermodynamic disequilibrium states of many Earth system processes, such as hydrologic and biogeochemical cycling requires motion, maximizing power associated with transport imprints this optimality on other

Earth system processes. In the following, I first discuss some of the limitations and potential extensions of the approach, link previously suggested optimality approaches to the view described here, and describe a few of its applications to do simple, physics-based Earth system science as well as its relevance.

### 4.1 Limitations and potential future extensions

There are, of course, several potential limitations of the formulations described here. After all, these formulations are kept as

simple as possible, so there is ample space of further improvement and refinement. In the following, I want to briefly describe some of these limitations and how they could affect the application of thermodynamic limits.

In the application of maximum power to atmospheric heat engines and associated motion, using radiative temperatures as the cold temperature for the heat engines is optimistic as it assumes that the emitted radiation to space is emitted at maximum entropy. Outgoing longwave radiation, however, is emitted as a mix from different levels of the lower atmosphere, so its

entropy is below its maximum value. This would then affect the outcome of the optimisation, resulting in less power and the values of the fluxes shifted to lower values (Dhara et al., 2016). In the example of large-scale atmospheric heat transport, the conservation of angular momentum was not considered, so that the outcome of maximum power is independent of the Earth's rotation rate. This outcome has been criticised by Goody (2007) in the context of the MEP hypothesis. There are a few attempts to include this constraint in the maximisation (Jupp and Cox, 2010), with the result then depending on the rotation

rate, less power is generated, and less heat being transported. This line of reasoning is supported by a set of climate model simulations in which the planetary rotation rate was varied and the energetics of the resulting atmospheric circulation was evaluated (Pascale et al., 2013). Interpreted in terms of maximum power, what these studies imply is that for certain cases of planetary rotation rate, an atmosphere may not be able to reach the maximum power limit due to angular momentum balance constraints. This, however, does not invalidate the maximum power limit, but rather emphasises the importance of a further

dynamical constraint. The inclusion of this additional constraint in the maximisation of power should then result in a more appropriate limit and associated estimates of the emergent climatological characteristics.

Also, the diurnal cycle and seasonality result in heat storage changes within the atmosphere that affect the Carnot limit, because these are associated with entropy changes within the system. This affects the outcome of the maximisation, the resulting power and fluxes (Kleidon and Renner, 2018). Furthermore, the maximisations of power of vertical and horizontal processes

are not independent, as the resulting horizontal heat advection affects the boundary conditions of the vertical heat engine, as recently shown by Ghausi et al. (2023). While these points would certainly affect the outcome of thermodynamic optimality,

they are unlikely to substantially alter the conclusion that these engines operate near their limit, because the agreement with observations is already rather good. The refinements described here are likely to improve the comparison to observations.

Evaporation was then related to maximum power indirectly by the transport limitation associated with generating vertical convective motion from the local radiative forcing. What this excludes is the contribution by moist convection, that is, the generation of vertical motion due to condensational heating within the atmosphere, and the contribution of mixing by surface friction from the large-scale circulation. These effects could play a role at certain times and in certain regions, for instance open water surfaces, where heating is much weaker than over land. Yet, the agreement with observation-based datasets is already very good, which suggests that these contributions are comparatively minor, at least on land, so that evaporation is predominantly shaped by the thermodynamic limit based on the local radiative forcing. This has some interesting implications that can be explored in future research, such as the lack of a dependency on the water vapor pressure deficit and wind speed (which is, for instance, assumed in the Penman-Monteith equation, Monteith (1965)). It may further help to improve the semi-empirical stability functions in drag parameterizations in climate models, where surface-atmosphere exchange is predominantly shaped by the frictional dissipation of large-scale wind fields.

Carbon uptake by vegetation was associated with the generation of vertical convective motion, so that the thermodynamic constraint applies indirectly, as in the case of evaporation over land. What this interpretation did not explain yet is the value of the water use efficiency that was derived from observations. In principle, it should be possible to derive this value from this approach as well, but it would require a refined representation of surface-atmosphere exchange. Specifically, it would need to represent the temperature difference between the surface and the near-surface atmosphere as it is this temperature difference that relates to buoyancy production at the surface as well as to the exchanges of water vapor and carbon dioxide. This would require further research, it could be compared to the wealth of data available from eddy covariance measurements, and could provide a simple, physics-based understanding of how this important property would potentially change under different climates. This would, however, not change the conclusion that thermodynamics and optimality apply mostly indirectly to vegetation activity through constraining gas exchange.

Overall, the conclusion that thermodynamics and optimality apply to natural Earth system processes predominantly in form of the maximum power limit from radiative heating to generating motion, and that the resulting motion then predominantly shapes associated Earth system processes would remain unaffected.

## 4.2 Optimality in context of the thermodynamic Earth system

I next want to relate this view of thermodynamics and optimality to other, previously suggested thermodynamic optimality approaches. A diverse range of thermodynamic optimality approaches have been suggested, focusing mainly on either minimising or maximising entropy production, power or dissipation. At first, it would seem that these contradict each other. What I want to show in the following is that it is mostly a matter of perspective, and that the focus on work being done and on the whole system helps to differentiate between these.

We start by taking a look at the thermodynamic setting of the whole Earth system, which is characterised by the planetary entropy budget. This budget is almost entirely determined by how much solar radiation is being absorbed, because most of it

is thermalised upon absorption, i.e. converted into heat at the low, prevailing temperatures of the Earth system. The processes that we typically associate with "dynamics", such as motion, hydrologic cycling, and biotic activity, contribute rather little to the planetary budget. If we exclude potential effects of albedo changes due to clouds and ice, this fixes the rate of absorption of solar radiation. This, in turn, fixes the magnitude of energy conversions as well as the total radiative entropy exchange by setting the radiative temperature of the emitted radiation to space. It leads to an almost fixed entropy exchange of the planet, so that the overall entropy production by Earth system processes is fixed as well as long as the planetary albedo does not change (which can also be seen in Fig. 10b). What this implies is that when one process enhances its entropy production, this comes mostly at the cost of another. In relation to optimisation, it implies that when one process maximises entropy production, the entropy production of other processes is minimised. It is thus a matter of perspective of which process is being looked at whether something is minimised or maximised, even if it sounds contradictory at first.

We next need to distinguish between processes that simply dissipate and produce entropy from those that perform work, generate free energy, generate a state of thermodynamic disequilibrium, drive the dynamics of dissipative structures and produce entropy when this free energy is dissipated (as described in Section 2.2, and shown in Figures 1 and 2 by the yellow boxes). When atmospheric processes operate at maximum power, they also maximise dissipation and associated entropy production. This maximisation comes at the expense of reducing, or even minimising, the entropy production by radiative exchange, which does not involve work being done, no free energy is generated and dissipated. The focus on work and the resulting generation of free energy is thus a critical aspect to distinguish Earth system processes, and it is those processes to which maximum power potentially applies to.

The application of optimality, however, does not only apply to the conversion of low-entropy solar radiation into free energy. Once free energy is generated from the solar forcing, it can either be directly dissipated, or converted into another form of free energy before it is eventually dissipated (Figure 15). This can result in sequences of free energy conversions. To these sequences, maximum power may also apply by maximizing the rate of free energy transfer down the sequence. An example for such a conversion chain in the climate system is the conversion of a fraction of the kinetic energy of atmospheric motion into wave energy at the ocean surface, with the other part of the free energy lost by frictional dissipation within the near-surface atmosphere. Wave energy is converted further into the kinetic energy of the wind-driven ocean circulation, with dissipative losses during the conversion. Maximum power in this context does not involve entropy, as it involves the maximized conversion of free energy into another form. Examples in the literature of such applications of maximum power are the shaping of dunes by winds (Rubin and Hunter, 1987), the physical limits of using wind energy as renewable energy at large scales (Miller et al., 2011; Miller and Kleidon, 2016; Kleidon, 2021a), or the mixing of freshwater in estuaries (Zhang and Savenije, 2019). It may also apply to the feedbacks that these sequences have on the forcing that would enable a higher generation rate (shown by the dashed lines in Figure 15), for instance by altering the radiative forcing. It remains to be seen how general the application of maximum power in such free energy conversion sequences is, leaving ample opportunities for future research.

We can apply this view of free energy conversion sequences to hydrology. A line of research applied "minimum energy expenditure" to understand fractal river networks (Rinaldo et al., 1992; Rodriguez-Iturbe and Rinaldo, 1997). This energy expenditure essentially relates to the minimisation of frictional dissipation within channel networks, similar to how frictional

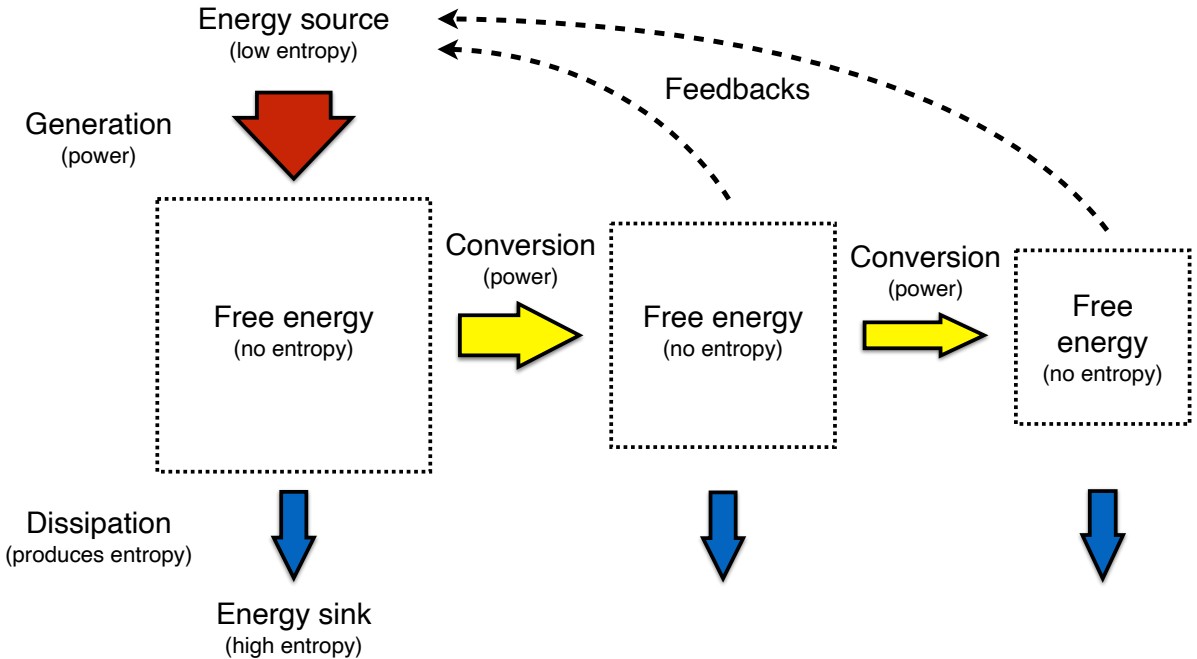

**Figure 15.** Schematic diagram to illustrate conversion sequences of free energy in the Earth system and potential applications of maximum power. Once free energy is generated from an energy source of low entropy, it can either be converted into another form of free energy by performing work or dissipated back into heat of high entropy. This can form a sequence of free energy conversions. Maximum power can also apply to these further conversions, or to the feedbacks that these processes have on the forcing of the generation process. See text for examples of Earth system processes.

dissipation is minimised in vascular networks (West et al., 1999). This minimisation fits into the picture here when we look at the whole system (Kleidon et al., 2013), and aim to understand the response of the whole catchment (Savenije and Hrachowitz, 2017). Rainfall at a certain elevation above sea level represents an input of free energy in form of potential energy, energy that was generated by hydrologic cycling by a moist heat engine driven by condensational heating within the atmosphere.

The input of free potential energy is converted into kinetic energy as water flows downhill, which is dissipated due to friction with the Earth's surface. When fractal river networks then minimise the frictional losses of water flow by the transition from sheet flow to channel flow (reducing the wetted perimeter per transported volume), it implies that the kinetic energy of the water flow performs maximum work on maintaining the flow and sustaining sediment transport. It represents one step in a free energy conversion sequence, starting with an atmospheric heat engine performing the work to lift water, and minimisation

of frictional dissipation being synonymous with the maximum power transfer to sediment transport along this conversion sequence. Similarly, power plants are engineered to minimize the internal loss by entropy production, referred to as "entropy generation minimization" as a design objective (Bejan, 1996, 1997).

Ecosystems represent such free energy conversion sequences as well that are fuelled by photosynthesis (rather than heat engines). Maximum power has been suggested to apply to living organisms dating back to Lotka (1922a; 1922b), with extensions to ecosystem development (Odum and Pinkerton, 1955; Odum, 1988; Fath et al., 2001; Joergensen, 2001; Hall, 2004). In principle, these previous applications are consistent with the view described here. There is nevertheless somewhat of a difference in terms of the focus. Here, the focus is not on the ecosystem itself, that is, the conversion sequence after the free energy was generated from the solar forcing, but rather on the link with the Earth's physical environment in which the free energy is being generated and how it acts to constrain this generation. This results in an interesting mismatch: While the constraint on biospheric activity derived here results in a flux per unit surface area, ecosystems operate with individual organisms. The missing link to overcome this apparent mismatch is accomplished by self-thinning laws, relationships that describe how the number of organisms within a certain area changes with their respective size. In this context, Enquist et al. (1998) have provided a simple, physical explanation for such self-thinning laws in tree communities given a resource constraint expressed per unit area (the total flux of water through xylems of the trees, which characterizes transpiration rate and carbon uptake). This work would directly connect the constraint described here to the organism-based view of ecosystems. Subsequent conversions of energy in food chains would then represent energy conversion sequences shown in Figure 15. The maximum power limit could potentially apply to each conversion step from one trophic level to the next, and it would include potential feedbacks by the effect of higher trophic levels on the producers, for instance by enhanced nutrient cycling (e.g., McNaughton, 1979; Vanni, 2002; Wolf et al., 2013; Buendia et al., 2018) that would allow for higher productivity or by changing the radiative forcing for the heat engines by changing greenhouse gas concentrations at geological time scales (Schwartzman and Volk, 1989; Berner, 1997; Kasting and Catling, 2003).

To conclude this discussion on thermodynamics and optimality within the Earth system, these examples show that there is ample room for further applications of maximum power and optimality to different aspects of Earth systems. Most of these applications would not involve thermodynamics directly, because maximum power would apply to the further conversion of free energy along energy conversion sequences. What I have described in the examples above, and is depicted in Figure 2, is merely the first step for a full description of these energy conversion sequences and their effects on the Earth system. This first step focuses on the processes that generate free energy from the solar forcing and represent the beginning of free energy conversion sequences (yellow boxes in Figure 2 and the leftmost box in Figure 15). This first step shows us that the overall, dissipative dynamics of Earth system processes are constrained and characterised very well by the combination of thermodynamics and optimality, directly or indirectly, at a rather simple, yet profound physical level.

### 4.3  Applications to Earth system science

At the end, I want to briefly describe three examples of how this thermodynamic Earth system approach can inform Earth system science.

The first example is the controversial Gaia hypothesis of Lovelock (1972a, b), which states that the Earth system with life is a self-regulatory system exhibiting homeostatic behavior at the global scale. While this may seem far-fetched, Lovelock's motivation came from the thermodynamics of life as described by Lotka (1925) and Schrödinger (1944), and the recogni-

tion that the Earth's chemical disequilibrium within the atmosphere is an imprint of life (Lovelock, 1965). In the context of the thermodynamic Earth system view described here, such homeostatic outcome can be imagined as the consequence of a biosphere that operates at its maximum power limit and the resulting interactions with the physical environment (Kleidon, 2023b). Section 3.3 showed that it is the physical gas exchange that limits biotic productivity, so that maximum productivity is intimately linked to a maximum in the gas exchange rate. At larger spatial and temporal scales, biotic productivity removes carbon dioxide from the atmosphere, which alters its greenhouse effect. The greenhouse effect, in turn, alters the radiative environment, which in turn can affect the outcome of the maximum power limit, and thereby the intensity of gas exchange. This could, in principle, result in an optimum state of the greenhouse forcing that would allow for maximum productivity. Such maximisation would then result in self-regulating behaviour, just like how atmospheric convection apparently self-regulates to maximise its power. As carbon dioxide is transformed by photosynthesis into reduced carbohydrates (in form of biomass) and atmospheric oxygen, this state would also reflect chemical disequilibrium of the Earth system. The outcome would then be rather similar to the postulated behaviour of the Gaia hypothesis, yet result as an outcome of thermodynamic maximisation associated with the biosphere and the interactions with the abiotic environment (Kleidon, 2023b). This would, however, require further evaluations.

The second example deals with more practical and quantitative applications to the Earth system response to global climate change. The maximum power limit has been applied to back-of-the-envelope estimates of how the hydrologic cycle responds to global warming and solar geoengineering (Kleidon and Renner, 2013b), its transient response (Kleidon et al., 2015), and to explain the land-ocean contrast in climate sensitivity (Kleidon and Renner, 2017). In each example, maximum power serves as a constraint to close the surface energy balance (as in Sections 2.4), so that analytical expressions were derived for the responses. The outcomes closely resemble the simulated responses of global climate models, demonstrating that thermodynamics and optimality dominate the response of the climate system to change. This should help us to advance our physical understanding of the dominant effects of global climate change that does not depend on the highly complex and often incomprehensible numerical simulation models. There are certainly many other aspects of global climate change that could be evaluated from this approach.

The last example is perhaps the most relevant in terms of practical implications. By focusing on how much work different processes in the Earth system can at best perform, this yields estimates of different forms of free energy that could in principle be used as renewable energy (Figure 16). The different means to generate free energy are shown in Figures 1 and 2 by the three yellow boxes, and they vastly differ in their efficiencies. Atmospheric heat engines can only convert about 10% (vertical) or 1% (horizontal) of the total absorbed solar radiation into kinetic energy and other forms derived from it because most of the potential of solar radiation to generate work is lost by thermalization. Photosynthesis operates with a somewhat similar efficiency of 1-2% of turning solar radiation into chemical energy in form of biomass because of the limitation imposed by gas exchange that is driven by the heat engines. Yet, photovoltaics has a theoretical efficiency above 70% (Press, 1976; Landsberg and Tonge, 1979; Kleidon et al., 2016). This much higher efficiency is possible because photovoltaics avoids the thermalization of solar radiation, so it does not use heat like an atmospheric heat engine, and it is not constrained by gas exchange like photosynthesis. Hence, photovoltaics has a much greater potential to yield renewable energy than all other

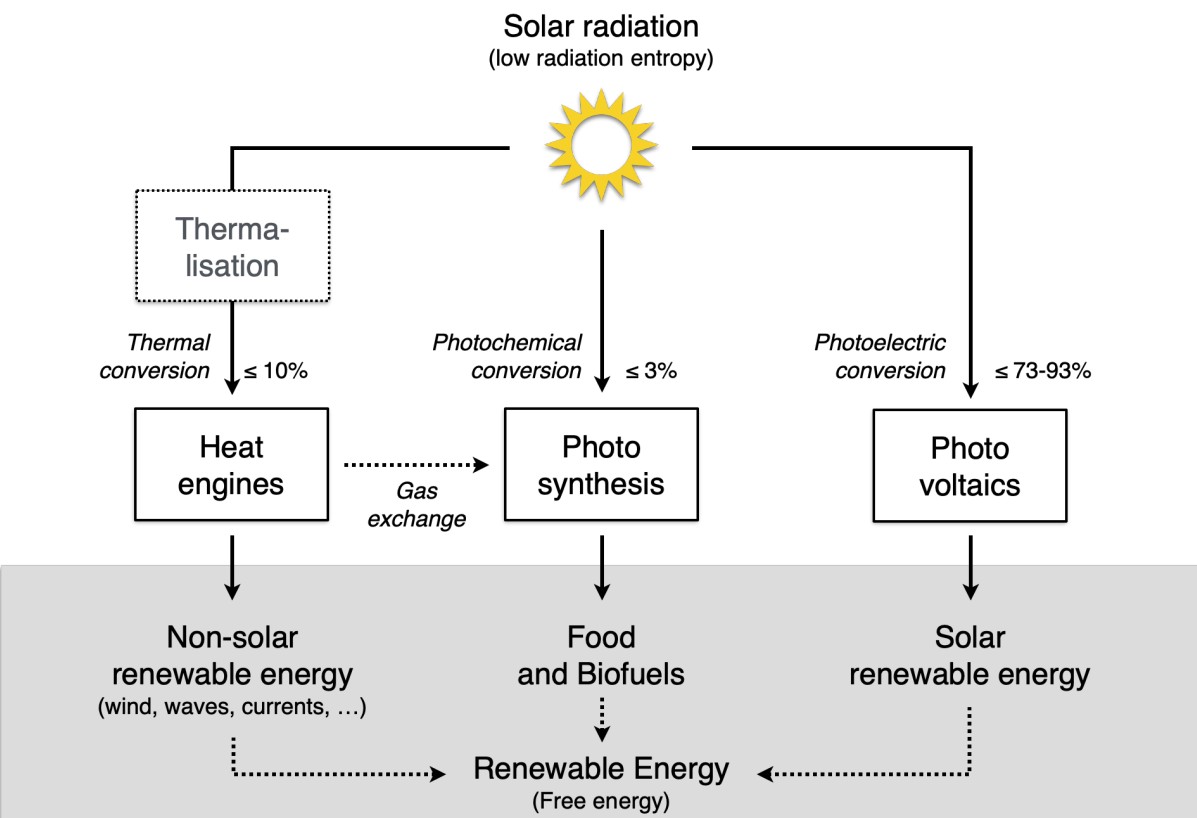

**Figure 16.** Schematic diagram to illustrate how the three mechanisms illustrated by the yellow boxes in Figures 1 and 2 generate free energy that can be used as renewable energy (after Kleidon et al. (2016)).

forms of renewable energy combined. This important insight comes directly from the application of thermodynamic limits in combination with an Earth system view, as the latter can explain why other forms of generating free energy are much less efficient than photovoltaics.

These three examples illustrate that the combination of thermodynamics, limits, connections, and interactions can provide a range of different insights for Earth system science. Clearly, there is ample space for future work. It does not mean to "reinvent the wheel", but rather a change in perspective. This perspective should focus on the work done and the free energy being generated by Earth system processes, where this free energy comes from, what it does, which factors constrain its generation, and what the consequences are. This should allow us to gain a more robust physical understanding of Earth system processes

as well as the role of interactions. It would allow us to understand the emergent simplicity of patterns and functioning in the Earth system by attributing this simplicity not to the simplicity of the processes involved, but rather to the notion that the dynamics are so complex and well tuned that these operate near or at the thermodynamic limit. This understanding should then further allow us to evaluate how Earth system processes evolve and react to change, not in terms of the usual state variables

such as temperature, but rather in terms of how the ability to perform work is affected and how the dissipative behaviour of the associated processes is going to change. In the end, this would imply that essentially all Earth system processes, from the physical processes of the climate system to the various activities in human societies, can be described by the same formalisms and concepts, allowing us to identify limits and generalities across the functioning of various Earth systems.

## 5   Conclusions

This review dealt with thermodynamic optimality approaches, aiming to provide a general and unifying view of how these apply to and shape Earth system functioning. At the core is the second law of thermodynamics. While every process needs to obey this law, I showed that the relevant constraint comes from its role in limiting how much work can be derived from the solar radiative forcing, generating free energy of some form. The relevance of thermodynamics to Earth system functioning thus boils down to its role in shaping the conversion of sunlight into work.

There are basically three different processes that accomplish this work in the Earth system: (i) heat engines convert differential heating and cooling into work, generating motion and shaping the emergent characteristics of the climate system; (ii) photosynthesis converts solar radiation directly into chemical free energy before radiation would otherwise be thermalised into heat, shaping the dynamics of the biosphere; and (iii) photovoltaics converts solar radiation directly into electricity, which has the potential to shape human societies in the future. The derived work generates free energy, and the dynamics within the associated systems convert this free energy in sequences into other forms before dissipating it back into heat with high entropy.

Thermodynamics and optimality then constrains the intensity of these energy conversion sequences by limiting the conversion of sunlight into work. For heat engines, optimality applies in terms of the maximum power limit. This limit originates from the combination of the well-established Carnot limit with the important effect of heat transport associated with motion, as this depletes the thermodynamic driver for deriving work, that is, it diminishes the efficiency term in the Carnot limit. This effect results in the maximum power limit. The examples in Section 3 showed that this maximum power limit can very well predict the magnitude of heat transport in the climate system as well as the distribution of surface temperatures. It then indirectly sets the magnitudes of hydrologic cycling and carbon exchange of terrestrial vegetation, because these two processes are limited by the transport that the heat engines of the climate system generate. We can thus understand the emergence of simple, climatological patterns as the reflection of atmospheric heat engines working as hard as they can, with its implications manifesting themselves in the linkages to other processes and further down the energy conversion sequences.

In conclusion, I hope that this thermodynamic Earth system view provides a more differentiated view of the critical role that thermodynamics and optimality play in shaping the Earth system. Different, seemingly contradictory approaches such as minimizing dissipation or maximizing power are not in contradiction, but a matter of how processes fit into the sequences of energy conversions once free energy is generated. This should provide ample opportunity for future research and understanding. After all, it would seem indisputable that performing work is of central importance for driving dynamics. It thus seems surprising that this focus on work and the factors that limit it has so far essentially been absent in Earth system science.

## Appendix A: Derivation of the Carnot limit of a dissipative heat engine

When the effect of dissipated heat is being included in the derivation of the Carnot limit, we should first note that we define the system such that we budget the thermal energy within the system. In other words, the kinetic energy that results from the work done by the heat engine is not considered as part of the system.

With this definition of the heat engine system, we write the first law of thermodynamics as

$$\frac{dU}{dt} = J_{in} - J_{out} - G + D \tag{A1}$$

where $U$ is the thermal energy of the system, $J_{in}$ and $J_{out}$ are the heat fluxes in to and out of the system, $G$ is the power of the heat engine that generates, for instance, free energy in kinetic form, and $D$ is the dissipation of the free energy back into heat.

The entropy budget of the system is then expressed by

$$\frac{dS}{dt} = \frac{J_{in}}{T_{in}} - \frac{J_{out}}{T_{out}} + \frac{D}{T_{diss}} + \sigma \tag{A2}$$

where $S$ is the thermal entropy of the system, $J_{in}/T_{in}$ and $J_{out}/T_{out}$ are the entropy fluxes in to and out of the system (at respective temperatures $T_{in}$ and $T_{out}$), $D/T_{diss}$ is the entropy production by the dissipation of free energy, characterized by a temperature $T_{diss}$, and $\sigma$ represents entropy production by unresolved processes.

To derive the Carnot limit, we first assume a steady-state setting, so that $dU/dt = 0$ and $dS/dt = 0$. This requires that $J_{in} = J_{out}$ and $G = D$ (Eq. A1). We further assume the best case in which $\sigma = 0$. Because $G = D$, we can obtain the limit to how much free energy can at best be generated directly from the entropy budget (Eq. A2) by solving it for $D$:

$$G = D = J_{in} \cdot T_{diss} \cdot \frac{T_{in} - T_{out}}{T_{in}T_{out}} \tag{A3}$$

If we consider the case where the dissipation occurs at the cold side of the heat engine, that is, $T_{diss} = T_{out}$, then we recover the Carnot limit in its original form (Eq. 4), with $T_{in}$ in the denominator. If we consider the other extreme case in which dissipation occurs at the warm side, so that $T_{diss} = T_{in}$, then we obtain the Carnot limit of a dissipative heat engine, with $T_{out}$ in the denominator. Since $T_{out} < T_{in}$, this limit yields somewhat more power, with the factor given by $T_{in}/T_{out} > 1$. Note that this could yield efficiencies greater than one, which are unphysical. It thus requires to carefully check the conditions of applicability before use.

## Appendix B: Entropy budget estimation

The thermal entropy budget terms shown in Figure 10b and d and summarized in Table 1 are estimated as follows.

The entropy production by thermalization due to the absorption of solar radiation within the two boxes, $\sigma_{s,t}$ and $\sigma_{s,p}$, is estimated in terms of the thermal entropy production (i.e., neglecting the expansion of the photon gas, which results in the factor of 4/3 in expressions for radiative entropy production, e.g., Lineweaver and Egan (2008))

$$\sigma_{s,t} = R_{s,t} \left( \frac{1}{T_{sun}} - \frac{1}{T_{s,t}} \right) \tag{B1}$$

and

$$\sigma_{s,p} = R_{s,p} \left( \frac{1}{T_{sun}} - \frac{1}{T_{s,p}} \right) \tag{B2}$$

with $T_{sun} = 5760$ K being the emission temperature of the Sun.

The entropy production by longwave radiative exchange as well as convective motion within each of the two boxes are estimated using the respective surface and radiative temperatures

$$\sigma_{l,t} = R_l(T_{s,t}) \left( \frac{1}{T_{r,t}} - \frac{1}{T_{s,t}} \right) \tag{B3}$$

and

$$\sigma_{l,p} = R_l(T_{s,p}) \left( \frac{1}{T_{r,p}} - \frac{1}{T_{s,p}} \right) \tag{B4}$$

where $R_l(T)$ is given by Eq. 11.

Entropy production by frictional dissipation, $\sigma_{fric}$, is estimated from the generated power by

$$\sigma_{fric} = \frac{1}{2} \cdot J \cdot \left( \frac{1}{T_{s,p}} - \frac{1}{T_{s,t}} \right) = \frac{1}{2} \cdot J \cdot \left( \frac{T_{s,t} - T_{s,p}}{T_{s,p} T_{s,t}} \right) \tag{B5}$$

where the factor of 1/2 is included to account for heat being transported from one hemisphere to another. Note that this expression is equivalent to the generation of kinetic energy at the Carnot limit and its subsequent dissipation at $T_{s,p}$. This expression is also equivalent to the thermal entropy production of mixing heat with temperature $T_{s,t}$ in the colder reservoir $T_{s,p}$ at a rate $J$.

The total thermal entropy production, $\sigma_{total}$, is then given by the sum of these contributions

$$\sigma_{total} = \frac{1}{2} \left( \sigma_{s,t} + \sigma_{s,p} + \sigma_{l,t} + \sigma_{l,p} \right) + \sigma_{fric} \tag{B6}$$

The factor of 1/2 is included here again for averaging.

## Appendix C: Estimating available potential energy

We can infer the available potential energy associated with the maximum power limit and compare it to estimates of the Lorenz Energy Cycle (LEC) from observations. To do so, we consider the vertical stratification in the atmosphere to be in hydrostatic equilibrium, an assumption similar to those made in estimates of the LEC terms. This equilibrium is modulated by air density, with a warmer atmospheric column having a lower density, less pressure drop with height, and hence a greater potential energy. This potential energy $U_{pe}$ can then be estimated by

$$U_{pe} = \left( \frac{R_a p_s}{g} \right) \cdot T \tag{C1}$$

with $p_s/g$ being the overall mass of the atmosphere per unit area with $p_s$ being surface air pressure, $g$ the gravitational acceleration, and $R_a$ the gas constant for air.

The temperature difference described by the two-box model (cf. Eq. 12) thus also describes a difference in potential energy, $\Delta U_{pe} = U_{pe}(T_{s,t}) - U_{pe}(T_{s,p})$. This difference, compared to the global mean, is shown in Figure 10f. With no heat transport, the temperature difference is largest, and so is the difference in $\Delta U_{pe}$. With increasing heat transport and a lower temperature difference, this difference in $\Delta U_{pe}$ declines as well.

However, only a fraction represented at best by the Carnot efficiency is available for conversion of $\Delta U_{pe}$ into available potential energy, $A$, equivalent to the conversion of the heat flux into kinetic energy. We thus get for $A$ the expression (Kleidon, 2021a):

$$A = \frac{1}{2} \cdot \left( \frac{R_a p_s}{g} \right) \cdot (T_{s,t} - T_{s,p}) \cdot \frac{T_{s,t} - T_{s,p}}{T_{s,t}} \tag{C2}$$

## Appendix D: Power and buoyancy production

A correspondence of maximum power associated with the sensible heat flux with maximum buoyancy production can directly be obtained when considering the buoyancy flux. This flux is commonly written as $\rho w'\theta'$, with $\rho$ being the air density (which is often left out in the notation for the Turbulent Kinetic Energy (TKE) budget, e.g., in the textbook of Stull (1989)), $w'$ the variation in the vertical wind component and $\theta'$ the variation in potential temperature. The buoyancy production term in the TKE budget, $G_b$, is then

$$G_b = \frac{g}{\theta} \cdot \rho w'\theta' \tag{D1}$$

where $g$ is the gravitational acceleration and $\theta$ is the mean potential temperature of the boundary layer.

This buoyancy production term can be linked to maximum power by first integrating it over the height of the boundary layer, $z_b$, to get the total buoyancy production in units of W m$^{-2}$ to make it comparable to the power derived from the Carnot

limit. The height of the boundary layer can be expressed by the temperature difference between the surface and the top of the boundary layer, using the dry adiabatic lapse rate, $\Gamma = c_p/g$, so that $z_b = c_p/g \cdot \Delta T$, with $\Delta T$ being the temperature difference between the surface and the top of the boundary layer.

The total buoyancy production term, $G_{b,total}$ then becomes

$$G_{b,total} = \frac{1}{2} \cdot c_p \rho \overline{w'\theta'} \cdot \frac{\Delta T}{\theta} \tag{D2}$$

or

$$G_{b,total} = \frac{1}{2} \cdot H \cdot \frac{\Delta T}{T_s} \tag{D3}$$

where $H$ is the sensible heat flux expressed as

$$H = c_p \rho \overline{w'\theta'} \tag{D4}$$

The factor of 1/2 comes from the approximately linear decrease of the buoyancy flux with height, and $T_s$ is the approximation for the potential temperature $\theta$ of the boundary layer.

In other words, the total buoyancy production term has the form of a thermodynamic Carnot limit of a heat engine driven by the sensible heat flux, with the temperature difference between the surface and the top of the boundary layer being the driving difference. The maximization of power would have the same outcome as described in Section 2.4, except for the additional factor of 1/2, which would reduce the power, but not the outcome of the optimum heat flux or temperature difference.

*Author contributions.* The author wrote the whole manuscript.

*Competing interests.* The author declares that he has no competing interests.

*Acknowledgements.* The author would like to thank Anke Hildebrandt for encouragement to write this paper and Sarosh Alam Ghausi, Jonathan Minz, Yinglin Tian and Vanessa Weinberger for constructive comments. The author thanks one anonymous reviewer, Remi Tailleux, Jonas Nycander, for their constructive and critical reviews, Vinod Gaur, Davor Juretic, and Hubert Savenije for additional discussions and comments, and the editor Gabriele Messori for comments. The author acknowledges funding through the ViTamins project by the VW foundation.

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
