# Peer review of "Working at the limit: A review of thermodynamics and optimality of the Earth system"

_Earth System Dynamics, 2022_

## Referee Comment (RC3)

**Review of "Working at the limit: A review of thermodynamics and optimality of the Earth system" by Kleidon**

Jonas Nycander

This manuscript proposes that both the horizontal and the vertical energy transport in the atmosphere, the evaporation and the biological productivity in the Earth system are determined by a simple optimality priciple. This principle is that the turbulent energy transport between two regions with different temperatures is obtained by maximizing the associated generation of free energy.

I find this hypothesis implausible, and the support for it given in the manuscript is weak. I therefore recommend rejection.

Detailed comments:

1. Let's consider a simple system with two boxes, similar to the one used in the manuscript. The temperature difference between the boxes is $\Delta T$, and the energy transport from the warm box to the cold box is $J$. The equilibrium temperature in each box is determined by a combination of external energy fluxes and the transport $J$, so that a larger transport cools the warm box and warms the cold one. Assuming a simple linear relation (as also done in the manuscript), this is described by the equation

$$\Delta T = \Delta T_0 - c_1 J. \tag{1}$$

Here $\Delta T_0$ is the temperature difference that would result from the external fluxes in the absence of transport between the boxes. We also define $J_0$ as the transport that would be required to equalize the temperatures, so that $\Delta T = 0$:

$$J_0 = \frac{1}{c_1} \Delta T_0.$$

We can regard the two states $(\Delta T, J) = (\Delta T_0, 0)$ and $(\Delta T, J) = (0, J_0)$ as extremes. A realistic state should be intermediate between them,

but to determine it we need to know how $J$ depends on $\Delta T$. Assuming that the transport is driven by the temperature difference, the simplest possible relation is

$$J = c_2 \Delta T. \tag{2}$$

The equilibrium state of the system is then

$$\Delta T = \frac{1}{1 + c_1 c_2} \Delta T_0, \tag{3}$$

$$J = \frac{c_1 c_2}{1 + c_1 c_2} J_0. \tag{4}$$

Thus, if $c_1 c_2 \ll 1$ it is close to one extreme, and if $c_1 c_2 \gg 1$ it is close to the other one.

Equation (1) is used in the manuscript to determine the temperature response to both the horizontal and the vertical atmospheric transport. The external fluxes are then the vertical radiative fluxes, and the coefficient $c_1$ (which corresponds to $2/b$ in eq. (11)) describes the Planck feedback, i.e. how much the temperature needs to change for the long-wave radiation to space to change by a prescribed amount.

In the manuscript, eq. (2) is not used directly to determine the transport. Instead, $J$ is chosen so that the generation G of free energy is maximized, subject to the constraint of the Carnot efficiency. Using eq. (3) in the manuscript, we have

$$G = \frac{\Delta T}{T_{\mathrm{w}}} J,$$

where $T_{\mathrm{w}}$ is the temperature of the warm box. We then substitute $J$ from eq. (1) and maximize $G$, neglecting possible variations of $T_{\mathrm{w}}$, as also done in the manuscript. This determines $\Delta T$, and by using eq. (1) also $J$:

$$\Delta T_{\mathrm{max}} = \frac{1}{2} \Delta T_0,$$

$$J_{\mathrm{max}} = \frac{1}{2} J_0.$$

Thus, the solution is exactly half way between the two extremes mentioned above.

Clearly, the optimization principle is equivalent to the choice $c_2 = 1/c_1$ in eq. (2). However, no motivation is given in the manuscript for why this particular choice should be universally valid. In fact, the external

fluxes and the transport between the boxes are usually independent physical processes. This certainly true in the cases considered in the manuscript, with the external fluxes given by Planck's law and the transport by the turbulent dynamics in the atmosphere, and it is easy to imagine changes that would affect one but not the other. For example, a faster planetary rotation should decrease the horizontal transport in the atmosphere without affecting the radiative fluxes, thus decreasing $c_2$ while leaving $c_1$ unchanged.

In the absence of physical motivation, the only support for the choice $c_2 = 1/c_1$ comes from the agreement with observations. However, in the case of horizontal transport in the atmosphere, the agreement is not very convincing. According to Fig. 8, $J$ is approximately $(2/3)J_0$, rather than $(1/2)J_0$. This very approximate agreement in only one data point could well be a coincidence, and if there is a fundamental reason for it, this is not given in the manuscript.

In the case of vertical energy transport in the atmosphere, no direct observations of this are given. Instead the energy transport is translated into evaporation by using the psychrometric constant, and the evaporation compared to observations. However, the theoretical prediction is first combined with a precipitation dataset to account for water limitation, and it is most likely the precipitation data that are mainly responsible for the seemingly good agreement in Fig 3b.

2. In the section about photosynthesis, the rate of $CO_2$ assimilation (usually called GPP, 'gross primary production') is obtained by multiplying the estimated evaporation by a typical value of the water use efficiency (WUE). Thus, no optimality assumption for the photosynthesis itself is involved. Instead, the result is a 'by-product' of the rate of evaporation obtained by using the optimality assumption for the vertical transport in the atmosphere.

The agreement with data is not surprising, given that WUE is known to vary only moderately, and that the evaporation estimate is constrained by precipitation, but is hard to see what this proves. There is hardly a direct causal relation, since excessive evaporation is typically detrimental to the plants, while a surplus of water is simply transported away as run-off. From the text the idea seems to be that the downward transport of $CO_2$ is governed by the same dynamics as the upward transport of water vapor, and that GPP is therefore another 'by-product' of the optimality assumption for the vertical transport in

the atmosphere.

However, the main resistance to $CO_2$-transport is not in the lower atmosphere, but in the stomata of the plants, and it is caused by the need to save water. Thus, $CO_2$ limitation and water limitation are two sides of the same coin, and increasing turbulent transport in the atmosphere is unlikely to increase GPP. The most natural explanation for the agreement seen in Figure 12 is that both the theoretical estimate and the real GPP are limited by the precipitation. This adds nothing to the common knowledge in the field.

---

## Author Comment (AC1)

**Response to Reviewer #1**

I thank the reviewer for the constructive and helpful review. In this response, I respond to all points raised and describe how I will accommodate these in the revision.

In the following point-by-point response, I will state the reviewer's comment in *italics*, followed by my response and description of the action that I will take in the revision.

***Major comment:*** *Overall, I think that the review is well written, focused on the aim of stimulating further research on the investigation of the climate system (in the wide sense, including the biogeochemical "layer") from the point of view of thermodynamic constraints. I do acknowledge, though, that there has been a lot of criticism on the motivation behind and the application of these approaches, and I believe that they should be more extensively addressed in this manuscript. I also think that this review might be a valid reference on the topic, that is why I recommend that the manuscript is accepted for publication, provided that a more robust set of references is included, and that the mentioned criticisms are taken into account, along with the specific comments and technical corrections proposed below.*

**Response:** Thank you for this assessment. I agree that providing more background of the criticisms will be helpful.

**Action:** In the revision, I will extend the part in the introduction on thermodynamic optimality principles (third paragraph), particularly regarding the criticisms that these have received. I will also add further references as suggested by the reviewer.

***Specific comments***

**Comment:** *ll. 35-37: here and elsewhere in the text, I believe that a more complete framework of the research literature on methods of computation of the entropy budget in the climate system. I can think, among others, of Goody 2000, Raymond 2013, Bannon 2015, Bannon and Lee 2017, Lucarini and Pascale 2014, Lembo et al. 2019...*

**Response:** Thanks for the suggestion, and I agree that this place would be a good place to give references to the entropy budget. Since the detailed evaluation of entropy production is not the main focus of the manuscript and because it is already quite long, I would prefer to keep it relatively short and focused on practical applications and estimates.

**Action:** I will add a paragraph in the introduction and briefly describe entropy budgets with some representative references. I will include mentioning the planetary entropy budget also in Section 2.5, and make connections to this budget at the end of the three examples in Section 3.

**Comment:** *l. 50: as the manuscript here proposed is a review, when the MEP principle is introduced, I think it is also worth informing the reader that the concept has not been unanimously accepted by the community in their theoretical derivation. I can think of Dewar 2007 or Grinstein and Linsker 2007, as examples of this ongoing debate...*

**Response:** Thank you for this suggestion. I agree that this is useful to add.

**Action:** I will extend part of the introduction and mention briefly the proposed MEP principle and its discussion, with some representative references.

**Comment:** *l. 130 and ll. 142-143: not sure I get the point here. Of course, it is impossible to evaluate the entropy production of the system in a microscopic sense, as jumps of quanta of energy. Is it relevant at all in this context?*

**Response:** The relevance of going down to the microscopic scale is that entropy is not just about heat, or thermal energy, but that there are other forms of entropy that describe radiation and chemical compounds. These aspects can be understood when describing how quanta of energy are being distributed at the microscopic scale, and how this results in the different forms of entropy. This picture is relevant when dealing with Earth system processes and their entropy production, particularly when it comes to deriving work from the solar radiative forcing. While the atmosphere derives work from entropy differences associated with differential heating, photsynthesis and photovoltaics are able to utilize the entropy of solar radiation, that is, before it turns into heat during absorption. Because of this, photosynthesis and photovoltaics could, in principle, derive much more work than atmospheric heat engines from the solar forcing (and, in fact, solar panels already derive much more free energy from solar radiation). This part is aimed to provide the basis to understand this critical difference when entropy is being discussed.

**Action:** I will rewrite parts of this section to clarify this aspect.

**Comment:** *l. 148: when I think of heat transport in the atmospheric medium, I do not see molecular diffusion as a mean of transport that is relevant in the macroscopic scale.*

**Response:** This part of the text describes the distribution of energy at the microscopic scale and thermodynamic equilibrium associated with the different forms of entropy. It does not deal with macroscopic phenomena like diffusion.

**Action:** I will rewrite this paragraph to clarify.

**Comment:** *l. 152: it would be interesting to know a bit more about what the author means when talking about "forms" of entropy, as it is not entirely clear at this point of the manuscript;*

**Response:** What is often not recognized is that Earth system processes deal with different types of entropy, not just with the concept of entropy in classical thermodynamics. In quantum physics/ statistical physics, this is very well established. This is what I aim to describe in this paragraph (and which is illustrated in Figure 2), although apparently, it was not clear to the reviewer.

**Action:** I will rewrite this paragraph and clarify, making it more specific what the change in the form of entropy is reflected in. For the sentence on line 152, this can be extended to be made more specific like this:

"When solar radiation is absorbed and heats the surface, it is not just the energy that changes its form from radiative to thermal energy. Also, the entropy changes its form, from entropy representing how photons are being distributed over wavelengths (radiative entropy) to how energy is distributed over the random vibrations and motions of molecules (thermal entropy)."

**Comment:** *l. 172: maybe "entropy change"?*

**Response:** No, it is not about that entropy does not change, but that free energy has no entropy. When, for instance, two reservoirs of heat have different temperatures, the total entropy of the two reservoirs is below its maximum. This maximum is reached when heat is transferred and the temperature difference is levelled out. The system could derive work from the initial temperature difference, which, if it is dissipated within the system, raises the entropy. In other words, the free energy that was generated has no entropy and only enhances the entropy of the system when it is being dissipated. This description is synonymous with exergy, although with the added notion that free energy is energy without entropy.

**Action:** As reviewer 2 also stated that this section is not specific enough, I will rewrite this paragraph.

**Comment:** *l. 173: related to my previous comment, if we are talking of "free energy" as a form of energy that is converted with no change of entropy, that would surely not be dissipation. That is why I am a bit confused by the whole definition of "free energy" that is proposed in this context. Could the author clarify on this point?*

**Response:** Ok, see explanation above.

**Action:** See action above.

**Comment:** *Figure 3: not sure I understood what is included in the term "Generation", although it is somewhat described in the text;*

**Response:** Given the previous comments and responses, and also those of Reviewer 2, I hope that this will become clearer in the revised version of the manuscript.

**Action:** See action above.

**Comment:** *l. 206: as far as I understand it, there are several more general ways to describe a thermodynamic cycle.*

**Response:** Well, when I look at textbooks (Feynman lectures on physics, Kondipudi and Prigogine) that I am familiar with, the Carnot limit is derived from a specific thermodynamic cycle, usually inferring the limit from specific steps of adiabatic expansion and isothermal heating and cooling. The derivation used in the manuscript starts directly from the laws of thermodynamics that do not invoke a specific thermodynamic cycle.

**Action:** I will make this sentence more specific by adding the references to the textbooks.

**Comment:** *ll. 241-242: this is in general not true, I believe. Despite the fact that you can of course have frictional dissipation anywhere, the contribution to the energy reservoirs is almost negligible, and I cannot think how it can affect the transport, at least in the atmosphere.*

**Response:** What I describe here are two extreme cases that are being considered where the dissipated heat is added. It is true that the heat being added is small, yet one can easily show that this assumption affects the temperature in the denominator in the Carnot efficiency. This has, for instance, be done previously by Renno and Ingersoll (1996) and Bister and Emanuel (1998). This distinction affects the work output of hurricanes, as shown by Emanuel (1999).

**Action:** I will clarify the text and state that these are two extreme cases being considered and that it affects the Carnot efficiency term, and add the reference to Emanuel (1999) to illustrate its relevance. I will add an Appendix with that derivation.

**Comment:** *ll. 306-307: I think that a similar formulation for the Carnot cycle within the climate system was provided in Pauluis and Held, 2002.*

**Response:** Pauluis and Held (2002) used the Carnot limit, as did previous work, e.g., by Renno and Ingersoll (1996). The important difference here is that the heat engine is coupled to the energy input from the surface, which results in the limit of maximum power.

**Action:** I will clarify the relation to previous work and the difference in the revision.

**Comment:** *l. 369: I think it is a bit misleading to suggest that the use of conceptual models is meant to facilitate proving a point in front of a reader. There is an illuminating communication by Isaac Held on the importance of establishing a hierarchy of models in order to understand how the climate system works, that might be of interest in this context (Held, 2005).*

**Response:** Thanks for making this point.

**Action:** I will add the reference and refer to the importance of having a hierarchy of models.

**Comment:** *ll. 436-438: as stated at the beginning, I am not again speculative arguments, but I find it hard to agree with this sentence, just because observations are in rough agreement with the proposed conceptual model.*

**Response:** I am not sure what the reviewer means by "speculative arguments" in the context of this paragraph. The calculation shows how much work can maximally be derived from differential solar radiative heating, and the resulting estimate compares well with observations, with respect to the power needed to sustain the large-scale circulation as well as the associated poleward heat transport. This was done in a two-box model, which, quite clearly, is very simple, although it similar to previous models that have been used before to demonstrate MEP with a very similar outcome. In my understanding, this is consistent with "Occam's razor" - a scientific explanation of a phenomenon based on physical principles with the fewest number of parameters needed.

There are certainly some limitations that were not considered in this extremely simple model, which I can add to highlight some of the uncertainties (e.g., the contribution by the Hadley cell and seasonal variations that I would take from the discussion in Kleidon 2021a). Yet, these would not affect the interpretation that this consistency implies that the atmosphere works close to its limit, that is, as hard as it can.

**Action:** I suggest I add a few sentences of the uncertainties.

**Comment:** *l. 449: if this was the aim, it would have been useful to give numbers in order to compare the different contributions. Maybe some table would have been helpful.*

**Response:** Very good point, thank you.

**Action:** I will add a table with the estimates of entropy production to support this difference between power and entropy production and extend the description to describe it more specifically.

**Comment:** *l. 464: it seems to me that you are rather arguing here that the conceptual model that has been designed in order to be consistent with observations is maximizing the energy conversions based on some thermodynamic constraints. Still, there is a missing step before one can claim that the atmosphere is actually operating at its maximum.*

**Response:** I do not understand which missing step the reviewer is referring to. It is not "some thermodynamic constraint", but rather power - that is, the work needed to sustain motion against friction. I can imagine that the reviewer may be referring to the potential mechanism of how this maximization may be achieved, an aspect that Reviewer 3 alludes to as well.

**Action:** I will add a subsection in section 2 to outline how the maximization of power may result from two contrasting feedbacks, as described in Kleidon (2016).

**Comment:** *ll. 469-470: but this is something you can also achieve by computing the material entropy production in the Lorenz Energy Cycle, as shown elsewhere (cfr. Lucarini et al. 2014, Lembo et al. 2019).*

**Response:** Yes, sure. As is described in the previous paragraph, the maximization of power, frictional dissipation or entropy production yield about the same result. The difference is that with power, we get a physical interpretation, which allows us to distinguish processes that perform work from those who do not (such as diffusion or radiative transfer), while entropy production is associated with every process.

**Action:** I will add references of MEP applications to the previous paragraph.

**Comment:** *ll. 551-553: shouldn't the phase changes also taken into account here? (cfr. Pauluis and Held 2002).*

**Response:** Actually, phase changes do not produce entropy in this equilibrium partitioning approach, because it occurs at saturation and is hence reversible. The irreversibility is associated with the mixing of the saturated surface air with the unsaturated near-surface air, but this entropy production by mixing does not affect the partitioning between sensible and latent heat.

**Action:** I will add this clarification in this discussion part and also include a more general description of the contribution of evaporation to the entropy budget (see also response to first specific comment above).

**Comment:** *ll. 607-609: I have nothing again these approximate calculations, but there are so many assumptions here that I really have the feeling no conclusions can be easily drawn here.*

**Response:** Actually, these are not many assumptions - the approaches that this calculation is based on are standard in vegetation modelling and land surface hydrology, and the numbers represent well-established values. Even if the values deviate, e.g., the water use efficiency is twice as high, it still results in an overall efficiency that is still much, much lower than the theoretical limit derived from radiation conversion.

**Action:** I will extend this estimate with additional references to support the numbers, and add the uncertainty estimate.

*Typos and technical corrections*

**Comment:** *l. 17: incredible -> incredibly;*

Thanks, corrected.

**Comment:** *l. 95: add "were" between "thermodynamics" and "developed";*

Thanks, added.

**Comment:** *l. 527: missing reference;*

Thanks, corrected.

---

## Author Comment (AC2)

**Response to Reviewer #2 (Remi Tailleaux)**

I thank the reviewer for the constructive and helpful review.  In this response, I respond to all points raised and describe how I will accommodate these in the revision of the manuscript.

In the following point-by-point response, I will state the reviewer's comment in *italics*, followed by my response and description of the action that I will take in the revision.

***Major comment 1: Free energy versus exergy versus APE.*** *I find the discussion of the concepts of free energy versus exergy versus APE somewhat confusing and unclear. In Tailleux (2013) cited by the author, I interpreted exergy as the available energy defined relative to a state of thermodynamic equilibrium with uniform temperature, and APE as available energy defined relative to a state of minimum potential energy obtained from the actual state by means of an adiabatic re-arrangement of mass as per Lorenz (1955) theory. In my mind, there is a significant difference between the two, as transforming exergy into useful work cannot be done without simultaneously destroying exergy and creating entropy irreversibly. In contrast, APE can be converted into useful work (kinetic energy) without the need to destroy any of it nor creating entropy irreversibly. In this regard, APE appears to be a `freer' form of free energy than exergy. Can the author try to highlight the differences and interrelations between the different concepts? Does the author define free energy relative to thermodynamic equilibrium, or does he also consider that APE is a form of free energy? An advantage of the exergy concept is that it provides a clear explanation of where the free energy comes from, i.e., from the convexity of the internal energy. If internal energy was not a convex function of entropy, it would not be possible to transform `heat' into `work'.*

**Response:**
Thank you for this elaboration.  What I describe in the manuscript is more or less consistent with what the reviewer writes, so his comments can be addressed by a revision.  In principle, free energy (or exergy) can be described as the fraction of energy within a system that, once dissipated, produces the entropy to bring the system from disequilibrium into a state of thermodynamic equilibrium and maximum entropy.  This is equivalent to what the reviewer writes, and I will elaborate on this in section 2.2 (see also response to Reviewer 1).

For the Earth system, this reference state of thermodynamic equilibrium is, however, not so meaningful.  This is because the planetary forcing in form of differential, radiative heating and cooling always generates temperature differences, and a state of thermodynamic equilibrium at the planetary scale is never achieved.  Hence, it seems to be more meaningful to think of free energy budgets in their respective steady states, and these are represented by generation and dissipation terms - that is, fluxes of free energy, rather than stocks of free energy.  When it comes to the generation of free energy, it can be seen from the derivation of the Carnot limit in section 2.3 that this generation is associated with an energy flux with no entropy as it does not enter the entropy budget.

Regarding the comment on APE, I think this can easily be rectified by noting that, essentially, the sum of available potential energy and kinetic energy represent both free energy, and the relative proportion of each of the components can freely be converted - just as it is the case in the dynamics of non-dissipative waves.

In terms of the "advantage of the exergy" concept, I would like to comment that the notion that free energy comes "from the convexity of internal energy" is quite a mathematical way to put it.  In this manuscript, I prefer to focus on the physics - and here, free energy comes from the disequilibrium associated with the frequency distribution of photons across wavelengths associated with solar radiation at the Earth's orbit.

**Action:**
I will clarify these points by extending description in Sections 2.2 and 3.1.  I will clarify the definition of free energy and the specific context of the Earth system as being a flux-driven system that includes a budget equation for free energy.  The point about APE is clarified in section

3.1. The point of solar radiation being at disequilibrium at the Earth's orbit will be clarified in section 2.5.

***Major comment 2: Dissipation/generation***. *It is not clear to me what the author means by 'dissipation' or `generation'. As regards dissipation, does the author mean 'viscous dissipation' or does his definition also include APE dissipation by diffusive processes, which Tailleux (2009) argued should be regarded as an additional form of Joule heating similar to viscous dissipation? The concept of APE dissipation seems important, because for simple fluids, it is proportional to the rate of irreversible entropy production by diffusive fluxes of heat, which means that there is a link between the production of entropy production due to the destruction of APE (viewed as a form of free energy) and the entropy production due to passive or dynamically inert heat diffusion through the background reference stratification. As regards 'generation', it seems to me that it is an ambiguous term. Indeed, I'd like to point out that in the oceans, for instance, oceanographers still do not agree on how to define the power input due to surface buoyancy fluxes as discussed in Tailleux (2010).*

**Response:**
These are good points! Yes, dissipation of kinetic energy (i.e., friction) represents viscous dissipation in the context of the paper, that is, the conversion of kinetic energy back into heat (or, microscopically, less random motion of molecules becomes random). The thermal dissipation part, e.g, by heat diffusion, would already be captured by the entropy production term in the entropy budget that was used to derive the Carnot limit. After all, thermal dissipation is not associated with the generation and dissipation of free energy.

In terms of APE dissipation, I think this can be seen to apply to the atmosphere as well, although this dissipation is done by the mass flow associated with the heat flux that drives the atmospheric heat engine. At present, I do not see how it is, however, relevant to the atmosphere. I therefore suggest that I mention this concept and the difficulty to clarify power inputs for the ocean briefly at the places in the manuscript where these things are being discussed.

**Action:**
I will describe dissipation in Section 2.2 more clearly and mention that such free energy budgets are not as simple as they may sound, as, e.g., the definition of power input for oceans is still being debated. The concept of APE diffusion and its role for oceans will be mentioned at the end of section 3.1.

***Major comment 3:*** *On the role of moisture. I believe that the author makes an important point in pointing out that the climate system differs from the kind of heat engines considered in textbooks in that the net heating and cooling are not fixed but to be obtained as part of the solution. Regarding the role of moisture, the author may be aware that Laliberte et al. (2015) have argued that moisture reduces the efficiency of the atmospheric heat engine relative to a dry one. It would be of interest if the author could comment on this and whether he agrees or disagrees from his perspective.*

**Response:**
Thank you for raising this question! The main difference of how atmospheric heat engines and the hydrologic cycle is being described here is that these processes are described including the energy balances of the system, and, as the reviewer rightly describes, that the heat fluxes are not fixed, but constrained. This picture results in a view that hydrologic cycling is a part of the turbulent fluxes that are derived from the maximum power limit (as in Section 2.4), and that the intensity of cycling is set by the thermodynamic equilibrium conditions (Section 3.2). The power output from dry convection is thus certainly reduced, which is consistent with Laliberte et al. (2015), because a part of the turbulent fluxes is represented by evaporation. Yet, as evaporation rates can be predicted very well by this approach, the magnitude of hydrologic cycling also in the atmosphere is strongly constrained by the surface input, and by the thermodynamic constraints described here. It would thus actually seem that - similar to section 3.1 - the surface inputs

already contain the strong constraint on hydrologic cycling within the atmosphere that makes it predictable from thermodynamics, as well as the response to global climate change.

**Action:**
I will add a paragraph of discussion to section 3.2 to discuss the relationship of the Laliberte et al. (2015) paper and related papers on thermodynamics and hydrologic cycling.

***Major comment 4:*** *Carnot efficiency versus maximum power efficiency. The author cites a number of studies related to maximum power. Wouldn't it be relevant to cite endoreversible heat engines and the ideas of Curzon-Ahlborn here?*

**Response:**
That's a good point! The Curzon-Ahlborn efficiency assumes that there is a dissipative loss term in a heat engine due to diffusive heat exchange at the system's boundaries. These loss terms are not considered here because atmospheric heat engines operate mostly with direct radiative heating and cooling, in which heat does not need to diffuse into the engine from a hotter source or out to a colder sink. The diffusive laminar layer right at the interface between the surface and the air is rather small and also not a fixed property of the engine, while the radiative cooling of the atmosphere has no diffusive layer across which the cooling takes place.

**Action:**
I will add this clarification to the discussion in the manuscript in section 2.3, which derives the Carnot limit.

***Minor comments:***

*Abstract. I find the first sentence to be particularly obscure. Could the author be somewhat more specific? Second line: what does 'it plays' refer to? If this refers to 'thermodynamics and optimality', plural should be used.*

I will reformulate the first two sentences of the abstract to clarify.

*Lines 23-26. Could the author be more specific about the 'simplicity' of the examples discussed? This seems to be a bit subjective and left to the appreciation of readers.*

I will modify the second paragraph and be more specific.

*Line 87. Might be important that entropy increases irreversibly only in closed systems.*

Understood - this will be reformulated, although it is important to note that even for open systems, entropy can only increase, but this takes place outside the system.

*Lines 180. Free energy budgets are important. It seems to me that these are widely used but not called that way. What about APE budgets or the kind of available energy budgets considered by Bannon and co-authors for instance? Could the author be more specific about what he has in mind exactly? What is his definition of free energy?*

Agreed. I will expand the description, as outlined in the response to Major Comment 2.

*Line 184: 'These all can be formulated in terms of free energy budgets, a concept that is rarely used in Earth system sciences' I am not sure that this is true, as for me, APE budgets or budgets of available energy represent 'free energy budgets' so the author needs to explain what a 'free energy budget' would look like and provide examples.*

Well, this sentence does not state that free energy budgets are not used at all, just that they are rarely used. I'll revise this paragraph and make the description more explicit. See also response to Major Comment 2.

*Line 233 – This derivation of the Carnot limit is general and quite different to common textbook derivations. I am a bit puzzled by this statement, as this is the derivation of the Carnot efficiency I am used to, and the one I was taught as an undergrad.*

I'll rephrase this paragraph.

*Line 240 – is given by G = D. This may be the case but because I think that there is no consensus about how to define G and D unambiguously in all cases (see my remarks about APE dissipation above), as is the case in the oceans for instance, it is unclear how to link MEP and ideas of maximum power or dissipation in the most general case.*

The goal of this manuscript is not to provide a general recipe on how $G$ and $D$ are being described - after all, this likely contains process-specific information. The examples in Section 3 actually show that heat engines primarily apply to the atmosphere, but other processes may nevertheless quite closely related to these heat engines (like evaporation and carbon uptake, examples described in Sections 3.2 and 3.3). I will include this clarification when introducing free energy budgets in Section 2.2 (see also related responses above).

*Lines 275-277. What about endoreversible engines and the Curzon-Ahlborn efficiency?*

As described in response to Major Comment 4, I will add a comment on the Curzon-Ahlborn efficiency already a bit earlier when the Carnot limit is derived in Section 2.3.

*Lines 376 – Against friction. As well as against APE dissipation may be.*

In the atmosphere, as far as I know, it is essentially the friction in the boundary layer and at the surface that dissipates the kinetic energy of the large-scale atmospheric circulation. I'll make this more specific in the revision.

*Lines 385-390. This assumes that we understand how to define and quantify both power and dissipation in all possible cases, but I don't think this is true, e.g., discussion in Tailleux (2010) for the oceanic case.*

This description does not claim to apply in all possible cases but deals specifically with the large-scale atmospheric circulation in this section, with no reference being made to the ocean. Also, the MEP applications of Paltridge dealt with the atmosphere. I do not think that it needs further clarification, so I suggest no alteration of this section.

*Line 443 – Here the author uses the term 'frictional dissipation' where he only used 'dissipation' before. See the need for clarifying the term 'dissipation' in major comments above.*

This term will be more clearly explained in the revision. See response to Major Comment 2.

*Line 450 – I agree that this is a particularly important point that I think will need to be more fully recognised and expanded upon in the future.*

Thank you! I guess this does not need modification, so I will leave it as it is.

*Line 463 – 'so that these conversions neither generate nor dissipate free energy' This is true only for APE but would not apply to exergy for instance, whose transformation into useful work requires destruction.*

As I am not referring to exergy (or free energy) here, but to descriptions of the Lorenz Energy Cycle, I think this sentence does not need adjustments.

*Line 527 – GPCP – missing reference*

Will be fixed.

---

## Author Comment (AC3)

**Response to Reviewer #3 (Jonas Nycander)**

I thank the reviewer for the critical review. In this response, I respond to all points raised. Most of these points can relatively easily be addressed by further clarifications. It should also be noted that this manuscript does not describe novel results, but summarizes previously published papers on the topic that were reviewed by experts of the respective disciplines and brings these together into one bigger picture in this review. So I think that the reviewers recommendation to reject is unjustified.

In the following, I tried to disentangle the review into separate points, addressed these points by clarifications, and describe how I will accommodate these in the revision. In the point-by-point response, I will state the reviewer's comment in *italics*, followed by my response and description of the action to be taken.

*Major Comment 1a: Thus, the solution is exactly half way between the two extremes mentioned above. Clearly, the optimization principle is equivalent to the choice $c_2 = 1/c_1$ in eq. (2). However, no motivation is given in the manuscript for why this particular choice should be universally valid.*

**Response:** I agree that the mechanism by which this maximum can be obtained is not included. The maximization is likely achieved by the way that frictional dissipation arranges itself within the atmosphere - a process that is currently described by semi-empirical parameterizations in climate models. This is consistent with earlier studies, e.g., Kleidon et al. (2003, GRL) with a simplified GCM that showed that one could get a state of Maximum Entropy Production (MEP) by adjusting the friction time scale. As I describe in this manuscript, MEP is very similar to maximizing power and dissipation.

Furthermore, the dynamics to maximization likely involve two contrasting feedbacks that operate at different time scales and that are modulated by the intensity of friction (Kleidon, 2016, Cambridge Univ. Press): If we perturb a system that is initially at rest, a fast, "power enhancing" positive feedback would enhance perturbations that yield more power, kinetic energy, more heat transport, thereby feeding back to yield more power; and a slower, negative "gradient depletion" feedback by which more heat transport leads to a greater temperature depletion, hence less power, kinetic energy and heat transport. The result is then that at the maximum power state, the state is governed overall by a negative feedback (as also described by Ozawa et al., 2003, Rev. Geophys.).

I would also like to point out that I do not claim that this choice is universally valid. It simply is a thermodynamic limit, so that the atmosphere cannot generate more power than this. For the present Earth, it seems that atmospheric motion operates at this limit, particularly when it comes to dry convection over land. It thus represents a highly relevant constraint that can provide closure to the surface energy balance. But other constraints, such as those imposed by the angular momentum balance, could reduce this limit to lower power and dissipation. This has, for instance, be shown by idealized GCM simulations by Pascale et al. (2013, Planetary Space Sciences) in which the rotation rate was varied.

**Action:** In the revision, I will add a subsection in Section 2 that describes the feedbacks involved in the maximization, I will clarify that this is a limit which is not necessarily reached, and add the reference to Pascale et al. (2013) in the discussion (Section 4.1) where the role of rotation rate is being discussed.

*Major Comment 1b: In fact, the external fluxes and the transport between the boxes are usually independent physical processes.*

**Response:** If I understand the reviewer correctly, I disagree. It is well known that the magnitude of atmospheric heat transport is reflected in the radiative exchange at the top of the atmosphere - they are not independent from each other. With no heat transport, thermal emission would

balance solar absorption, but observations show very clearly that tropical regions absorb more solar radiation than they emit, while polar regions emit more than they absorb.

**Action:** I will add this explanation at the beginning of Subsection 3.1.

**Major Comment 1c:** *This certainly true in the cases considered in the manuscript, with the external fluxes given by Planck's law and the transport by the turbulent dynamics in the atmosphere, and it is easy to imagine changes that would affect one but not the other. **For example, a faster planetary rotation should decrease the horizontal transport in the atmosphere without affecting the radiative fluxes, thus decreasing $c_2$ while leaving $c_1$ unchanged.***

**Response:** With a faster planetary rotation rate and less heat transport, there is certainly a change in the radiative fluxes, as the tropics get warmer and the polar regions get colder. This is consistent with the simulations by Pascale et al. (2013) mentioned above.

**Action:** I think that the addition of the Pascale et al. (2013) reference, as described above, should address this point in the revision.

**Major Comment 1d:** *In the absence of physical motivation, the only support for the choice $c_2 = 1/c_1$ comes from the agreement with observations. However, in the case of horizontal transport in the atmosphere, the agreement is not very convincing. According to Fig. 8, J is approximately (2/3) $J_0$, rather than (1/2) $J_0$. **This very approximate agreement in only one data point could well be a coincidence**, and if there is a fundamental reason for it, this is not given in the manuscript.*

**Response:** There is not just the approximate agreement in terms of the heat flux, but also in terms of the power generated, which is similar in magnitude to the generation rate of kinetic energy in the Lorenz Energy Cycle and which serves as another means of comparison. Since power, heat transport, and temperature difference are closely interconnected, the agreement cannot be a coincidence. Furthermore, there are previous studies which have obtained very similar results from applying the maximization of entropy production (MEP), as described in the text.

**Action:** I will add this clarification to the end of Section 3.1.

**Major comment 1e:** *In the case of vertical energy transport in the atmosphere, no direct observations of this are given. Instead the energy transport is translated into evaporation by using the psychrometric constant, and the evaporation compared to observations. **However, the theoretical prediction is first combined with a precipitation dataset to account for water limitation, and it is most likely the precipitation data that are mainly responsible for the seemingly good agreement in Fig 3b**.*

**Response:** No, this is not correct. The thermodynamic estimate works equally well in humid regions where water availability does not constrain evaporation rate (i.e., precipitation rates are greater than potential evaporation), so that the rates in these regions are not determined by precipitation. This is shown in the revised figure 10b below, in which the individual gridpoints were separated according to water limitation into humid (blue) and arid (red) grid points. Since there are roughly as many humid (54.5%) and arid (45.5%) grid cells in the dataset, thermodynamics acts to constrain and set the evaporation rate in many regions on land, while less than half are determined by precipitation rate.

[Figure]

**Action:** I will update Figure 10b in the manuscript with the separation between humid and arid regions (as shown above) and the associated text. For consistency, I will update Figure 12b as well.

*Major comment 2a: In the section about photosynthesis, the rate of $CO_2$ assimilation (usually called GPP, 'gross primary production') is obtained by multiplying the estimated evaporation by a typical value of the water use efficiency (WUE). **Thus, no optimality assumption for the photosynthesis itself is involved**.*

**Response:** This is correct - there is no optimality assumption in here, and the low efficiency of photosynthesis is related to the bottleneck of gas exchange. Hence, the magnitudes of photosynthetic carbon uptake reflect the thermodynamic limit of atmospheric heat engines indirectly because these set the rates of evaporation and, therefore, gas exchange.

The application of optimality/maximization to photosynthetic carbon uptake is being described at the end of this section (lines 638ff). One primary example regards the role of rooting zone depth. In seasonal climates with prolonged dry periods, like in much of the seasonal tropics, a sufficiently deep rooting zone allows for seasonal soil water storage to contribute to evaporation from the surface. This effect is well known and documented in publications (see text for some references). This effect of rooting zone depth can enhance the power generated by terrestrial vegetation by 12%, as recently estimated by Kleidon (2023, Ecology Economy and Society - the INSEE Journal, https://doi.org/10.37773/ees.v6i1.915).

So there are aspects by which terrestrial vegetation can maximize the free energy generation further, as described in the manuscript.

**Action:** I will clarify the role of where optimality comes in and extend the description of the example of rooting depth.

*Major comment 2b: Instead, the result is a 'by-product' of the rate of evaporation obtained by using the optimality assumption for the vertical transport in the atmosphere. The agreement with data is not surprising, given that WUE is known to vary only moderately, and that the evaporation estimate is constrained by precipitation, **but is hard to see what this proves**.*

**Response:** As described above - evaporation is more than constrained by precipitation, and it needs a more differentiated picture to describe actual evaporation rates. This, after all, is one of the holy grails in hydrology to figure out the partitioning of precipitation into evaporation and

runoff.  So the notion that evaporation is simply precipitation is not correct.  I refer to the explanations above regarding humid regions and the importance of sufficient seasonal soil water storage.

**Action:** See actions described above.

***Major comment 2c:*** *There is hardly a direct causal relation, since excessive evaporation is typically detrimental to the plants, while a surplus of water is simply transported away as run-off. From the text the idea seems to be that the downward transport of $CO_2$ is governed by the same dynamics as the upward transport of water vapor, and that GPP is therefore another 'by-product' of the optimality assumption for the vertical transport in the atmosphere.  However, the main resistance to $CO_2$-transport is not in the lower atmosphere,* **but in the stomata of the plants, and it is caused by the need to save water***.*

**Response:** No, this is not correct.  As described in the text, current stomatal optimization theory describes their function to maximize the carbon uptake for a given water loss (Medlyn et al. 2011, cited in the text).  Additionally, the agreement of evaporation rates to those derived by thermodynamic constraints (as shown by the diagram above) suggest that stomata play very little role in controlling evaporation rates (see also Conte et al. 2019, cited in the text).

**Action:** I will include further clarification in the paragraph as there is an apparent need to clarify some of the basics.

***Major comment 2d:*** *Thus, $CO_2$ limitation and water limitation are two sides of the same coin, and increasing turbulent transport in the atmosphere is unlikely to increase GPP. The most natural explanation for the agreement seen in Figure 12 is that both the theoretical estimate and the real GPP are limited by the precipitation.* **This adds nothing to the common knowledge in the field.**

**Response:** Again, the reviewer has a too simplistic view on the controls of evaporation (see responses above).

**Action:** See above.

---

## Author Response (AR1)

**Revision of MS "Working on the limit: A review of thermodynamics and optimality of the Earth system"**

Dear Editor,

I would like to submit the revised version of my manuscript for possible publication in Earth System Dynamics. In this revision, I have accommodated the constructive comments by the reviewers, as well as those made by a few other colleagues who have commented on the original manuscript that I shared with them. The changes in text in the manuscript are either highlighted in blue (to address reviewer comments) or in red (to address comments by colleagues).

The revised version of the manuscript includes mostly clarifications and additional text to explain things to address the reviewer comments. Specifically, I addressed the reviewer comments as detailed further below (and as described in more detail in the replies).

Best regards,
Axel

**Reviewer #1:**

- Added a brief description of the entropy budget to the introduction. To support this description, I moved the old Figure 6 to make it Figure 1 and modified it to complement this description.

- Added some of the criticisms about MEP to the introduction, including the references mentioned by the reviewer.

- Clarified the forms of entropy and its relevance in Section 2.1.

- Clarified the description of free energy in Section 2.2 and its budgeting by including Eq. 1.

- Clarified the two cases of the Carnot limit in Section 2.3, supplemented with its derivation in the new Appendix A.

- Clarified the difference to Pauluis and Held's work in Section 2.4.

- Added the reference to Held (2005) at the beginning of Section 3.

- Clarified the level of agreement and discrepancies in Section 3.1.

- Added a table (new Table 1) with the entropy budget in Section 3.1.

- Clarified link to Lorenz Energy Cycle, material entropy production, and added reference to Lucarini et al. (2014).

- Clarified that phase transitions play practically no role in entropy production in hydrologic cycling (because they proceed at saturation) at the beginning of Section 3.2 and the discussion at the end of Section 3.2.

- Provided more detail of how the limit on carbon uptake is estimated and described some uncertainty in Section 3.3.

**Reviewer #2 (Remi Tailleux):**

- Clarified description of free energy and its relation to exergy in Section 2.2, and the link to APE at the end of Section 3.1.

- Clarified the use of generation and dissipation in Section 2.2, including a new equation (Eq. 1) to illustrate this budgeting.

- Added a discussion on how the description of the hydrologic cycle of the manuscript relates to the work of Laliberte et al. (2015) at the end of Section 3.2.

- Added a note on the Curzon-Ahlborn limit at the end of Section 2.3.

- Minor comments addressed as described in the reply to the review.

**Reviewer #3 (Jonas Nycander):**

- Clarified throughout the manuscript that the manuscript does not claim that there is one general thermodynamic optimality principle for Earth system science, but that it requires a more differentiated view of direct vs. indirect manifestations of thermodynamic limits.

- Added a new Section 2.5 to provide a possible mechanism of how a state of maximum power is achieved by feedbacks.  This is not new work, but summarizes previously published work in Kleidon et al. (2013) and Kleidon (2016).  I also included that this line of research would need more work to be substantiated.

- Added an evaluation plot in Figure 6 in Section 2.4 regarding the evaluation of the max. power limit.

- Added text and a new Figure 8 to clarify that external radiative fluxes and heat transport between the tropics and the poles are not independent from each other.

- Added text and reference to Pascale et al. (2013) about the role of planetary rotation rate in Section 3.1 and in the discussion (Section 4.1).

- Clarified the level of agreement and discrepancies in Section 3.1.

- In evaluation of evaporation (Section 3.2), I separated humid from arid regions to clarify that evaporation are not only shaped by precipitation (see also modified Figures 12 and 14).  I also added the link to potential evaporation, a well established concept in hydrology.

- Extended the description on how optimality links to vegetation activity using this thermodynamic interpretation at the end of Section 3.3, specifically regarding the effect of rooting depth and stomata.

---

## Referee Report (RR1)

**Review of the revised version of "Working at the limit: A review of thermodynamics and optimality of the Earth system" by Kleidon**

Jonas Nycander

Some changes have been made in response to my comments, but none of them address my main objections. I therefore still recommend rejection.

Much of the text consists in lengthy qualitative descriptions of how entropy is produced as a result of various energy transformations. These descriptions are correct, but unsurprising, since everything said follows immediately from basic thermodynamic principles. They also do not lead to any quantitative predictions.

The scientific core of the paper is a consideration of a simple model with two boxes. All radiative energy fluxes are assumed to be either given externally or determined by the temperature of the boxes. There is also a turbulent energy flux $J$ between the boxes. A consideration of the energy budget then leads to an equation of the form

$$\Delta T = f(J), \tag{1}$$

where $\Delta T$ is the temperature difference between the boxes, and $f$ is a decreasing function. This relation shows how an increased turbulent flux leads to a smaller temperature difference.

To close the problem we need one more equation that specifies how the turbulent transport depends on the temperature difference:

$$J = g(\Delta T), \tag{2}$$

where $g$ is an increasing function. Solving the two coupled equations (1) and (2) gives the actual values of $J$ and $\Delta T$.

The function $f$ describes the radiative fluxes and the energy budget, while $g$ describes the turbulent flux. Thus, the two functions describe independent physical processes. But, of course, the turbulent flux affects the temperatures, and thereby indirectly the radiative fluxes, as a part of determining the state of the whole system.

Some information can be obtained from eq. (1) without using eq. (2). The temperature difference is maximal when there is no turbulent transport, i.e. $J = 0$ corresponds to $\Delta T = \Delta T_{max}$. Furthermore, the turbulent transport cannot be larger than what is required to equalize the temperatures, so $J = J_{max}$ corresponds to $\Delta T = 0$. Thus, $\Delta T$ must lie between 0 and $\Delta T_{max}$, and $J$ must lie between 0 and $J_{max}$.

Turbulent transport is in general a complex process that is difficult to describe. The author claims that this is not needed, and that $J$ can instead be determined by maximizing the free energy production, subject to the Carnot constraint (the 'optimum principle'). If the $\Delta T$ is much smaller than the absolute temperature, this means that the product $J\Delta T = Jf(J)$ should be maximized. Further assuming that $f$ is linear, this gives the solution

$$J = \frac{J_{max}}{2},$$

$$\Delta T = \frac{\Delta T_{max}}{2}.$$

The only support for the optimum principle is that it gives good agreement with observations. In the case of meridional heat transport in the atmosphere we have, according to the manuscript, $\Delta T_{max} \approx 60$ K, which gives the prediction $\Delta T \approx 30$ K. This should be compared to the observed value 20 K. I don't find this agreement very impressive, given that we know a priori that $\Delta T$ lies between 0 K and 60 K. Moreover, it is clear that many factors, for example the planetary rotation rate, affect the turbulent transport, and therefore the function $g$, without affecting the function $f$. Thus, changing the rotation rate will change the state of the system, while the proposed optimum solution remains the same.

The idea that you can determine the state of the system without knowing anything about the mechanisms of turbulent transport is scientifically unsound. This can clearly be seen in the case of vertical energy transport. Here, $\Delta T_{max}$ is the temperature difference given by a pure radiation balance. Defining $\Delta T = T_r - T_s$ and using the same notation and approximations as in the manuscript we have

$$\Delta T_{max} = \frac{1}{k_r}(R_s + R_{l,down} - R_{l,0}).$$

The prediction of the optimum principle is again

$$\Delta T = \frac{\Delta T_{max}}{2}$$

which is eq. (9) in the manuscript.

This result is different from the established theory, which says that the vertical temperature gradient is close to the adiabatic lapse rate. The mechanism is that the turbulent transport is negligible if the atmosphere is stably stratified, and increases very rapidly if the temperature decreases upward faster than the threshold for convective instability. Translating this to the box model used in the manuscript, this means that the function $g(\Delta T)$ in eq. (2) is essentially zero for $\Delta T < \Delta T_{conv}$ (where $\Delta T_{conv}$ is the threshold for convective instability), and increases very rapidly for $\Delta T > \Delta T_{conv}$. The solution of the problem is then essentially that $\Delta T$ equals the smallest of $\Delta T_{max}$ and $\Delta T_{conv}$. Thus, if $\Delta T_{max} > \Delta T_{conv}$ (as in the real troposphere) the state is entirely determined by the mechanism of turbulent transport, which is ignored by the optimum principle.

The simple theory outlined above is a corner stone of climate science since many decades, and it is based on an clear physical mechanism. If you want to replace it by another theory, which lacks physical motivation, it is not enough to show that its prediction of $\Delta T$ agrees roughly with observations. You must explicitly compare it with the established theory, and show that it gives a better prediction.

The discussion of photosynthesis seems to imply that the evaporation somehow drives the photosynthesis. That is a strange idea, since increased evaporation is in general detrimental for plant growth. The text is not very clear on this, but in the reply to my previous comments the author rejects my comment that the main resistance to $CO_2$ transport is in the stomata, and is caused by the need to save water. He also states that "stomata play very little role in controlling evaporation rates".

Restricting water loss while at the same time allowing $CO_2$ to enter the leaves is the reason for the existence of the stomata. This can be seen in any text book, for example 'Plant Physiology', 3rd ed, Lincoln Taiz and Eduardo Zeiger, p. 59 (open source). A quick web search reveals similar statements in the abstract or opening paragraph of a large number of scientific articles. Here are two examples: "Almost all water used for plant growth is lost to the atmosphere by transpiration through stomatal pores on the leaf epidermis. By altering stomatal pore apertures, plants are able to optimize their $CO_2$ uptake for photosynthesis while minimizing water loss." (L.T. Bertolini et al, Front. Plant Sci., 2019, vol. 10, https://doi.org/10.3389/fpls.2019.00225). "In order for plants to function efficiently, they must balance gaseous exchange between inside and outside the leaf to maximize CO2 uptake for photosynthetic carbon assimilation (A) and to minimize water loss through transpiration. Stomata are the 'gatekeepers' responsible for all gaseous diffusion, and they adjust to both internal and external environmental stimuli governing $CO_2$ uptake and water loss. " (T. Lawson and M.R.Blatt, Plant Physiol., 2014, vol.164, pp 1556–1570.)

This means that the plants need to restrict the flux of $CO_2$ when access to water is limited. The facts that the water use efficiency is fairly constant across ecosystems, and that the energy efficiency of photosynthesis is far below the theoretical limit, imply that this regime of simultaneous water and $CO_2$ limitation is normal. The plants handle this by regulating the size of the stomata openings.

---

## Author Response (AR2)

11 July 2023

**Response to Re-Review of my manuscript "Working at the limit: …"**

Dear Gabriele,

Thank you (and the reviewers) for another round of reviews.

I have now revised the manuscript according to these comments - they mostly dealt with additional clarifications.  The response to the comments raised in your letter as well as the comments by reviewers #1 and #3 are addressed separately point-by-point on the following pages.

In the revised manuscript, the modified text is highlighted in cyan.  In addition to text modifications to address reviewer comments, I have also added a further, new reference on maximum power that was just published (Ghausi et al. 2023, PNAS, https://doi.org/10.1073/pnas.2220400120).  I added this new reference for two reasons: first, it shows a much better agreement with observations at seasonal scales (that is, if more detail is added).  Since the "more detail" part is not the main focus of the manuscript, I simply refer to it in Section 2.4 (lines 426ff).  Second, this study also shows potential aspects for refinement of the maximum power limit, as it takes horizontal heat advection into account and demonstrates its importance in modifying the limit.  This part is mentioned in the discussion section in Section 4.1 (lines 974ff).

With these revisions, I hope that this will help to satisfy the reviewer comments (although, honestly, I am not so sure if that is at all possible with reviewer #3).

Best,
Axel

**Response to Editor**

*Editor:* *The reviewers have diverging opinions on the revised manuscript, and I believe that some further refinements may be needed before taking a final decision on the manuscript.*

*Response:* I thank the reviewers for their efforts on reviewing the revised manuscript.

*Action:* I have addressed all comments in the responses below, mostly by providing more explanations. The new modifications in the re-revised manuscript are marked in **cyan**.

*Editor:* *In particular, one of the reviewers, although recommending acceptance, argues that this is more of a perspective paper than a review, since it favours specific viewpoints as opposed to providing an unbiased overview of the field. This is not necessarily a problem per se, and on the contrary may contribute to the value of the paper, as long as this is made clear to the reader - which is not always the case in the current text.*

*Response:* Agreed.

*Action:* I added a few sentences on this after line 105 where the aims of this review are being described and why I think that this biased view may be beneficial.

*Editor:* *The same reviewer also argues that some of the edits you have implemented add text without necessarily providing clear definitions of the concepts you are describing, notably in the case of free energy. While I do not advocate a heavy mathematical treatment of the topic, since this is not the style of this study, some precise definition may aid understanding (e.g. along the lines of how Tailleux (2013) defined exergy in terms of the relatively simple eq. 5).*

*Response:* It is not as simple as expressing free energy in one equation. If one only deals with heat and pressure work, then free energy (or exergy) can be expressed by one equation. But the Earth system deals with different forms of free energy, and some conversions do not involve entropy, or involve different forms of entropy. The free energy in radiation, for instance, is similar to the one for heat, but it includes modifications due to photon pressure, which is different from the pressure of an ideal gas. Hence, the expression for free energy is different as well. The kinetic energy of motion that can be converted further into electric energy using wind turbines (another conversion process) is, again, different.

So instead of providing an equation that may be misused because it is applied to contexts for which it has not been derived for, I think a clarification on the need for a more general, yet qualitative description of free energy is more helpful in the text.

*Action:* As the reviewer did not find the previously revised text helpful, I revised it again (after line 244) and shortened it along the lines described in the response.

*Editor:* *The third reviewer recommends again rejection. While this study is a review of past results from the literature, this does not mean that such results may not be partly at odds with the literature in other subfields, as some of the reviewer comments on e.g. the vertical energy transport argue.*

*Response:* Actually, the results are mostly *not* at odds with literature, particularly regarding the example of vertical energy exchange. In that particular example, the difference concerns the perspective on whether one focuses on the inside of the system, or on the whole system that includes its boundary conditions. The importance of the latter can easily be seen by looking at a heat engine: the laws of thermodynamics tell us the limit of energy generation irrespective of the

details inside the engine, but these limits follow from the boundary conditions, not from the details inside. Likewise, the limits of power are not about adiabatic temperature profiles within the atmosphere, but rather about the entropy exchange at the system boundaries. The reviewer, however, seems to only focus on the interior, and not at the boundaries of the system.

*Action:* In the revision, I aimed to clarify these aspects further on lines 430ff.

*Editor: Relating back to the comment above on a perspective versus review paper, it is important to highlight clearly cases where the proposed principles, presented in peer-reviewed studies, may not fully agree with other equally peer-reviewed results.*

*Response:* Again, it is less a lack of agreement but rather the lack of a system's perspective that may lead to contrasting interpretations. When it comes to stomatal functioning, for instance, interpretations from plant physiologists typically focus on the scale of a leaf, and not on the larger scale. At the larger scale, the environmental controls on evaporation are very well described and established in the context of the Budyko framework. As this larger scale is the focus of the manuscript, the latter is more applicable here.

*Action:* I added clarifications around line 831ff and described the relationship to the Budyko framework more explicitly.

*Editor: Finally, a large part of the ESD readership is likely not to be familiar with plant physiology. It is therefore important to clarify some basic concepts there, and avoid statements that may be understood as being at odds with basic physiological processes.*

*Response:* I refer to the previous two responses, as this comment aims at these.

*Action:* See previous comments.

*Editor: When revising the manuscript, I would encourage you to try to formulate your replies to the comments by the third reviewer in as clear and detailed a fashion as possible, copying in your replies the whole text provided by the reviewer. One of the salient features of ESD is the possibility for the community to follow the review process and discussion between the authors and reviewers, and the format of the replies should facilitate this process.*

*Response:* The responses to the reviewer comments is included below.

*Action:* See below.

**Response to Reviewer #1 (Anonymous)**

*Reviewer:* I carefully read through the revised version of the article "Working at the limit: A review of thermodynamics and optimality of the Earth system" by Axel Kleidon, submitted for consideration to Earth System Dynamics. I found that the suggestions and comments by the other reviewers' and myself have significantly improved the quality of the manuscript.

*Response:* Thank you. I am glad to see that the modifications are seen as an improvement.

*Action:* No action taken.

*Reviewer:* I still think that **some of the figures are a bit too qualitative**, and something could be done to **make them more self-explanatory** and **suitable** for a scientific publication. Given that I do not have specific advice on how this shall be accomplished, I leave it to the author to decide whether they are fine like this or they want to work a bit more on that.

*Response:* I am somewhat confused by this comment. While some figures are taken from previous publications, the ones that are not from publications (Figs. 1, 4, 5, 9, 11, 13, 15) are used to either convey concepts or supplement model formulations described in the main text.

*Action:* I made minor modifications to the captions to describe the purpose of the schematic figures and aimed to better link them to the text. I also slightly adjusted Figure 16 to better match the description in the text. I hope with these modifications, the figures are more self-explanatory.

**Response to Reviewer #3 (Jonas Nycander)**

*Reviewer:* Some changes have been made in response to my comments, but none of them address my main objections. I therefore still recommend rejection. Much of the text consists in lengthy qualitative descriptions of how entropy is produced as a result of various energy transformations. These descriptions are correct, but unsurprising, since everything said follows immediately from basic thermodynamic principles. They also do not lead to any quantitative predictions.

*Response:* It is nice to see that the author thinks that the "lengthly qualitative descriptions" are correct but unsurprising.  My experience over the last decades has been the opposite - that these aspects of entropy and thermodynamics are neither widely known nor recognised.

*Action:* No action taken.

*Reviewer:* The scientific core of the paper is a consideration of a simple model with two boxes. All radiative energy fluxes are assumed to be either given externally or determined by the temperature of the boxes. There is also a turbulent energy flux J between the boxes. … The function f describes the radiative fluxes and the energy budget, while g describes the turbulent flux. Thus, the two functions describe independent physical processes. But, of course, the turbulent flux affects the temperatures, and thereby indirectly the radiative fluxes, as a part of determining the state of the whole system.

*Response:* It may come down to splitting hairs when it comes to define what an independent process is. The presence of turbulent fluxes on land are clearly intimately linked to surface temperature, because it is mostly buoyancy developing at the surface that drives them, and buoyancy on the other hand develops because the surface is warmer than the overlying air.  One can formulate these separately, but they are clearly not independent from each other.

*Action:* None taken.

*Reviewer:* Turbulent transport is in general a complex process that is difficult to describe. The author claims that this is not needed, and that J can instead be determined by maximizing the free energy production, subject to the Carnot constraint (the 'optimum principle'). If the ΔT is much smaller than the absolute temperature, this means that the product JΔT = Jf(J) should be maximized.

*Response:* The author does not claim anywhere in the manuscript that turbulence is a simple process.  As a matter of fact, it may be so complex that in the end, the only relevant constraint is how much free energy is generated to drive turbulent dissipation, because this limit is not set by turbulence, but rather by the boundary conditions of the system.  The Carnot limit explains why the outcome is then very simple:  This is because the limit of free energy generation is only described by the conditions at the system's boundary.  The derivation of the Carnot limit does not require any information of the complexity of the interior of the heat engine that it deals with.  Similarly, one can understand the success of maximum power - that the dynamics are governed by the constraints imposed by the boundary conditions, for which the surface plays a pivotal role.

*Action:* I have added this interpretation to the end of Section 2.4 (lines 440ff).  I have also added and emphasized that the likely interpretation of how this maximization is achieved is through the complexity in turbulence (Section 2.5, line 475).

*Reviewer:* The only support for the optimum principle is that it gives good agreement with observations. In the case of meridional heat transport in the atmosphere we have, according to the manuscript, $\Delta T_{max} \approx 60$ K, which gives the prediction $\Delta T \approx 30$ K. This should be compared to the observed value 20 K. I don't find this agreement very impressive, given that we know a priori that $\Delta T$ lies between 0 K and 60 K. Moreover, it is clear that many factors, for example the planetary rotation rate, affect the turbulent transport, and therefore the function g, without affecting the function f. Thus, changing the rotation rate will change the state of the system, while the proposed optimum solution remains the same.

*Response:* The example given in Section 3.1 is not the only support for the optimum principle - section 2.4, 3.2, and 3.3 provide further support, as well as the new added reference to Ghausi et al (2023). Further support comes from climate model simulations that have been done previously (e.g., on the related MEP hypothesis, Kleidon et al., 2003, 2006 - cited in the text in Sections 2.5 and 3.1). In Section 3, however, I have written explicitly that (lines 514ff) "*I will specifically use models that are as simple as possible to describe these examples. The motivation for this simplicity in formulation is to provide transparency and accessibility to the reader, and to not obscure outcomes with complex mathematical formulations.*" In other words, the manuscript favours insights over precision, and this is clearly stated in the beginning of the Section.

Regarding the role of rotation rate, the manuscript already mentions rotation rate in both, section 3.1 and in the discussion section. The thermodynamic limit remains valid irrespective of rotation rate, but for certain conditions, the maximum power limit may not be reached. For Earth, however, this does not seem to be the case.

*Action:* I have slightly reworded the text towards the end of Section 3.1 (Lines 607/608), and added a further reference to support maximum power (Ghausi et al, 2023). The discussion on the role of rotation rate has been clarified (Lines 967ff).

*Reviewer:* The idea that you can determine the state of the system without knowing anything about the mechanisms of turbulent transport is scientifically unsound. This can clearly be seen in the case of vertical energy transport. Here, $\Delta T_{max}$ is the temperature difference given by a pure radiation balance. … This result is different from the established theory, which says that the vertical temperature gradient is close to the adiabatic lapse rate. The mechanism is that the turbulent transport is negligible if the atmosphere is stably stratified, and increases very rapidly if the temperature decreases upward faster than the threshold for convective instability. Translating this to the box model used in the manuscript, this means that the function g($\Delta T$) in eq. (2) is essentially zero for $\Delta T < \Delta T_{conv}$ (where $\Delta T_{conv}$ is the threshold for convective instability), and increases very rapidly for $\Delta T > \Delta T_{conv}$. The solution of the problem is then essentially that $\Delta T$ equals the smallest of $\Delta T_{max}$ and $\Delta T_{conv}$. Thus, if $\Delta T_{max} > \Delta T_{conv}$ (as in the real troposphere) the state is entirely determined by the mechanism of turbulent transport, which is ignored by the optimum principle. The simple theory outlined above is a corner stone of climate science since many decades, and it is based on an clear physical mechanism. If you want to replace it by another theory, which lacks physical motivation, it is not enough to show that its prediction of $\Delta T$ agrees roughly with observations. You must explicitly compare it with the established theory, and show that it gives a better prediction.

*Response:* The claim that the approach is unsound is not justified. The Carnot limit of a heat engine can be derived without the detailed knowledge of the inside of the heat engine and how it actually works. This is because the Carnot limit follows directly from thermodynamic constraints at the system's boundary, irrespective of how the engine functions. The details that the reviewer describes, e.g., regarding the adiabatic lapse rate, deal with the interior organization of the atmospheric heat engine that are not being considered in the maximum power approach. The maximum power approach uses the radiative boundary conditions of the surface-atmosphere

system (i.e., the radiative heating at the surface and the radiative cooling of the atmosphere), in addition to the strong interaction with surface temperature, but not with the interior organization of the heat engine within the convective boundary layer. Furthermore, it seems that the reviewer neglects the fact that the presence of unstable vs. stable conditions - at least on land - are predominantly caused by the presence or absence of solar radiative heating of the surface. In other words, if these established concepts are viewed in a more holistic way that encompasses the whole system, then one notices that there is no contradiction between what the reviewer writes and the applicability of the maximum power limit.

*Action:* I have added an extra paragraph at the end of Section 2.4 along this explanation (lines 430ff). I have also added another reference (Ghausi et al., 2023, https://www.doi.org/10.1073/pnas.2220400120) in which the maximum power limit is evaluated at greater detail and which shows how remarkably well this approach can describe observed temperature variations.

*Reviewer:* *The discussion of photosynthesis seems to imply that the evaporation somehow drives the photosynthesis. That is a strange idea, since increased evaporation is in general detrimental for plant growth. The text is not very clear on this, but in the reply to my previous comments the author rejects my comment that the main resistance to $CO_2$ transport is in the stomata, and is caused by the need to save water. He also states that "stomata play very little role in controlling evaporation rates".*

*Response:* The carbon uptake by plants is intimately connected to water loss and evaporation - they are not independent processes. I am not sure why this should be a strange idea. It relates to the need of plants for carbon dioxide, which plants need to take up from the air. This is very well established. It is also very well established in hydrology that land surfaces typically evaporate near their potential rate set by energy availability when unconstrained by water availability, as reflected in the common Budyko framework. Stomata are surely the way by which the gas exchange takes place, and stomata react when soil water becomes limiting to protect the plants, but their primary purpose is not to save water.

*Action:* I have rewritten and added text (lines 827ff) to clarify that climatological evaporation rates and therefore gas exchange are set mainly by physical factors, not stomata.

*Reviewer:* *Restricting water loss while at the same time allowing CO2 to enter the leaves is the reason for the existence of the stomata. This can be seen in any text book, for example 'Plant Physiology', 3rd ed, Lincoln Taiz and Eduardo Zeiger, p. 59 (open source). A quick web search reveals similar statements in the abstract or opening paragraph of a large number of scientific articles. Here are two examples: "Almost all water used for plant growth is lost to the atmosphere by transpiration through stomatal pores on the leaf epidermis. By altering stomatal pore apertures, plants are able to optimize their CO2 uptake for photosynthesis while minimizing water loss." (L.T. Bertolini et al, Front. Plant Sci., 2019, vol. 10, https://doi.org/10.3389/fpls.2019.00225). "In order for plants to function efficiently, they must balance gaseous exchange between inside and outside the leaf to maximize CO2 uptake for photosynthetic carbon assimilation (A) and to minimize water loss through transpiration. Stomata are the 'gatekeepers' responsible for all gaseous diffusion, and they adjust to both internal and external environmental stimuli governing $CO_2$ uptake and water loss. " (T. Lawson and M.R.Blatt, Plant Physiol., 2014, vol.164, pp 1556–1570.)*

*Response:* Even if this is published in the scientific literature, these statements are incorrect. Plants do not simultaneously maximize CO2 uptake while minimizing evaporative loss. This incorrect perception may result from the lack of a system's perspective when focussing on the gas exchange of an isolated leaf. When looking at the broader scale of the land-atmosphere system, it is well known and confirmed with observations that land surfaces essentially evaporate at their

potential rates if water availability does not act as an additional constraint. This is, for instance, reflected in the Budyko framework (see previous comment).

**Action:** I have added text to clarify that this is not the case (see also previous comment). I have also added a few explicit sentences on this misinterpretation (lines 905ff): "What this does not imply, however, is that vegetation maximizes carbon gain while minimizing water loss, as it is sometimes incorrectly stated in the plant-ecophysiological literature." I have, however, decided to not include the references provided by the reviewer. This misconception is widespread, and I do not want to single out one or two references for this notion.

**Reviewer:** *This means that the plants need to restrict the flux of $CO_2$ when access to water is limited. The facts that the water use efficiency is fairly constant across ecosystems, and that the energy efficiency of photosynthesis is far below the theoretical limit, imply that this regime of simultaneous water and $CO_2$ limitation is normal. The plants handle this by regulating the size of the stomata openings.*

**Response:** Yes, plants do reduce their stomatal conductance in the presence of water stress. In the climatological mean and for natural ecosystems, this water limitation is accounted for by limiting evaporation to the precipitation rate in water-limited regions. As can be seen by the evaluation in Figure 14, this works rather well - the thermodynamic efficiency of terrestrial ecosystems can be predicted very well in both, humid and arid regions, irrespective of water limitation and specific stomatal functioning. Stomatal effects thus do not need to be taken into account for this insight.

**Action:** As this review is not about the role of stomatal functioning in the presence of water stress, I have decided to not include additional text to clarify this point.